# Genome architecture evolution in an invasive copepod species complex

Zhenyong Du [1] ✉, Johannes Wirtz[2], Yifei Joye Zhou [1], Anna Jenstead[1], Taylor Opgenorth[1], Angelise Puls [1], Cullan Meyer [1], Gregory W. Gelembiuk [1] & Carol Eunmi Lee [1] ✉

Chromosomal fusions are hypothesized to facilitate evolutionary adaptation, but empirical evidence has been scarce. Here, we analyze chromosome-level genome sequences of three sibling species within the copepod *Eurytemora affinis* species complex, known for its remarkable ability to rapidly colonize new habitats. Genomes of this species complex show expansions of ion transport-related gene families, likely related to adaptation to various environmental salinities. Among three genetically distinct sibling species, we discover notable patterns of chromosomal evolution, with chromosomal fusions observed in two different sibling species. As a result of these chromosomal fusions, functionally linked ion transport-related genes located near the telomeres become joined near the newly formed centromeres, where recombination is low. Notably, for the highly invasive *E. carolleeae* and to a lesser extent for *E. gulfia*, the ancient chromosomal fusion sites, especially the centromeres, are significantly enriched with contemporary signatures of selection between saline and freshwater populations. This study uncovers intriguing patterns of genome architecture evolution with potentially important implications for mechanisms of adaptive evolution in response to rapid environmental change.

The recent deluge of high-quality genome sequences is providing unprecedented opportunities to study genome architecture evolution and its impacts on adaptive processes. An emerging question is how genome architecture shapes evolutionary responses, particularly during rapid adaptation to new environments[1–5]. An increasing body of research indicates that adaptive loci within genomes are not randomly distributed but often cluster in specific genomic regions. Theoretical simulations support this hypothesis, suggesting that the clustering of adaptive loci is more likely explained by genomic rearrangements than by the chance establishment of new mutations near existing adaptive loci[6]. Some of these clusters might form "genomic islands" of differentiation, whereas others might constitute "supergenes," which are tightly linked loci that function together to control complex traits[5–12]. Despite this emerging recognition, the evolutionary origins and mechanisms leading to the formation of such genomic features remain largely enigmatic, especially on a genome-wide scale.

Chromosomal evolution, which can encompass large-scale alterations, such as chromosomal fusions, fissions, and other rearrangements (e.g., inversions and translocations), has been proposed as a key mechanism contributing to the clustering of adaptive loci. Theoretical and empirical results, particularly focusing on chromosomal inversions, indicate that the joining of beneficial loci and the repositioning of these loci in regions of low recombination could preserve advantageous allelic combinations favored by natural selection, and thus facilitate adaptation[6–8,10,11,13–15]. As such, ancient chromosomal evolution events can reorganize the genome in ways that subsequently affect selection responses of contemporary populations.

In particular, chromosomal fusions constitute a potentially potent mechanism of facilitating adaptive evolution, by bringing together unlinked coadapted alleles from different chromosomes. By relocating loci to more centralized chromosomal regions, chromosomal fusions can create novel genomic regions with reduced recombination for sets

---

[1]Department of Integrative Biology, University of Wisconsin, Madison, WI, USA. [2]CEFE, CNRS, EPHE, IRD, Montpellier, France. ✉e-mail: zdu53@wisc.edu; carollee@wisc.edu

of beneficial loci[4,10]. However, despite the potential contribution of chromosomal fusions to adaptation, empirical evidence has been very limited. Much more intensive study has focused on chromosomal inversions[7,8,11,12], which can only cluster loci located on the same chromosome.

A few recent empirical studies have explored the evolutionary implications of chromosomal fusions. For instance, comparisons among three species of the nematode *Pristionchus* revealed that chromosomal fusions could repattern chromosome-wide recombination rates of fused chromosomes, with shifts in recombination sites contributing to reproductive isolation[4]. Two additional studies comparing the genomes of three stickleback fish species found that two fused chromosomes in the threespine stickleback *Gasterosteus aculeatus* were significantly enriched with 130 quantitative trait loci (QTL) and genomic signatures of selection related to freshwater adaptation[5,16]. In fritillary butterflies of the genus *Brenthis*, a comparison of genomes of five species found a signature of a strong and recent selective sweep around one chromosomal fusion site among twelve fusions sites observed[17]. Collectively, these findings suggest that chromosomal fusions might contribute to adaptation or become fixed through natural selection, possibly by altering the positions of adaptive loci and the recombination landscape. However, these studies do not provide direct empirical support that the fusion events are joining specific loci under selection or that the chromosomal fusion sites are under natural selection. As such, a key question remains on whether these chromosomal fusion events are associated with adaptive evolution.

The calanoid copepod *Eurytemora affinis* species complex (Fig. 1a) presents a compelling model to examine the hypothesis that chromosomal fusions might facilitate adaptation by impacting the arrangement of loci under selection. Populations of the *E. affinis* complex form an enormous biomass in many estuaries and coastal habitats, serving as dominant grazers of algae in the Northern Hemisphere and crucial food sources for many important fisheries[18–20]. Despite considerable morphological stasis (Fig. 1d, e)[21], large genetic divergences in mitochondrial gene sequences separate at least six geographically distinct sibling species (clades) within this species complex (Fig. 1b), with idiosyncratic patterns of reproductive isolation among the clades[22,23].

The *E. affinis* complex currently serves as a valuable model system for investigating evolutionary adaptation during habitat transitions, both during biological invasions and in response to climate change[24–33]. Populations of this species complex reside across a wide range of salinities, and several populations have demonstrated the capacity to evolve in response to changes in habitat salinity[29–31,34–36]. Notably, certain populations from particular clades are known for their remarkable capacity to adapt from saline to freshwater environments[23,24,36,37]. Specifically, populations from the sibling species *E. carolleeae* (Atlantic clade)[38], *E. gulfia* sp. nov. (Gulf clade), *E. affinis* proper (Europe clade)[39], and the Asia clades[24,40] have independently invaded freshwater habitats from saline ancestral habitats within a time span of only a few decades, likely through the transport and dumping of ship ballast water (Fig. 1c)[24,30,41–43]. For example, estuarine populations of *E. carolleeae* were introduced into the North American Great Lakes approximately 66 years ago, following the opening of the St. Lawrence Seaway[24,42]. Similarly, within the past 80 years, coastal Gulf of Mexico populations of *E. gulfia* were transported into inland freshwater reservoirs throughout the Mississippi drainage system[24,41]. In Europe, saline populations of *E. affinis* proper adapted to freshwater conditions after becoming trapped following the diking of a saltwater bay and its transformation into freshwater lakes IJsselmeer and Markermeer over 6 years[24,43].

An intriguing feature of this species complex is that the different sibling species (clades) vary in their capacity to invade and adapt to new salinities, with some clades completely lacking the ability to colonize different habitats[22,44]. This variation in capacity raises intriguing questions regarding the impacts of genome architecture on the evolutionary potential to adapt to new environments[29,31,36,37,45]. Even among the *E. affinis* complex clades that can colonize new habitats, their ability to transition to new salinities varies considerably. In particular, populations from *E. carolleeae* have been the most successful as invaders, both into saline and freshwater habitats[24,44]. For instance, in addition to invading inland lakes[24], Atlantic coastal populations of *E. carolleeae* have invaded San Francisco Bay[24] and are now displacing *E. affinis* proper populations in the Baltic Sea and other parts of Europe[26–28] (Fig. 1c, red triangles).

Our previous study generated the first chromosome-level genome for *E. carolleeae*, contributing important insights into the genome architecture of the *E. affinis* complex[46,47]. In *E. carolleeae* we found a peculiar genome architecture, which was characterized by a low number of chromosomes of only four, an extraordinary expansion of ion transport-related gene families, and significantly greater than expected clustering of ion transport-related genes on the chromosomes[46]. The low chromosome number of *E. carolleeae*, relative to other copepod species, suggested that chromosomal fusion events might have occurred during its evolutionary history[46,48,49]. The significant clustering of ion transport-related genes within the *E. carolleeae* genome might facilitate the coexpression of functionally related genes and could enable coadapted alleles at different genes to be inherited together and undergo selection as a unit[50]. As such, we hypothesized that ancient chromosomal fusions in the *E. affinis* complex facilitated adaptation in the past by bringing together beneficial loci into close physical linkage. In addition, such linkages, which reduce recombination and help maintain co-adapted gene complexes, might continue to be beneficial and constitute the targets of natural selection during salinity change in contemporary populations[5,6,10,29,46].

Thus, to address these hypotheses and explore patterns of genome architecture evolution, we examined the association between ancient chromosomal evolution events, particularly chromosomal fusions, and genome-wide signatures of natural selection in contemporary populations. As such, our specific goals were to explore patterns of chromosomal evolution across genomes from three sibling species of the *E. affinis* complex to determine how functionally beneficial loci (under selection in the past) might have become repositioned following the ancient fusion events. In addition, we determined whether the ancient chromosomal fusion sites show an enrichment of genomic signatures of natural selection (e.g., allele frequency shifts, association with salinity, LD) in contemporary populations undergoing rapid habitat salinity change.

To accomplish these goals, we generated high-quality chromosome-level genomes of two additional sibling species within the *E. affinis* complex, specifically *E. gulfia* sp. nov. and *E. affinis* proper. These genomes were derived from inbred lines generated through 10 to 20 generations of full-sibling mating under controlled laboratory conditions. When combined with the recently sequenced genome of *E. carolleeae*[46], these new genome sequences provided us with a comparative framework to investigate patterns of genome architecture evolution across the species complex.

While a few recent comparative studies have suggested the evolutionary importance of chromosomal fusions[4,5,17], direct evidence linking fusion sites with signatures of selection had been lacking. This study makes strong inferences by explicitly tracking the movement of specific adaptive genes (the loci under selection) across the different genomes to determine how the chromosomal fusions affect their repositioning. In addition, we test all the fusion sites against the genomic background for population genomic signatures of selection. Thus, this study is noteworthy for making the association between genome architecture evolution and the rearrangement of adaptive loci, particularly for those under selection in response to rapid habitat change.

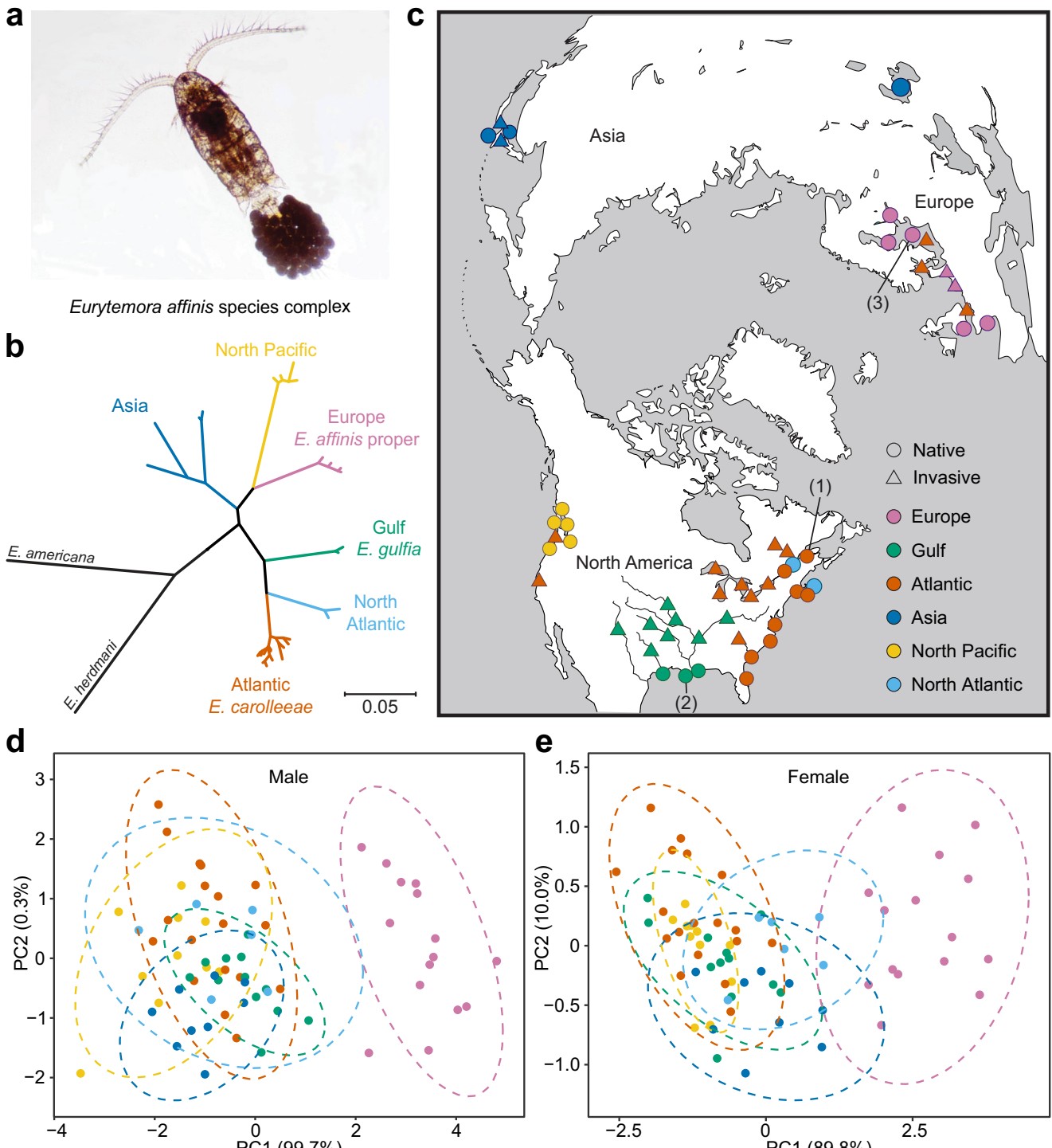

**Fig. 1 | Phylogeographic patterns and morphological stasis of the *Eurytemora affinis* species complex. a** Image of an adult ovigerous female *E. affinis* complex copepod from the North Pacific clade, collected from the Columbia River estuary, Oregon, USA. Photograph by Carol Lee. **b** Mitochondrial phylogeny of the *E. affinis* complex based on concatenated *COI* and *16S rRNA* gene sequences constructed using MrBayes[77]. Maximum-likelihood and Bayesian inference phylogenetic reconstructions using *COI* and *16S rRNA* sequence data from our previous study[22] (Supplementary Fig. 1) corroborated the phylogenetic topology and divergence among the six clades (sibling species) found previously[22]. *E. americana* and *E. herdmani* are outgroup species. **c** Geographic distribution of the genetically distinct clades within the *E. affinis* complex. Colors of dots refer to the genetically distinct clades in the phylogeny in **b**. The circles represent native saline populations of the distinct clades, while the triangles represent populations invading both saline and freshwater habitats. The numbers in parentheses correspond to the sampling locations of our reference genome populations used in this study: (1) Baie de L'Isle Verte, St. Lawrence Estuary, Quebec, Canada (*E. carolleeae*); (2) Blue Hammock Bayou, Louisiana, USA (*E. gulfia*); (3) Stockholm, Sweden (*E. affinis* proper). Morphological stasis among the clades of *E. affinis* complex, as indicated by PCA plots showing the morphological variation in secondary sex characters among male (**d**) and female (**e**) copepods of the *E. affinis* complex using data from Lee and Frost[21]. Only the Europe clade (*E. affinis* proper, purple) shows deviation from the cloud of points. The colors on the plots refer to the clades in the phylogeny in **b**. Source data are provided as a Source Data file.

## Results

### Distinct genome architectures of the three sibling species of the *E. affinis* complex

In this study, our chromosome-level genome assemblies revealed strikingly different chromosome numbers for three sibling species (clades) of the *E. affinis* complex (Fig. 1). Using Hi-C genome sequencing, we reconstructed 15 chromosomes in the genome of *E. affinis* proper (Europe clade, Fig. 2e, f) and seven chromosomes in the genome of *E. gulfia* sp. nov. (Gulf clade, Fig. 2c, d). Karyotyping using fluorescence microscopy confirmed these chromosome numbers for these two sibling species (Supplementary Fig. 2a, b). In addition, our previous genomic analysis and karyotyping for *E. carolleeae* (Atlantic clade)[46] had revealed the presence of only four chromosomes (Fig. 2a, b and Supplementary Fig. 2c).

To assemble genomes for *E. gulfia* (Gulf clade) and *E. affinis* proper (Europe clade), we utilized high-coverage PacBio Continuous Long Read (CLR) sequencing, Illumina short-read sequencing, and Hi-C technology (Supplementary Data 1). We obtained genome sizes of 521.1 Mb for *E. gulfia* (contig N50 = 3.6 Mb, scaffold N50 = 42.5 Mb) and 670.9 Mb for *E. affinis* proper (contig N50 = 1.0 Mb, scaffold N50 = 69.6 Mb). These results were consistent with the estimated genome size of 502 Mb (*E. gulfia*) and 687 Mb (*E. affinis* proper) for these two sibling species based on k-mer analysis (Supplementary Fig. 3). The Benchmark of Universal Single-Copy Orthologs (BUSCO) analyses revealed 92% complete BUSCOs from the arthropod odb10 dataset (Supplementary Data 2). Notably, the genome size of *E. gulfia* (521.1 Mb) and that previously reported for *E. carolleeae* (529.3 Mb)[46] were both smaller than the size of the *E. affinis* proper genome (670.9 Mb), yet GC content (33%) was remarkably conserved across these genomes (Supplementary Data 2). These reference genomes stand as the most contiguous and complete calanoid copepod genomes to date, offering an invaluable resource for genomic studies of copepods.

We predicted a total of 22,391 protein-coding genes in the *E. gulfia* genome and 24,176 in the *E. affinis* proper genome (Supplementary Data 2). These gene numbers were comparable with the number of genes we had previously found in the *E. carolleeae* genome (20,262)[46]. These numbers of genes for the sibling species of the *E. affinis* complex exceeded those of other copepod and daphnid species, including the tidepool copepod *Tigriopus californicus* (15,500 genes), the salmon louse *Lepeoptheirus salmonis* (13,081 genes), and the water fleas *Daphnia pulex* (15,295 genes) and *D. magna* (16,891 genes) (Supplementary Data 6).

By integrating our de novo repetitive sequence database with existing public databases, we ascertained that 45.5% of the *E. gulfia* genome and 50.2% of the *E. affinis* proper genome consisted of repetitive sequences (Fig. 2c, e, Supplementary Data 3 and 4). For the three sibling species in the *E. affinis* complex, DNA transposons and Long Terminal Repeat (LTR) elements constituted the largest proportions of repetitive sequences (Supplementary Data 3 and 4).

The annotation of protein-coding genes was based on extensive transcriptome data for the *E. affinis* complex, alongside homologous proteins from other arthropods and ab initio predictions. For both *E. gulfia* and *E. affinis* proper, nearly all genes (22,385 out of 22,391 for *E. gulfia* and 24,117 out of 24,176 for *E. affinis* proper) were functionally annotated using at least one of eight different functional annotation databases (Supplementary Data 5).

### Phylogeny of the *E. affinis* complex and patterns of gene family expansions

To determine the evolutionary relationships among the three sibling species of the *E. affinis* complex, we determined phylogenetic relationships among seven high-quality chromosome-level genomes of copepods and daphnids based on 1153 single-copy ortholog genes (Fig. 3 and Supplementary Data 6). Using a secondary calibration

(183–365 million years ago [Mya]) for the timing of the most recent common ancestor (MRCA) of copepods[51], we estimated the MRCA of the *E. affinis* complex to be around 60 Mya, following the Cretaceous–Paleogene (K–Pg) boundary (Fig. 3). The timing of divergence between *E. carolleeae* and *E. gulfia* was traced back to approximately 45 Mya during the Eocene Epoch (Fig. 3).

Despite the morphological stasis observed in this species complex (Fig. 1d, e)[21], the extended time period of divergence and difference in chromosome number (Fig. 3) support the designation of the three clades as distinct sibling species. Therefore, we formally named the Gulf clade as *E. gulfia* sp. nov., a nomenclature reflective of its native habitat in the Gulf of Mexico region. Additionally, we use the name *E. carolleeae* to refer to the Atlantic clade[38] and *E. affinis* proper to refer to the Europe clade, as *E. affinis* was first described in Europe[39].

At the node leading to the *E. affinis* species complex, we observed an expansion of 699 gene families, encompassing 5574 genes (Supplementary Data 7) and a contraction of 799 gene families (Fig. 3 and Supplementary Data 8). In addition, we found 6886 gene families unique to the *E. affinis* complex genomes and absent in the other copepod and daphnid genomes sampled for this study (Fig. 4; Supplementary Data 9). Of these unique gene families, 2805 gene families comprising 10,902 genes were shared across all three sibling species of the *E. affinis* complex. Additionally, 2243 gene families consisting of 8196 genes were shared between two sibling species of the complex. Conversely, only 341 gene families, encompassing 1851 genes, were lost in *E. affinis* complex but present in the other four copepod and daphnid species (Supplementary Data 10). Notably, the identification of unique and lost gene families in the *E. affinis* complex genomes might be influenced by limited sampling of other calanoid copepod genomes. As such, assessment of gene gains and losses are tentative and dependent on the availability and analysis of genomic data from additional taxa.

Functional annotation of the expanded and unique gene families of the *E. affinis* complex genomes strongly implicated salinity as a major factor imposing selection throughout the evolutionary history of this species complex (Fig. 4, Supplementary Data 7 and 9). To ascertain the functional roles of the expanded and unique gene families in the *E. affinis* complex, we conducted gene function enrichment analysis using Gene Ontology (GO) annotation. This analysis revealed that ion transport activity and ion homeostasis functions formed a substantial proportion of the significantly enriched GO terms in the expanded and unique gene families (Fig. 4). Specifically, of the GO terms enriched in the expanded gene families, ion transport/homeostasis functions constituted 25.0% (133 out of 531) in the Molecular Function (MF) category, 11.8% (311 out of 2645) in the Biological Process (BP) category, and 6.5% (20 out of 310) in the Cellular Component (CC) category (Supplementary Data 7). Moreover, among the top 20 GO terms in each category (based on *P*-values, Fig. 4a and Supplementary Data 7), ion transport/homeostasis comprised 80% (16 out of 20) of the enriched GO terms in the MF, 75% (15 out of 20) in the BP, and 20% (4 out of 20) in the CC categories. The most enriched GO terms included those with key ion regulatory functions, such as "P-type sodium transporter activity" (GO:0008554), "sodium: potassium-exchanging ATPase complex" (GO:0005890), and "sodium ion transmembrane transport" (GO:0035725) (Fig. 4 and Supplementary Data 7).

Similarly, GO enrichment analysis of the unique gene families revealed that ion transport/homeostasis occupied 37.2% (58 out of 156) of the MF category, 17.8% (73 out of 411) of the BP category, and 10.1% (9 out of 89) of the CC category (Supplementary Data 9). Of these, the most enriched GO terms included "potassium ion transmembrane transporter activity" (GO:0015079), "monoatomic ion transmembrane transporter activity" (GO:0015075), "sodium channel complex" (GO:0034706), and "potassium ion transport" (GO:0006813) (Fig. 4b and Supplementary Data 9).

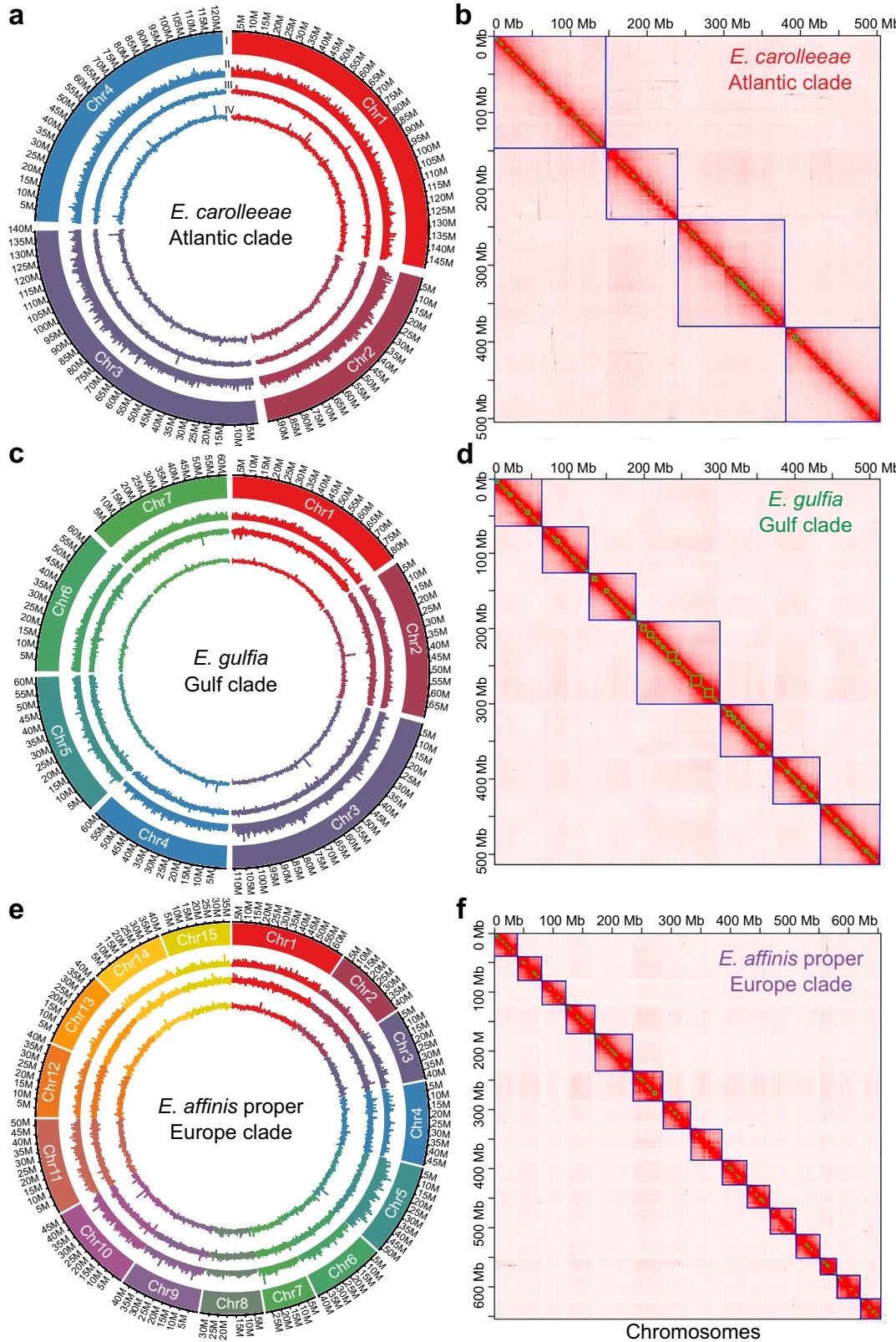

**Fig. 2 | Distinct genome architectures of three sibling species of the *Eurytemora affinis* species complex. a**, **c**, **e** Genomic landscape of each sibling species, depicted in circular diagrams with the following layers (starting from the outer layer): I. Chromosomes on a megabase (Mb) scale; II. Density of protein-coding genes; III. Distribution of GC content; IV. Distribution of repetitive sequences. All distributions except GC content (III) were calculated with 100 kb non-overlapping

sliding windows, while distribution of GC content was calculated with 10 kb sliding windows. **b**, **d**, **f** Hi-C contact maps of genomes of each species, generated by Juicebox software[88]. These maps show the different chromosome numbers assembled in the three species, offering a visual representation of their genomic organization. Source data are provided as a Source Data file.

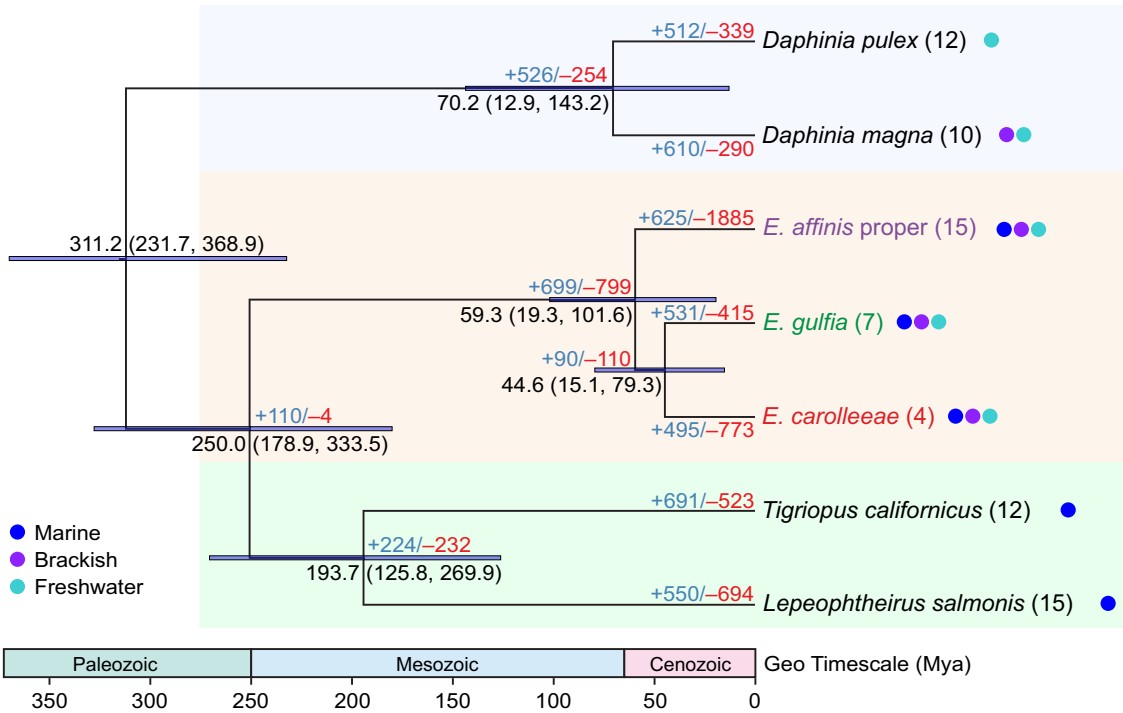

**Fig. 3 | Phylogeny, divergence times, and gene family expansions and contractions in seven copepod and daphnid species, focusing on the *Eurytemora affinis* species complex.** Maximum likelihood phylogenetic reconstruction of seven high-quality genomes was performed using RAxML based on concatenated 1153 single-copy ortholog genes. All nodes show bootstrap values of 100%. The timing of the most recent common ancestor of copepods, 183–365 million years ago (Mya), was calibrated with confidence time intervals retrieved from the Timetree database and applied in MCMCTree[118]. Mean estimated divergence times are shown at each node, with the horizontal blue lines and numbers in parentheses indicating 95% highest posterior density intervals. The numbers of expanded gene families (in blue) and contracted gene families (in red) are shown above the branch tips and next to each node. The chromosome numbers for each species are noted in parentheses next to the taxon names. The habitat types of these species are depicted with color-coded dots.

Based on our functional annotation in this study, the expanded and unique gene families in the *E. affinis* complex genomes included ion transport-related gene families that had been found to be repeatedly under natural selection during saline to freshwater transitions in *E. affinis* complex populations in our prior studies[29–31,34,35]. The expanded and unique gene families in the *E. affinis* complex genomes in this study (Fig. 4) included Na$^+$/K$^+$-ATPase α subunit (NKA-α), Na$^+$/K$^+$-ATPase β subunit (NKA-β), Na$^+$/H$^+$ antiporter (NHA), Na$^+$/H$^+$ exchanger (NHE), Na$^+$,K$^+$,Cl$^-$ cotransporter (NKCC), Vacuolar-type H$^+$ ATPase (VHA), Rh protein (Rh), and Carbonic anhydrase (CA). These ion transport-related gene families are considered to play crucial roles in models of ion uptake from the environment[25] (Figs. 5b, 5c, Supplementary Data 7 and 9). Notably, for these critical ion transport-related gene families *E. affinis* complex genomes contain 6–13 paralogs, whereas other crustacean genomes tend to have fewer paralogs. For example, each of the three sibling species within the *E. affinis* complex contains seven paralogs of the NHA gene family, whereas the tidepool copepod *Tigriopus californicus* and the salmon louse *Lepeoptheirus salmonis* each possess only one paralog. Additionally, the NHA gene family is absent in the genomes of the water fleas *Daphnia pulex* and *D. magna*.

**Evolutionary history of chromosomal fusions in the *E. affinis* complex**

The divergent genome architectures within the *E. affinis* species complex present an exceptional opportunity to study the process of chromosomal evolution among closely related taxa. We identified instances of chromosomal fusions occurring in two different sibling species, based on phylogenomic analysis, computational reconstruction of the ancestral karyotype, and synteny analysis of the three genomes in this species complex. That is, the four chromosomes identified in *E. carolleeae* (Fig. 2b) and seven chromosomes in *E. gulfia*

(Fig. 2d) were inferred to be products of chromosomal fusion events, likely with some independent fusions, from an ancestor with 15 chromosomes (Fig. 6a).

Our phylogenetic analysis clearly established the basal position of *E. affinis* proper (Europe clade) relative to the other two sibling species we examined within the *E. affinis* complex (*E. carolleeae* and *E. gulfia*) (Fig. 3). This phylogeny, based on 1153 of single-copy ortholog genes, was highly concordant with the mitochondrial phylogeny of *E. affinis* complex based on previous data (Fig. 1b). Thus, the 15-chromosome genome observed in *E. affinis* proper (Fig. 2f) likely represents the ancestral karyotype, basally positioned relative to the seven- and four-chromosome genomes of the other two sibling species of the *E. affinis* complex. We gained further support for this inference using the ancestral chromosome reconstruction tools Agora[52] and ANGES[53]. These computational tools utilize current genome assemblies to model the most likely chromosomal configurations of ancestral states. These analyses revealed that the ancestral genome of *E. carolleeae* and *E. gulfia* likely possessed 15 chromosomes, while the ancestral genome of the entire *E. affinis* complex likely contained 21 chromosomes (Fig. 6a).

The chromosomal fusions from an ancestral state of 15 chromosomes to seven (*E. gulfia*) and to four (*E. carolleeae*) chromosomes were evident by mapping the most parsimonious pathways of fusions (Fig. 6b, c). We found highly conserved arrangements of syntenic blocks between *E. affinis* proper and *E. carolleeae* (258 syngenetic blocks containing 10,268 pairs of genes, Fig. 6b) and between *E. affinis* proper and *E. gulfia* genomes (224 syngenetic blocks containing 11,198 pairs of genes, Fig. 6c). Mapping fusions from *E. affinis* proper to *E. carolleeae* (Fig. 6b) and from *E. affinis* proper to *E. gulfia* (Fig. 6c) required only the simple joining of chromosomes, with no chromosomal fissions. For example, in *E. carolleeae*, four chromosomes from the ancestral genome (of 15 chromosomes) were fused together to

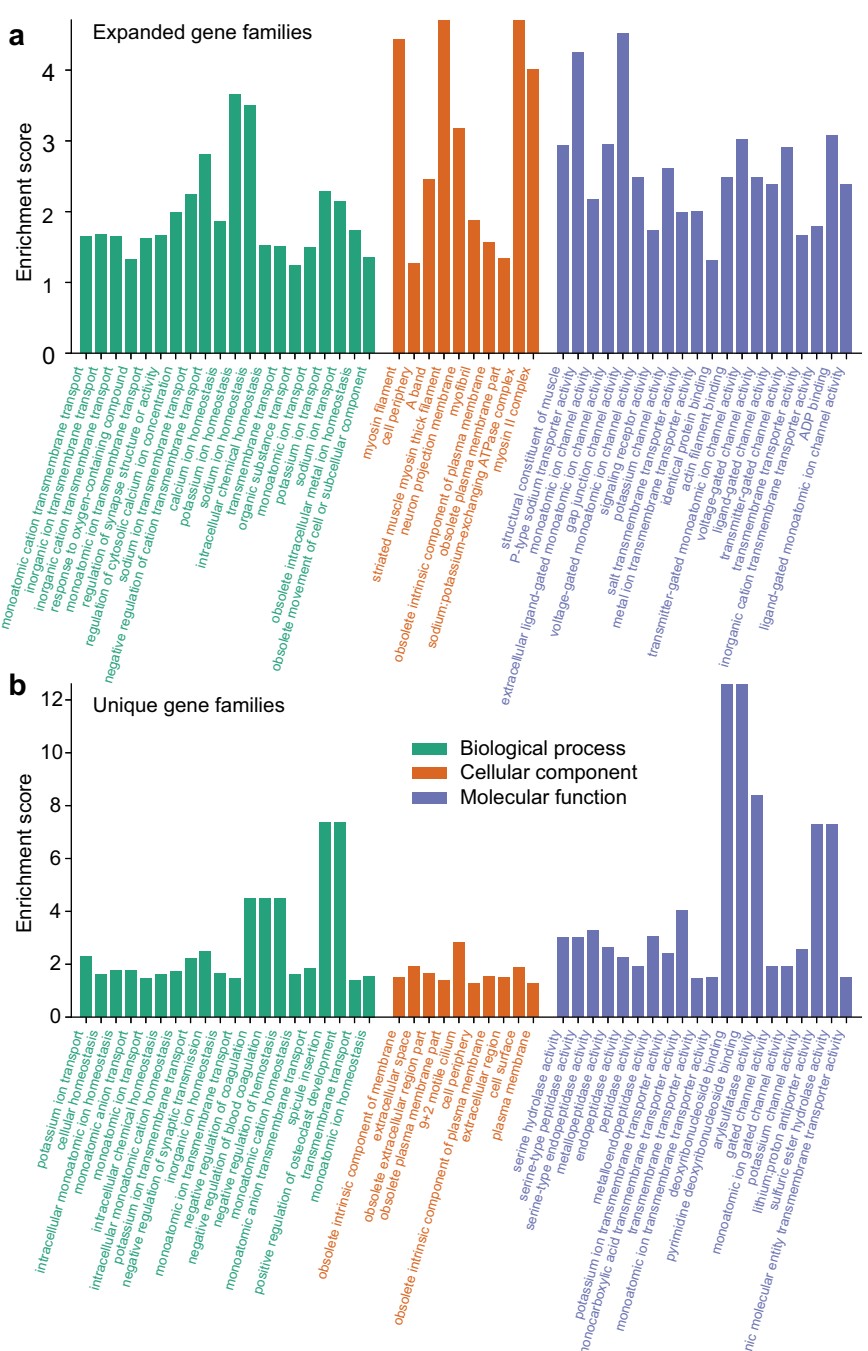

**Fig. 4 | Enriched Gene Ontology (GO) terms in expanded and unique gene families of the *Eurytemora affinis* species complex. a** Significantly enriched GO terms among the expanded sets of gene families. **b** Significantly enriched GO terms among the unique sets of gene families. The GO terms are arranged in ascending order of their *P*-values (left to right), indicating the strength of their association with the expanded and unique gene sets within each category. These figures include the top 20 GO terms for the Biological Process and Molecular Function categories and the top 15 for the Cellular Component category, with higher *P*-values towards the right in each category. GO term enrichments were tested using hypergeometric tests (equivalent to one-sided Fisher's exact tests for over-representation), and *P*-values were adjusted for multiple comparisons using the Benjamini−Hochberg false discovery rate (FDR) method. All enriched GO terms, including their respective *P*-values and functional annotations, are listed in Supplementary Data 7 and 9. Source data are provided as a Source Data file.

form each chromosome, except for Chromosome 2, which was formed by fusing three chromosomes (Fig. 6a, b).

In sharp contrast, the genomes of *E. carolleeae* and *E. gulfia* showed much lower conservation in the arrangement of syntenic blocks, especially between *E. carolleeae*'s Chromosomes 2, 3 and *E. gulfia*'s Chromosomes 3, 4, 5 (Fig. 6d). This discordant alignment between the *E. carolleeae* and *E. gulfia* genomes would require extensive chromosomal rearrangements to reconcile the misalignments

between the two genomes, requiring five fusions and at least two fission events (Fig. 6d). Thus, the evidence does not support a scenario where initial fusion events from the ancestral fifteen to seven chromosomes were followed by subsequent fusions from seven to four chromosomes.

While the fusions involving *E. carolleeae*'s Chromosomes 2, 3 and *E. gulfia*'s Chromosomes 3, 4, 5 likely arose independently in each sibling species, it remains possible that certain fusions were shared and

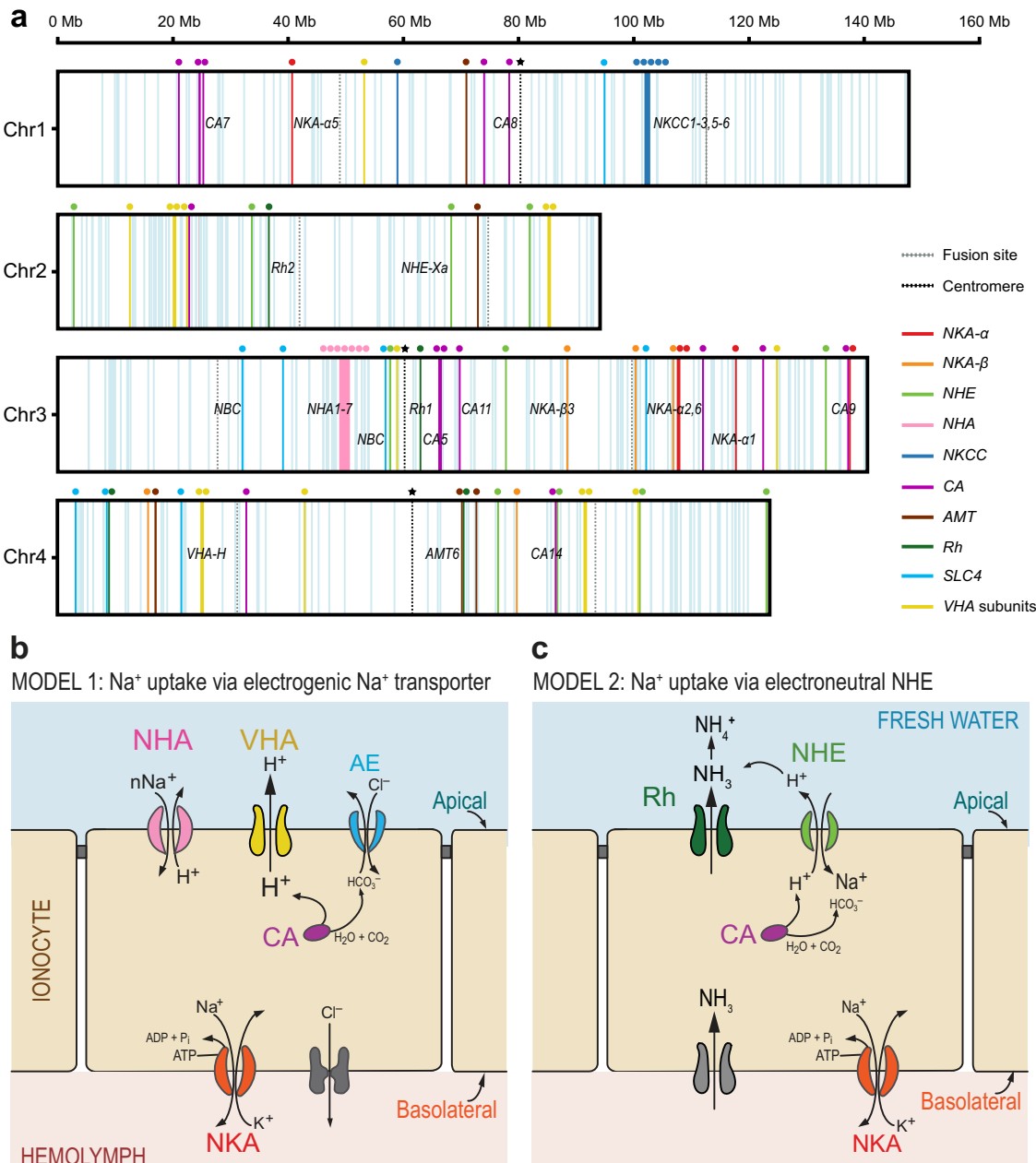

**Fig. 5 | Localization of ion transport-related genes in the *Eurytemora carolleeae* genome and hypothetical models of ion uptake from fresh water. a** Localization of ion transport-related gene paralogs in the *E. carolleeae* genome. Vertical light blue lines represent 490 genes annotated with cation or anion transporting functions. Colored vertical lines and circles indicate 80 key genes that have shown evolutionary shifts in gene expression and/or selection signatures in our previous studies[30–32,35,36,47], and all known paralogs and subunits from the same gene families. These genes include those thought to be involved in hypothetical models of ion uptake (**b**, **c**)[36]. Vertical dashed lines marked with stars denote the positions of centromeres. The positions of these centromeres on Chr 1, 3, and 4 were determined through the Hi-C analysis of the *E. carolleeae* genome (Supplementary Fig. 4)[47]. The centromere position for Chr 2 was not identified and might be absent. (**b**, **c**) Hypothetical models of ion uptake in freshwater habitats performed by ion transporters within epithelial ion transporting cells (ionocytes, beige). In

freshwater habitats (blue), the concentrations of ions are very low, such that the ions must be transported into the ionocyte (beige) against very steep concentration gradients. **b** Model 1: VHA (yellow) pumps out protons ($H^+$) into the freshwater environment to generate a proton gradient. Using this proton gradient, $Na^+$ is transported into the cell through a $Na^+$ transporter (likely NHA, pink). CA (fuchsia) produces protons for VHA. (**c**) Model 2: An ammonia transporter Rh protein (dark green) exports $NH_3$ out of the cell and then this $NH_3$ reacts with $H^+$ to form $NH_4^+$. This consumption of extracellular $H^+$ causes NHE (light green) to export $H^+$ out in exchange for the import of $Na^+$ into the cell. CA produces protons for NHE. In both models, NKA (red) transports $Na^+$ to the hemolymph (pink). These models are not comprehensive for all tissues or taxa and are not mutually exclusive. NKA = $Na^+/K^+$ ATPase, NHE = $Na^+/H^+$ exchanger, NHA = $Na^+/H^+$ antiporter, CA = Carbonic anhydrase, Rh = Rh protein, VHA = Vacuolar-type $H^+$ ATPase. Figure adapted from Lee et al. (2022)[26].

inherited from a more recent common ancestor. For example, *E. carolleeae*'s Chromosomes 1, 4 and *E. gulfia*'s Chromosomes 1, 2, 6, 7 (Fig. 6d) might have shared chromosomal fusions originating from an intermediate ancestor possessing nine or ten chromosomes (Fig. 6a, *n* = ?). Notably, a ten-chromosome karyotype was documented in a

population from Rhode Island, USA that may belong to the North Atlantic clade[54], which is more closely related to *E. carolleeae* and *E. gulfia* than to *E. affinis* proper[22,24] (Fig. 1).

Despite the lower conservation of syntenic block positions between the *E. carolleeae* and *E. gulfia* genomes (124 syngenetic blocks,

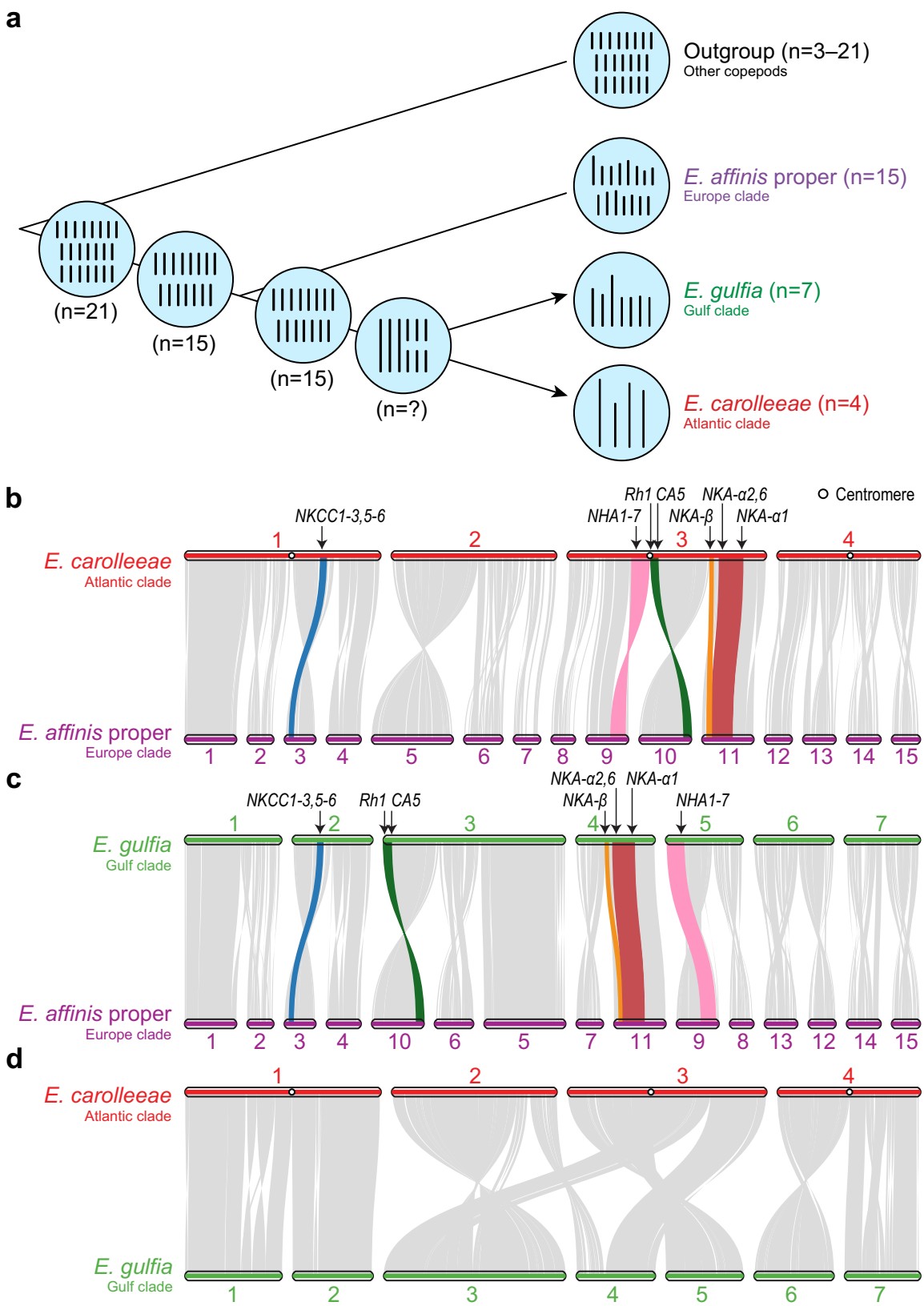

14,019 pairs of orthologous genes), these genomes showed greater genome-wide nucleotide similarity with each other than with the *E. affinis* proper genome. Specifically, 357.2 Mb of the *E. carolleeae* genome (67.5%) aligned with 356.2 Mb of the *E. gulfia* genome (68.4%), showing an average nucleotide similarity of 90.1%. In contrast, only 137.9 Mb of the *E. carolleeae* genome (26.1%) aligned with 138.3 Mb of

the *E. affinis* proper genome (20.6%), with an average nucleotide similarity of 87.4%. Likewise, only 128.1 Mb of the *E. gulfia* genome (24.6%) aligned with 128.5 Mb of the *E. affinis* proper genome (19.2%), with an average nucleotide similarity of 87.5%. These results further support that *E. carolleeae* and *E. gulfia* have diverged more recently from a closely related common ancestor (closer than *E. affinis* proper),

**Fig. 6 | Evolutionary history of chromosomal fusions and synteny patterns within the *Eurytemora affinis* species complex. a** Evolutionary reconstruction of chromosomal fusion events across the *E. affinis* complex. The blue circles show the distinct karyotypes for different sibling species and internal nodes of the *E. affinis* complex and outgroup taxa. The black lines within these circles denote the number of chromosomes in each species, which are also shown in parentheses next to the species names. The chromosome numbers for the internal nodes were reconstructed using Agora[53] and ANGES[54] software. Chromosome numbers for the outgroup were derived from data reported in our previous study[47]. The arrows indicate the chromosomal fusion events leading to *E. gulfia* and *E. carolleeae*. **b**–**d** Syntenic relationships between genomes of *E. carolleeae*, *E. gulfia*, and *E. affinis* proper, showing patterns of chromosomal fusions and rearrangements. Also shown are a

few examples of ion transport-related genes becoming aggregated following the chromosomal fusion events. Syntenic relationships between the genomes of (**b**) *E. affinis* proper and *E. carolleeae*, (**c**) *E. affinis* proper and *E. gulfia*, and (**d**) *E. carolleeae* and *E. gulfia*. Grey ribbons indicate conserved syntenic blocks shared between the genomes, while colored ribbons indicate blocks containing ion transport-related gene clusters, including those containing *NHA*, *NKCC*, *NKA-α*, *NKA-β*, *Rh*, and *CA* paralogs. The circles within the *E. carolleeae* chromosomes indicate the positions of the centromeres. The syntenic relationships between the genomes support chromosomal fusions from *E. affinis* proper to the *E. carolleeae* and *E. gulfia* genomes. The much lower synteny between the *E. carolleeae* and *E. gulfia* genomes makes it far less likely that chromosomal fusions occurred between them. Source data are provided as a Source Data file.

likely with 15 chromosomes or an unknown intermediate number of chromosomes (Fig. 6a).

## Chromosomal fusions reposition ion transport-related genes within genomes

Within the *E. affinis* species complex, we found that chromosomal fusions resulted in the rearrangement of key ion transport-related genes across the chromosomes (Fig. 6b–d). Some of these fusions created extended linked gene clusters (potentially forming supergenes) and often repositioned these genes within regions of reduced recombination, often at or near the centromeres. Many of these repositioned genes belong to expanded and unique gene families in the *E. affinis* complex, with critical functions likely associated with ion uptake from the environment[25] (Fig. 5b), including *NKA-α*, *NKA-β*, *NHE*, *NHA*, *NKCC*, *CA*, *AMT*, *Rh*, *SLC4* (*AE*, *NBC*, *NDCBE*), and *VHA* subunits.

For example, in the *E. carolleeae* genome, chromosomal rearrangements resulted in the creation of hotspot regions with high densities of key ion transport-related gene paralogs and subunits, especially on Chromosome 3 (Figs. 5a and 6b). Specifically, an inverted part of *E. affinis* proper's Chromosome 9, containing the *NHA* gene family (along with *NBC*, *NHE*, *VHA*), and the inverted *E. affinis* proper's Chromosome 10, containing *Rh* protein and *CA*, became fused to join these syntenic blocks near the centromere of *E. carolleeae*'s Chromosome 3 (Fig. 6b). In addition, the syntenic block containing *NKA-β* on the edge of *E. affinis* proper's Chromosome 11 moved closer toward the center of *E. carolleeae*'s Chromosome 3 (Fig. 6b). We had previously observed that these ion transport-related gene paralogs and subunits are distributed unevenly across chromosomes of the *E. carolleeae* genome (Fig. 5a), deviating significantly from a uniform distribution and tending to cluster more than expected by chance[46].

We found that the chromosomal fusions often resulted in the significant repositioning of gene clusters of key ion transport-related genes from the edges toward more central portions of the chromosomes. Specifically, the chromosomal fusions resulted in significant shifts from the edges of the *E. affinis* proper chromosomes toward the central regions of the *E. carolleeae* chromosomes for 25 ion transport-related gene paralogs and subunits, which had been identified as targets of natural selection in our prior studies[29,30]. The positions of these genes were indicated by their distances from the nearest edges of each chromosome divided by the chromosome length, with values ranging from 0% (edges of chromosome) to 50% (center of chromosome). These 25 genes exhibited significantly higher mean and median position values in the *E. carolleeae* genome, indicating more central positioning within chromosomes, relative to their positions in the *E. affinis* proper genome (Mean: 32.8% vs. 23.5%; Welch's *t*-test, $t = 2.91$, $P = 5.47e\text{-}3$; Median: 35.4% vs. 16.5%; Mann–Whitney *U* test, $U = 445$, $P = 1.04e\text{-}2$; Supplementary Data 11). For the *E. gulfia* genome, there was no significant repositioning of these 25 genes toward the center of chromosomes, as these genes did not exhibit significantly different mean and median position values relative to those in the *E. affinis* proper genome (Mean: 22.4% vs. 23.5%; Welch's *t*-test, $t = 45.9$, $P = 0.724$; Median: 19.6% vs. 16.5%; Mann–Whitney *U* test, $U = 296$,

$P = 0.869$; Supplementary Data 11). These results provide robust support for the significant central repositioning of selected ion transport-related genes within chromosomes of the *E. carolleeae* genome following the chromosome fusion events (Fig. 6).

We found that the 25 ion transport-related genes with selection signatures, associated with salinity change in both wild and laboratory populations[29,30], were more centrally located within chromosomes of the *E. carolleeae* genome than other sets of ion transport-related genes lacking such selection signatures. We conducted a comparative analysis of these 25 ion transport-related genes under selection against two sets of ion transport-related genes without selection signatures. These two sets included (**1**) 36 ion transport-related genes lacking selection signatures but belonging to the same gene families as the 25 ion transport-related genes under selection and (**2**) 253 genes annotated with putative ion transporting functions (excluding the 25 ion transport-related genes under selection). We found that the selected 25 genes showed significantly higher mean (32.8% vs. 25.0%; Welch's *t*-test, $t = 2.44$, $P = 0.018$) and median positioning (35.4% vs. 26.2%; Mann–Whitney *U* test, $U = 606$, $P = 0.023$) relative to the 36 gene paralogs and subunits with no selection signatures (Supplementary Data 11). Similarly, relative to the 253 genes with putative ion transporting functions, the 25 selected genes showed significantly higher mean (32.8% vs. 24.5%; Welch's *t*-test, $t = 3.32$, $P = 2.36e\text{-}3$) and median values (35.4% vs. 24.4%; Mann–Whitney *U* test, $U = 4351$, $P = 3.01e\text{-}3$) of chromosomal positioning, also indicating more centralized localization.

To further validate these findings, we compared the 25 ion transport-related genes under selection with randomly chosen subsets of 25 genes from the 253 genes with putative ion transporting functions. Using permutation tests, we executed the Welch's *t*-test and Mann–Whitney *U* test $10^6$ times with randomly selected gene sets. The null hypothesis for this comparison was that the 25 ion transport-related genes with signatures of selection would show no significant difference in mean or median positioning on the chromosomes compared to 25 genes randomly selected from 253 genes with putative ion transporting functions. Results from this test showed that 66.3% of Welch's *t*-tests and 63.4% of Mann–Whitney *U* tests yielded *P*-values less than 0.05, indicating significantly higher mean and median positioning for the 25 selected ion transport-related genes. Together, these results demonstrate that ion transport-related genes under contemporary selection for salinity adaptation tend to be non-randomly centralized within chromosomes of the *E. carolleeae* genome. This positioning, shaped by ancient chromosomal fusion events, likely continues to influence selection responses of contemporary populations (see next section).

## Signatures of selection enriched at the chromosomal fusion sites

We found significant associations between the ancient chromosomal fusion sites in both the *E. carolleeae* and *E. gulfia* genomes and population genomic signatures of selection during contemporary salinity invasions. Our analysis revealed that contemporary population

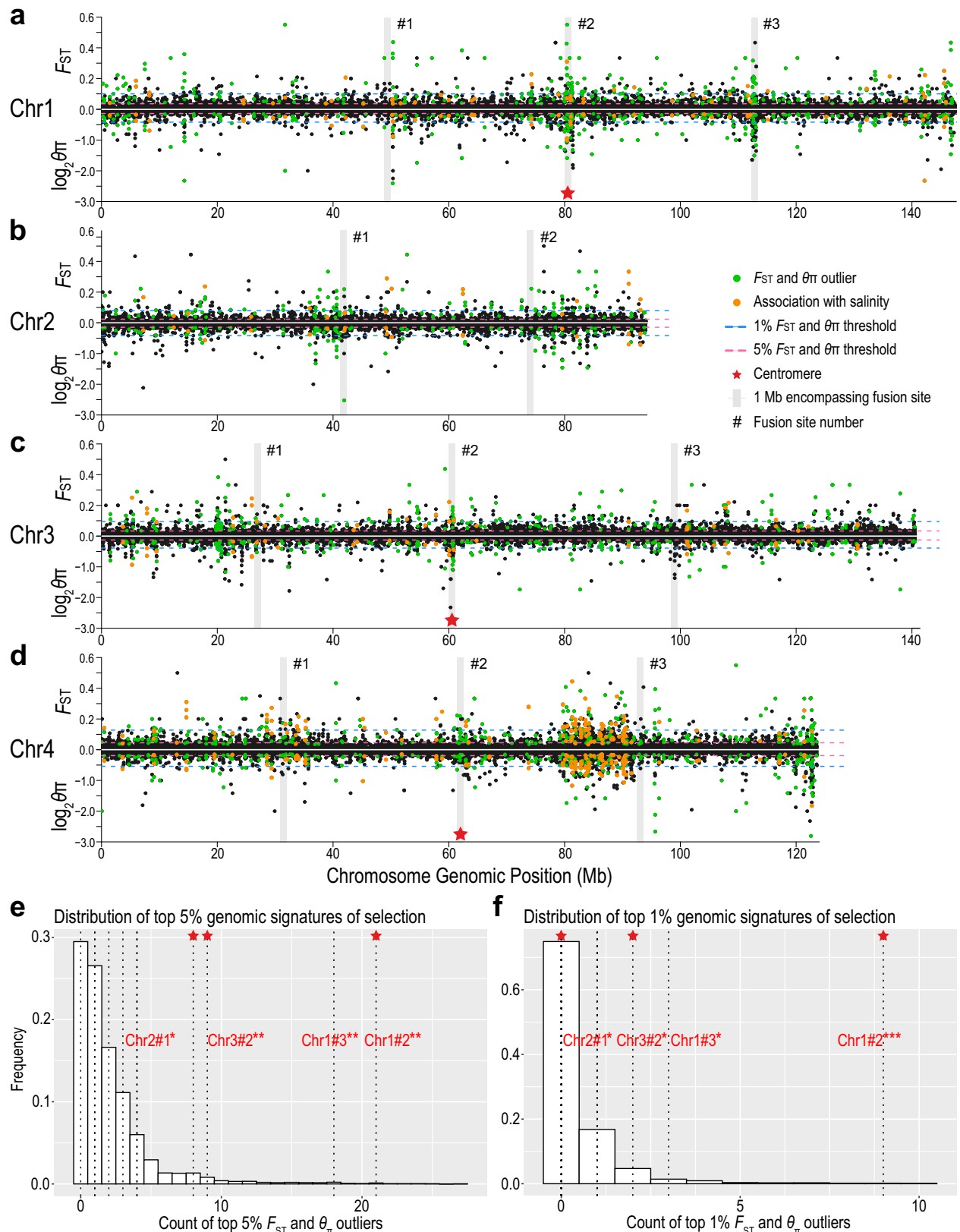

genomic signatures of selection associated with salinity change (i.e., SNP frequency shifts [$F_{ST}$ and $\theta_\pi$], association with salinity [BayPass], and LD) were enriched (or present) at several chromosomal fusion sites, especially at the centromeres, in genomes of the *E. affinis* complex sibling species with fused chromosomes. These genomes include the *E. carolleeae* genome, with 4 chromosomes (Fig. 7, Supplementary

Figs. 5–10), and the *E. gulfia* genome, with 7 chromosomes (Supplementary Figs. 11–17). The extent of enrichment of selection signatures at the fusion sites was much more pronounced for *E. carolleeae* than for *E. gulfia*.

To investigate whether the ancient chromosomal fusions and the formation of new gene linkages in the *E. carolleeae* and *E. gulfia*

**Fig. 7 | Signatures of selection associated with recent salinity change (< 80 years ago) mapped onto four chromosomes of the *E. carolleeae* genome.** **a–d** Distribution of selection signatures on each chromosome based on population genomic data from saline and freshwater populations[30]. Upper Manhattan plots (above 0) represent selection signatures detected by $F_{ST}$. Negative values are not displayed in this figure. Lower Manhattan plots (below 0) show selection signatures detected by nucleotide diversity, $\theta_\pi$ ratio ($\theta_{\pi\text{-invasive}}/\theta_{\pi\text{-native}}$). Green dots indicate 10 kb windows with selection signatures in the top 5% of $F_{ST}$ and the bottom 5% of $\log_2(\theta_{\pi\text{-invasive}}/\theta_{\pi\text{-native}})$. Orange dots represent the top 5% of $F_{ST}$ and $\theta_\pi$ outliers (10 kb windows) that were significantly associated with salinity by BayPass analyses. Blue and pink horizontal dashed lines denote thresholds for the top 1% and 5% values, respectively. Vertical grey stripes indicate 1 Mb regions that encompass fusion sites on each chromosome, with red stars indicating centromere positions. Histograms showing the count distribution of the top 5% (**e**) and top 1% (**f**) 10 kb windows with selection signatures generated by $10^6$ random sampling of 1 Mb intervals. Vertical dotted lines indicate the number of 10 kb windows under selection (based on $F_{ST}$ and $\theta_\pi$) within 1 Mb of fusion sites. Hashtags and numbers indicate the fusion site number on each chromosome shown in **a–d**. Centromeres are indicated by red stars. Asterisks denote fusion sites with significantly higher numbers of 10 kb windows with selection signatures compared to the background genomic distribution on the corresponding chromosomes. Significance levels were calculated using one-sided permutation tests and indicated as: ***$P < 0.001$, **$P < 0.01$, and * = $0.01 < P < 0.05$. Exact $P$-values: (**e**) Chr1#2 = 0.0016, Chr1#3 = 0.0039, Chr2#1 = 0.038, Chr3#2 = 0.0094; (**f**) Chr1#2 = $2.2 \times 10^{-16}$, Chr1#3 = 0.019, Chr2#1 = 0.031, Chr3#2 = 0.0090. Additional results for *E. carolleeae* are shown in Supplementary Figs. 5–10. Results for *E. gulfia* are shown in Supplementary Figs. 11–17. Source data are provided as a Source Data file.

genomes are potentially adaptive in contemporary populations, we mapped previous population genomic data from saline and freshwater populations[30] onto the *E. carolleeae* and *E. gulfia* genomes and reanalyzed population genomic signatures of selection (e.g., SNP allele frequency shifts). These population comparisons included three independent saline to freshwater invasions that occurred within the past ~80 years[24,30]. Specifically, we mapped onto the *E. carolleeae* genome population genomic sequences from two ancestral saline populations from the St. Lawrence estuary and two freshwater populations that invaded the Great Lakes 66 years ago. For the *E. gulfia* genome, we mapped population genomic sequences from two ancestral saline populations from the Gulf of Mexico and three populations that recently invaded freshwater reservoirs in the Mississippi drainage system within the past 80 years. We had previously found population genomic signatures of natural selection, in terms of allele (SNP) frequency shifts, between these saline and freshwater populations[30] using an older *E. carolleeae* reference genome assembly based on Illumina-seq[55].

We determined population genomic signatures of selection associated with salinity change (SNP frequency shifts) by identifying $F_{ST}$ and nucleotide diversity ($\theta_\pi$ ratio) outliers in our comparisons between saline and freshwater populations from *E. carolleeae* and *E. gulfia*[30]. We considered only the intersection between two estimates of selection, namely, the 5% highest $F_{ST}$ and the 5% lowest $\theta_{\pi\text{-invasive}}/\theta_{\pi\text{-native}}$ values, as indicative of signatures of selection (referred to as "top 5% $F_{ST}$ and $\theta_\pi$ outliers"; Fig. 7a–d, pink horizontal dotted lines). Additionally, we analyzed signatures of selection at the 1% threshold of high $F_{ST}$ and low $\theta_{\pi\text{-invasive}}/\theta_{\pi\text{-native}}$ values (referred to as "top 1% $F_{ST}$ and $\theta_\pi$ outliers"; Fig. 7a-d, blue horizontal dotted lines). As a result, we found significant signatures of selection for 1035 sliding 10 kb windows containing 645 genes in *E. carolleeae* and 1556 sliding 10 kb windows containing 1267 genes in *E. gulfia*. We found the same ion transporter genes under selection, such as *NHA*, *NKCC*, *Rh*, *NBC*, and *AMT*, as in our previous analysis[30]. As in the previous study, our GO enrichment analysis here revealed that ion transport functions were significantly overrepresented among the genes under selection in both the *E. carolleeae* and *E. gulfia* populations (Supplementary Data 12–15).

As an additional test for signatures of selection, we also employed the BayPass method to associate changes in allele frequency with salinity change[56], and integrated these results with the significant outliers based on $F_{ST}$ and $\theta_\pi$. Thus, we identified the top 5% of $F_{ST}$ and $\theta_\pi$ outliers (10 kb windows) that were also significantly associated with salinity based on BayPass, and we considered these as signatures of selection (referred to as "BayPass + 5% $F_{ST}$ and $\theta_\pi$ outliers"). As a result, we found 234 and 439 10 kb windows designated as BayPass + 5% $F_{ST}$ and $\theta_\pi$ outliers in the *E. carolleeae* and *E. gulfia* genomes, respectively.

Notable similarities and differences emerged when comparing genome-wide signatures of selection, based on multiple measures, between the *E. carolleeae* (Fig. 7a–d and *E. gulfia* genomes (Supplementary Fig. 11). While both *E. carolleeae* and *E. gulfia* showed similar percentages of fusion sites with selection signatures, 36.4% and 37.5% respectively, *E. carolleeae* displayed more consistent enrichment of selection signatures at the fusion sites across multiple statistical measures (Supplementary Data 16–19). In *E. carolleeae*, we found a striking enrichment of selection signatures at the fusion sites, relative to the background distribution, especially proximate to the centromeres (Fig. 7a–d, Supplementary Figs. 5–10, Supplementary Data 16 and 17). We had previously localized these centromeres using Hi-C interaction plots, based on distinctive patterns of reduced interaction frequencies at the centromeres due to their densely packed, heterochromatic nature (Supplementary Fig. 4)[46]. This distinctive pattern was not observed as clearly in the *E. gulfia* genome, where the centromeres appeared less distinct.

To determine the statistical significance of this enrichment of selection signatures at the fusion sites in both genomes, we sampled $10^6$ random 1 and 2 Mb intervals across each chromosome. We then counted the number of 10 kb windows with selection signatures (i.e., top 5% $F_{ST}$ and $\theta_\pi$ outliers, top 1% $F_{ST}$ and $\theta_\pi$ outliers, and BayPass + 5% $F_{ST}$ and $\theta_\pi$ outliers) within these intervals to generate the genomic background distributions of selection signatures for windows across each chromosome (see Methods). We compared the frequency distribution of windows under selection within intervals encompassing each fusion site against the background distribution of the corresponding chromosome. Significance levels for each fusion site, in terms of the number of windows with signatures of selection within 1 or 2 Mb intervals encompassing the fusion site, were determined by the probability of observing a higher count of windows under selection relative to the background distribution on each chromosome.

Assessment of selection signatures in the *E. carolleeae* genome revealed that four out of eleven chromosomal fusion sites (Chr1#2 [i.e., Chromosome 1, second fusion site], Chr1#3, Chr2#1, Chr3#2) exhibited significantly higher numbers of genomic windows with selection signatures (top 5% and 1% $F_{ST}$ and $\theta_\pi$ outliers, BayPass + 5% $F_{ST}$ and $\theta_\pi$ outliers) relative to the chromosome-wide background distributions (Fig. 7e, f, Supplementary Figs. 5–10, Supplementary Data 16 and 17). Furthermore, two centromeres (Chr1#2 and Chr3#2, Fig. 7e) consistently ranked within the upper 5% quantile for the number of windows with selection signatures. When evaluating all fusion sites collectively, significant enrichment of selection signatures at fusion sites was confirmed for both 1 Mb (top 5%: Kolmogorov-Smirnov [KS] test, $P = 0.0142$; top 1%: KS test, $P = 0.0122$; BayPass + 5% $F_{ST}$ and $\theta_\pi$: KS test, $P = 3.79e-7$; Supplementary Data 16) and 2 Mb intervals (top 5%: KS test, $P = 0.0233$; top 1%: KS test, $P = 4.17e-4$; BayPass + 5% $F_{ST}$ and $\theta_\pi$: KS test, $P = 3.83e-4$; Supplementary Data 17).

Similarly, chromosomal fusion sites of the *E. gulfia* genome also showed significant enrichment of selection signatures at three out of eight fusion sites (Supplementary Figs. 12–17), though the signals were less pronounced than for the *E. carolleeae* genome. Specifically, when we examined all fusion sites relative to the genomic background for *E. gulfia*, we found significant enrichment of selection signatures at

fusion sites based on BayPass + 5% $F_{ST}$ and $\theta_\pi$ outliers within 1 Mb intervals (KS test, $P = 9.22e{-}4$). Using 2 Mb intervals, significant enrichment was evident based on multiple measures, including top 5% $F_{ST}$ and $\theta_\pi$ outliers (KS test, $P = 0.0401$), top 1% $F_{ST}$ and $\theta_\pi$ outliers (KS test, $P = 0.0280$), and BayPass + 5% $F_{ST}$ and $\theta_\pi$ outliers (KS test, $P = 0.0226$) (Supplementary Data 18 and 19). Three out of eight fusion sites, one each on Chromosomes 1, 5, and 7, exhibited significantly higher numbers of genomic windows with signatures of selection, based on BayPass + 5% $F_{ST}$ and $\theta_\pi$ outliers, relative to the genomic background (Supplementary Fig. 17). One of these fusion sites, on Chromosome 5, exhibited more robust enrichment of selection signatures, as this fusion site was also enriched for genomic windows that were top 1% $F_{ST}$ and $\theta_\pi$ outliers (Supplementary Figs. 14, 16, and 17).

In terms of patterns of recombination across the genome, the *E. carolleeae* genome showed evidence of reduced recombination at the fusion sites compared to the genome-wide background (Supplementary Fig. 18). We investigated patterns of linkage disequilibrium (LD) across the genome, using whole-genome sequences of 14 individual copepods, to evaluate whether fusion sites exhibit signatures of reduced recombination relative to genome-wide levels. This analysis was motivated by the hypothesis that chromosomal fusions are clustering co-adapted loci and alleles into regions of suppressed recombination. This clustering of co-adapted alleles into low recombination genomic regions would then facilitate their joint inheritance, with the potential to promote adaptive evolution. Notably, the fusion sites of the *E. carolleeae* genome did indeed exhibit significantly elevated LD, relative to the genomic background, in both 1 Mb (Paired *t*-test; $t = 264.4$, $df = 299{,}980$, $P < 0.0001$) and 2 Mb genomic intervals (Paired *t*-test; $t = 275.1$, $df = 299{,}980$, $P < 0.0001$) (Supplementary Fig. 18). Fusion sites with significant enrichment of selection signatures demonstrated even higher LD relative to all fusion sites in both 1 Mb (Paired *t*-test; $t = 206.9$, $df = 299{,}980$, $P < 0.0001$) and 2 Mb genomic intervals (Paired *t*-test; $t = 240.0$, $df = 299{,}980$, $P < 0.0001$) (Supplementary Fig. 18). Recombination patterns were not explicitly analyzed for *E. gulfia* in this study due to the lack of genome sequences for individual copepods.

Overall, our results reveal contemporary genomic signatures of selection at the chromosomal fusion sites in both sibling species (*E. carolleeae* and *E. gulfia*) of the *E. affinis* complex. While *E. carolleeae* displayed more prominent and statistically robust enrichment of selection signatures, particularly around clearly defined centromeres (Fig. 7a–d), *E. gulfia* also showed significant enrichment of selection signatures at several fusion sites (Supplementary Fig. 11).

## Discussion

Chromosomal fusions have long been hypothesized to facilitate adaptation, but empirical support had been scarce and indirect[5,6,13,17]. Theoretical studies had predicted that genomic rearrangements such as chromosomal fusions or inversions could enhance adaptive potential by repositioning functionally interacting genes into closer proximity and altering the recombination landscape[6,10]. While several empirical studies have found that chromosomal inversions could reposition adaptive loci and reduce recombination among beneficial alleles, thereby altering how populations respond to selection[12,14,15,57,58], none had actually linked chromosomal fusions with population responses to selection.

A unique feature of our study is that we identified elevated signatures of natural selection at the chromosomal fusion sites, particularly around the centromeres. We found that ancient chromosomal fusions repositioned key ion transporter genes into central chromosomal regions of reduced recombination in certain clades (sibling species) of the *E. affinis* complex. While these chromosomal fusions were ancient, these fusion sites exhibited population genomic signatures of selection (see next paragraph), suggesting that selection in contemporary populations could act upon the resulting pre-existing combinations of adaptive alleles. As such, genome architecture, particularly chromosomal fusions, could have profound impacts on population responses to natural selection.

We found that the fusion sites were enriched with signatures of selection associated with contemporary shifts in habitat salinity (< 80 years ago)[30]. We used multiple measures to demonstrate signatures of selection at these fusion sites, including genomic divergence outlier analysis (increased $F_{ST}$), reduced nucleotide diversity (reduced $\theta_\pi$), association with salinity (BayPass), and linkage disequilibrium (elevated LD). Intriguingly, we observed the significant enrichment of selection signatures at the fusion sites in two distinct sibling species, *E. carolleeae* and *E. gulfia*. As such, these results revealed the convergent evolution of chromosomal reorganization that brought together adaptive loci in different sibling species. The population genomic signatures of selection associated with the chromosomal fusions suggest that these ancient genomic reorganizations conferred a selective advantage that benefits contemporary populations.

The signatures of selection at the chromosomal fusion sites (in terms of $F_{ST}$, $\theta_\pi$, association with salinity, and LD) were particularly pronounced in the *E. carolleeae* (Atlantic clade) genome. *E. carolleeae* is the most invasive within the *E. affinis* complex and has a history of repeated saline to freshwater invasions[24,44], as well as invasions among saline habitats[26–28], surpassing other clades (sibling species) in salinity range and geographic breadth[24,26–28] (Fig. 1c, red triangles). In the *E. carolleeae* genome, we found salinity-associated selection signatures at 36% of the fusion sites, most strikingly at the centromeres newly created by the chromosomal fusions (Fig. 7a–d). For instance, the centromere region of Chromosome 3, which includes seven paralogs of the *NHA* gene family, showed enriched selection signatures in this study (Fig. 7) and contained the highest density of SNPs under selection between replicate sets of wild saline and freshwater populations in a previous study[30]. The presence of signatures of elevated LD around the chromosomal fusion sites in *E. carolleeae* (Supplementary Fig. 18) supported that recombination was low enough to avoid disassociating selected alleles at multiple loci at these fusion sites. The much more prominent contemporary signatures of selection at the fusion sites for *E. carolleeae*, relative to *E. gulfia*, might be associated with a greater aggregation of functionally related beneficial alleles at the fusion sites, potentially due to strong selection pressure that brought these loci together in the ancient past. In particular, the enrichment of ion transport-related genes under selection near the fusion sites of *E. carolleeae* suggests that the ancient selection pressure was related to salinity change and/or fluctuating salinity (see below).

Moreover, we demonstrated that these chromosomal fusions had repositioned crucial genes with signatures of selection toward more central regions of the chromosomes in *E. carolleeae*, particularly at the centromeres, where recombination is low. Notably, the fusion events relocated ion transport-related genes, which are targets of selection during salinity change[29,29] from chromosome edges of the ancestral karyotype (15 chromosomes) toward more central regions in the derived *E. carolleeae* genome (4 chromosomes; examples shown in Fig. 6b). Consequently, many of these ion transport-related genes that were originally located on separate chromosomes in *E. affinis* proper (15 chromosomes) were joined by fusions onto the same chromosomes in *E. carolleeae*. Indeed, as a result of the fusions, we observed that 25 crucial ion transport-related genes with signatures of selection[30,30] had moved significantly toward chromosome centers (away from telomeres) compared to their positions in the unfused *E. affinis* proper genome.

These ion transporters and carbonic anhydrase are considered to be functionally linked, as they must cooperate to perform ion uptake from the environment (Fig. 5b, c)[25]. By bringing together formerly unlinked loci into close linkage, the fusions have reorganized the genome in ways that could facilitate the joint inheritance of functionally-related coadapted genes and alleles. Moreover, we found

genomic signatures consistent with synergistic epistasis among specific alleles of ion transport-related genes during a laboratory evolution experiment, suggesting that these alleles are under selection as a unit[30]. In addition, such a genomic reorganization toward the clustering of ion transport-related genes could greatly enhance physiological responses to salinity change by promoting co-regulation of these genes[6,8,13]. Our previous results had found the coordinated regulation of gene expression of these ion transporters and carbonic anhydrase[31], and their colocalization within ion transporting cells[59,60].

Notably, the multiple paralogs within ion transport-related gene families appear to collectively contribute to salinity adaptation and undergo natural selection (allele frequency shifts) during saline to freshwater invasions. The different ion transport-related paralogs appear to show functional differentiation, as they are expressed in different osmoregulatory tissues (e.g., maxillary glands, swimming legs, or digestive tract) and show differences in gene expression patterns[31,59–61]. In addition, the multiple ion transport-related paralogs appear to be nonredundant and essential for salinity adaptation, given that particular sets of paralogs are repeatedly under selection in wild populations and laboratory selection lines[29,30]. Therefore, it does appear that the ion transport-related gene family expansions have played an important role in promoting the invasiveness of certain clades of the *E. affinis* complex.

The large clusters of ion transport-related genes brought together in the *E. carolleeae* genome might constitute "supergenes" underlying salinity tolerance. Supergenes are distinct from "genomic islands of divergence," in forming tightly linked sets of co-functional loci maintained by suppressed recombination. The clusters of ion transport-related genes found in this study constitute not only islands of high differentiation, but also potential coadapted gene complexes that could be inherited and evolve together as a unit. Furthermore, our evidence of high LD and functional relatedness among the fused loci in *E. carolleeae* are consistent with these fusions forming supergenes. The findings of this study suggest that chromosomal fusions might serve as an important mechanism for generating supergenes. Such a process of supergene formation through chromosomal fusions has been hypothesized in theory[5,10] but rarely documented empirically in the past.

Our results complement previous work indicating that chromosomal inversions serve as a key mechanism for supergene formation[12,14,57,58]. Previous examples of supergenes, such as those governing mimicry in butterflies or social behavior in ants and birds[7,8,11,12], are typically facilitated by inversions that lock together multiple loci that are already on the same chromosome. In contrast, our study reveals that chromosomal fusions can form novel linkages among adaptive genes by joining chromosomes that were previously unlinked. This mechanism can profoundly alter the recombination landscape of the genome, generating new low-recombination regions that harbor coadapted allelic combinations.

Our study also emphasizes how contrasting genome architectures can lead to different evolutionary strategies for adaptation. Despite the stark differences in chromosomal architecture, the parallel evolution we had previously observed across the *E. affinis* complex clades (sibling species) is intriguing, with selection favoring the same loci (and often the same SNPs) in response to salinity change in all three sibling species examined in this study (i.e., *E. carolleeae*, *E. gulfia*, *E. affinis* proper)[29,30,34]. In the more ancestral genome architecture of *E. affinis* proper, many of the adaptive genes that respond to salinity change are unlinked and localized at the edges of its 15 chromosomes (see next paragraph). In sharp contrast, the chromosomal fusions in *E. carolleeae* have merged the loci under selection in response to salinity change in the more central regions of the chromosomes, especially at the centromeres, at positions of reduced recombination. Thus, selection is likely to act on sets of beneficial alleles that are linked together and unlikely to be separated through recombination (see below).

Our prior experimental results indicate how genome architecture could have profound effects on population responses to natural selection. In our prior laboratory experiment, we imposed selection for reduced salinity on replicate selection lines from an *E. affinis* proper population (15 chromosomes). We found remarkably high parallelism among the 10 selection lines, with selection favoring many of the same alleles among the lines, especially ion transporter alleles[29]. Extensive simulations of our experimental conditions revealed that this parallelism was consistent with positive synergistic epistasis among alleles, far more than other mechanisms tested, such as physical linkage[29]. The extent of parallelism among the selection lines increased with more generations of selection, favoring the same freshwater-adapted alleles. This result indicated that recombination events, which would be more frequent with greater numbers of chromosomes, brought together freshwater-adapted alleles within genomes, likely facilitated by positive synergistic epistasis among the beneficial alleles[29].

In sharp contrast, for *E. carolleeae*, with only four chromosomes, positive synergistic epistasis (enabled by recombination) would not serve as a plausible mechanism to induce parallel evolution. As the ion transport-related genes are already linked in regions of low recombination, recombination would not be able to shuffle alleles among haplotypes and bring the beneficial alleles together via positive epistasis. Rather, the sets of beneficial alleles would need to be already physically present in the genome. Thus, adaptation to salinity change in *E. carolleeae* would require selection favoring either freshwater-adapted genomes, with linked freshwater-adapted alleles, or saltwater-adapted genomes, with linked sets of saltwater-adapted alleles. Standing variation of these differentially adapted genomes might persist in many native saline *E. carolleeae* populations due to seasonally fluctuating salinity, resulting in balancing selection acting on salinity tolerance[62,63]. In many native range habitats of *E. carolleeae*, salinity fluctuates seasonally between 5 and 35 PSU, such that different salinity tolerance haplotypes would be favored by selection across seasons[30]. In addition, beneficial reversal of dominance of adaptive alleles during salinity shifts would contribute to the maintenance of both saltwater-adapted and freshwater-adapted supergenes in a population[63]. Under beneficial reversal of dominance, dominance switches between environments, such that alleles that contribute to freshwater tolerance are dominant under freshwater conditions, whereas alleles that contribute to saltwater tolerance are dominant under saltwater conditions. This beneficial reversal of dominance, which has been demonstrated in *E. affinis* complex populations[63], potentially arises from the more fit allele in a heterozygote compensating for the lower function of the less fit allele[64,65]. Thus, even though selection is acting on the same ion transporter genes in different sibling species in response to salinity change, the mode of selection could vary considerably according to genome architecture, including differences in chromosome number and positions of beneficial alleles.

The timing of divergence among the three sibling species within the *E. affinis* complex is surprisingly old and might coincide with catastrophic geological events (Fig. 3). While estimates of divergence times among the sibling species are highly uncertain, the separation among these species might coincide with the Cretaceous Mass Extinction (K–Pg boundary) around 65 Mya and the subsequent India–Asia tectonic collision around 50 Mya. These tectonic events led to extensive oceanic changes, such as acidification and regional sea level changes, which might have promoted speciation events within the *E. affinis* complex[66,67]. More recently, glaciation cycles and repeated sea level change during the Pleistocene Epoch of ~18,000 years ago, with the creation of brackish to saline tide pools along the coasts, likely contributed to the speciation process[68]. These recent events might be reflected in the history of population contractions in our reconstruction of demographic histories (Supplementary Fig. 19). Both *E. carolleeae* and *E. gulfia* populations experienced reductions in

effective population size around this time period (Supplementary Fig. 19).

While the exact timing of the chromosomal fusions among the three sibling species of the *E. affinis* complex is uncertain, the relatively uniform GC content across the chromosomes (Supplementary Data 20) suggests that most of the fusion events are quite ancient[69,70]. At this point we do not know how the timing of chromosomal fusions corresponds with the timing of divergence among the clades. While the ancient chromosomal fusions and the recent freshwater invasions are not temporally linked, the evolutionary history of copepods of the genus *Eurytemora* in subarctic coastal habitats with fluctuating salinity[68] might have contributed to genome architecture evolution in this genus. The ancient evolutionary processes that led to the current genome architecture of *E. carolleeae*, and other invasive sibling species within *E. affinis* complex, might be contributing to its remarkable ability to respond to contemporary changes in habitat salinity.

These chromosomal fusions likely played a crucial role in speciation processes within this species complex[47,52]. Chromosomal fusions are increasingly recognized as important drivers of speciation, especially in systems where closely related species or populations exhibit divergent karyotypes[4,17,48,71,72,73]. Chromosomal changes can promote speciation events by impeding gene flow between populations, thereby facilitating the accumulation of genetic differences among them[4,17,48,71,73]. In our prior studies, intermating between the genetically distinct sibling species of the *E. affinis* complex often yielded F1 hybrids and occasional F2 hybrids between the crosses[22]. While postzygotic isolation between the clades is present[22], the capacity of crosses between the species to produce viable hybrids is surprising. Key questions remain regarding how these sibling species can produce hybrid offspring despite differences in chromosome number. Thus, it would be worth exploring the relationship between chromosomal evolution and speciation in the *E. affinis* complex, to assess the extent to which chromosomal evolution drives speciation events.

The distinct genome architectures in closely related sibling species of the *E. affinis* complex provide a valuable framework for exploring the interplay between chromosomal evolution, particularly chromosomal fusions, and natural selection in response to environment change. Our study achieved strong inferences by explicitly tracking the relocation of specific adaptive loci following chromosomal fusion events. Overall, our study suggests that the potential importance of chromosomal fusions in facilitating adaptation have been undervalued compared to other genomic rearrangements, such as inversions. As such, our findings contribute to the broader discourse on genome architecture evolution and adaptation by providing direct support for the role of chromosomal evolution in aggregating adaptive loci and influencing evolutionary trajectories. Importantly, our study emphasizes that differing genome architectures among closely related species could lead to alternative evolutionary strategies for rapid adaptation to environmental change. A deeper focus on the interaction between natural selection and genome architecture evolution could enhance our understanding on the adaptability of populations in the face of rapidly changing environments.

## Methods

### Mitochondrial phylogeny and morphological analyses
The phylogenetic relationships within the *E. affinis* complex were reconstructed using concatenated mitochondrial *COI* and *16S rRNA* gene sequences from 29 globally distributed population samples (taken from Lee, 2000[22]), with both maximum-likelihood (ML) and Bayesian inference (BI) approaches. *Eurytemora americana* and *E. herdmani* were used as outgroup species. Gene alignment was performed with TranslatorX web server[74]. For the ML analysis, IQ-TREE v1.6.5[75] was employed conducting 1000 bootstrap replicates to determine node support. The *COI* and *16S rRNA* genes were treated as separate partitions, and the best-fit substitution models (HKY + I + G)

were selected using IQ-TREE's auto model detection. In the BI analysis, MrBayes v3.2.7[76] was used for two Markov chain Monte Carlo (MCMC) runs of two million generations each to obtain posterior probabilities for the nodes. Tree sampling was performed every 1000 generations, with the initial 25% discarded as burn-in.

The morphological data (taken from Lee and Frost, 2002[21]) using 70 male and 71 female *E. affinis* complex individuals from all the known clades were reanalyzed. Female measurements included genital segment width/prosome length (GSW/PL), genital segment width/genital segment length (GSW/GSL), and ratios of the 22$^{nd}$ and 24$^{th}$ segments of the first antennule (A1 22:24) (Supplementary Data 21). Male measurements consisted of right exopod 1 length/prosome length (rtP5 exo1/PL), right exopod 2 length/prosome length (rtP5 exo2/PL), left basipod 2 width/prosome length (lftP5 Bp2W/PL), right exopod 1 length/left basipod 2 width (rtP5 exo1/lft P5 Bp2W), and ratios of the 22$^{nd}$ and 24$^{th}$ segments of the first left antennule (A1 22:24) (Supplementary Data 22). Principal component analysis (PCA) was performed in R to explore the morphological variation among different clades of the *E. affinis* complex.

### Population sample collection and laboratory inbreeding
For inbreeding, followed by comprehensive genome sequencing (next section), wild population samples were collected for the sibling species *E. gulfia* (Gulf clade) and *E. affinis* proper (Europe clade) of the *E. affinis* species complex (Fig. 1). An *E. gulfia* population sample was collected from Blue Hammock Bayou, Louisiana, USA in 2012, whereas an *E. affinis* proper population sample was collected from Stockholm, Sweden in 2019 (Fig. 1c). To reduce heterozygosity in these wild populations, inbred lines were established through full-sibling mating in the Lee laboratory at the University of Wisconsin–Madison. This process involved 20 generations of full-sibling mating for *E. gulfia* (starting on March 29, 2012) and 10 generations for *E. affinis* proper (starting on March 19, 2019). The inbred lines were maintained at 12 °C on a 15D:9D light cycle. The *E. gulfia* lines were maintained at 5 PSU and *E. affinis* proper lines were maintained at 15 PSU. The saline water was prepared using sea salt (Instant Ocean, Blacksburg, VA, USA), with Primaxin (20 mg/L) to prevent bacterial infections. The lines were fed thrice weekly with the marine alga *Rhodomonas salina*. This study specifically used the inbred lines Gulf-Square-1 and Stockholm-B-1 for genome sequencing.

### Genome sequencing
For genome sequencing, approximately 2000 adult copepods each from the *E. gulfia* and *E. affinis* proper inbred lines were collected. The copepods were randomly selected with an approximate sex ratio of 1:1 from our beakers of inbred lines to ensure a representative mix. To minimize contamination from gut contents and the microbiome during DNA extraction, the copepods underwent a rigorous antibiotic treatment. Two weeks before DNA extraction, the inbred lines were treated with antibiotics (20 mg/L Primaxin, 0.5 mg/L Voriconazole) and D-amino acids (10 mM D-methionine, D-tryptophan, D-leucine, and 5 mM D-tyrosine), with bi-weekly water changes. In the final three days, the copepods received five additional antibiotics (20 mg/L Rifaximin, 40 mg/L Sitafloxacin, 20 mg/L Fosfomycin, 15 mg/L Metronidazole, 3 mg/L Daptomycin), with daily water changes. For the last 48 h, the copepods were starved and then fed 90 µL/L of 0.6-micron copolymer beads (No. 7505 A, Sigma-Aldrich, St. Louis, MO, USA) to remove the gut microbiome.

To generate the DNA sequence data, a CTAB-based phenol/chloroform/isoamylol method was employed for DNA extraction[46], yielding 15 µg of high molecular weight genomic DNA per sample. DNA quality was assessed using pulsed-field gel electrophoresis and Qubit 3.0 fluorometry (Thermo Fisher). A Pacific Biosciences (PacBio, Menlo Park, CA, USA) CLR library with 10–20 kb insert sizes was constructed using the SMRTbell Template Prep Kit 1.0 (PacBio) according to the

manufacturer's protocol. Sequencing was performed on a PacBio Sequel SMRT Cell 8 M using the Sequel II platform at Novogene (Sacramento, CA, USA), generating 6.7 million reads (180.3 Gb, ~360× coverage) for *E. gulfia* and 4.2 million reads (161.5 Gb, ~240× coverage) for *E. affinis* proper. Additionally, 0.5 µg of DNA from each species was used to construct a 350 bp insert size library, sequenced on the Illumina NovaSeq 6000 platform (San Diego, CA, USA) at Novogene with the 150 bp paired-end (PE) mode, yielding 270.6 million reads (40.6 Gb, ~80× coverage) for *E. gulfia* and 441.8 million reads (66.3 Gb, ~100× coverage) for *E. affinis* proper.

To generate transcriptome data, total RNA was extracted from 100 adult copepods per species using the TRIzol reagent (No. 15596026, Invitrogen, Waltham, MA, USA), following the manufacturer's protocol. The quality of the isolated RNA was rigorously assessed using a Nanodrop Spectrophotometer, gel electrophoresis, and an Agilent 2100 Bioanalyzer to ensure integrity and purity. Subsequently, messenger RNA (mRNA) was enriched by oligo(dT) magnetic bead capture, with other RNA types being excluded. This procedure was followed by mRNA fragmentation and cDNA reverse transcription prior to RNA library preparation. The resulting cDNA samples were processed into libraries for sequencing. These libraries were then sequenced on the NovaSeq 6000 platform, employing a 150 bp Paired-End (PE) sequencing mode. This process resulted in high yields of RNA sequence data, with 242.7 million reads (36.3 Gb) for *E. gulfia* and 245.6 million reads (36.7 Gb) for *E. affinis* proper.

To obtain chromatin interaction data, two Hi-C sequencing libraries were prepared following established protocols[77] at Novogene. Chromatin from 500 copepods per species was cross-linked with 2% formaldehyde, followed by DNA extraction. After MboI restriction endonuclease digestion, non-ligated DNA fragments were removed. The ligated DNA was sheared to 350 bp and processed through a standard Illumina library preparation protocol. Hi-C libraries were also sequenced on the Illumina NovaSeq 6000 platform with the 150 bp PE mode, resulting in 319.3 million reads (47.9 Gb, ~95× coverage) for *E. gulfia* and 504.1 million reads (75.6 Gb, ~110× coverage) for *E. affinis* proper.

## Genome assembly

For the genome assembly of *E. gulfia* and *E. affinis* proper, genome size was initially estimated using Illumina sequence data. Raw sequence reads were processed with Fastp v0.23.0[78] for quality trimming. JELLYFISH v2.3.1[79] was then employed for k-mer distribution analysis (count -m 21 -C -s 1 G -F 2, histo -h 1,000,000) to estimate the genome size. GenomeScope v2.0[80] was subsequently utilized to determine the genome size, heterozygosity, and repetitive sequence proportion, using a k-mer size of 21.

The PacBio CLR sequencing reads were self-corrected using NextDenovo v2.3.1[81]. This step included all-to-all alignments by minimap2 and Nextgraph within NextDenovo to construct the primary genome assembly. NextPolish v1.4.1[82] was then applied for assembly polishing, integrating both PacBio CLR and Illumina short reads. This process consisted of one round of long-read polishing followed by three rounds of short-read polishing to enhance assembly quality. The assembly's contiguity was evaluated using the N50 statistic, while its completeness was assessed using BUSCO v5.2.2, targeting 1,013 genes from the arthropod odb10 database[83]. Purge_dups v1.2.6[84] was employed to remove heterozygous duplicates of the genome assembly.

For chromosome-level scaffolding, Juicer v1.6[85] and 3D-DNA v180922[86] were utilized. Visualization of Hi-C plots was conducted in Juicebox v1.91[87]. Additionally, microbial scaffolds that were disconnected from the main assembly were identified and removed, namely, nine from *E. gulfia* and thirty-two from *E. affinis* proper, using BLAST v2.8.1[88] against the microbial subset of the Nucleotide database.

## Karyotyping procedure

Karyotyping was performed to confirm the chromosome counts for *E. gulfia* and *E. affinis* proper. To prepare chromosomes for karyotyping, tissues were dissected from embryos within 24 h post-ovulation. These embryos were separated from females and incubated for 1 h in 15 PSU solution with 0.05% Colchicine. Following incubation, embryos were treated in 0.075 M KCl hypotonic solution for 20 min before fixation with fresh Carnoy's fixative (3 parts methanol to 1 part acetic acid) for 20 min. The fixed embryos were subsequently placed on microscope slides and gently minced using a tungsten needle. The slide preparation followed the protocol described in Kao et al.[89]. Chromosomes were stained with DAPI using SlowFade Antifade Mountant (No. S36967, Invitrogen) for 20 min. Imaging was performed using an Olympus BX60 epifluorescence microscope equipped with a Uplan-SApo 60×/1.35 NA objective and an Olympus DP72 camera. The imaging was conducted at the Newcomb Imaging Center (Birge Hall), Department of Botany, University of Wisconsin–Madison.

## Genome annotation

For annotating the genomes of *E. gulfia* and *E. affinis* proper, a comprehensive approach was employed to identify repetitive sequences, transposable elements, and protein-coding regions. RepeatMasker v4.07[90] was used to scan for repetitive sequences and transposable elements. Several databases were referenced for this purpose, including Repbase v202101[91], Dfam v3.7[92], and a de novo repeat library created by RepeatModeler v1.0.8[93]. Tools such as RECON v1.08[94], TRF v4.09[95], and RepeatScout v1.06[96] were integrated into this process. Additionally, long terminal repeat (LTR) elements were specifically searched using LtrHarvest[97], CD-HIT[98], and Ltr_retriever[99]. Unknown transposable elements were reclassified by DeepTE[100]. For annotating protein-coding regions, MAKER v3.01 pipeline[101] was utilized. This process involved integrating homology-based, transcriptome-based, and ab initio prediction strategies. Homology evidence was provided by protein sequences from pancrustacean species, including *Drosophila melanogaster*, *Daphnia pulex*, *Tigriopus californicus*, *Lepeophtheirus salmonis*, and *E. carolleeae*, sourced from the NCBI RefSeq database. Transcriptomic evidence was drawn from 53 datasets, including those from our previous *E. affinis* complex genome sequencing and gene expression studies[31,46,55,102] and newly sequenced data for *E. gulfia* and *E. affinis* proper from this study (see previous section). These datasets were reassembled using HISAT v2.0.4[103] and StringTie v2.2.1[104]. For ab initio prediction, SNAP[105] and GeneMark-ES[106] were employed, with iterative runs of MAKER refining the gene models.

Functional annotation involved BLASTP searches against databases, including invertebrate subsets of the NCBI RefSeq, UniProtKB/Swiss-Prot[107] databases, and NCBI RefSeq annotation on *E. carolleeae* (GCF_000591075.1-RS_2023_10). EggNOG-mapper v2.1.9[108] was also utilized for searches in GO[109], KEGG[110], COG, and eggNOG[111] databases. The Pfam database in InterPro[112] was analyzed with HMMER v3.2[113] to gain additional functional insights.

## Gene family analyses

Patterns of gene family expansions and contractions were examined for the three sibling species of *E. affinis* species complex and four additional crustacean (copepod and daphnid) species, by reconstructing a phylogeny and analyzing their gene content. OrthoFinder v2.5.4[114] was employed to identify unique and shared orthologous gene families within the species complex. Protein sequences from the four additional copepod and daphnid species (*Daphnia pulex*, *Daphnia magna*, *Tigriopus californicus*, *Lepeophtheirus salmonis*), which had high-quality, chromosome-level genome assemblies from GenBank, were included (Supplementary Data 6). These genome assemblies of other copepod and daphnid species were annotated using a comparable approach to our study (see Section on Genome annotation

above), which incorporates transcriptome data, ab initio predictions, and homology to known proteins. This methodological consistency ensured that the gene counts across all species were comparable for our comparative analyses. After filtering to retain only the longest transcript for each gene, BLASTP alignments (e-value < 1e−5) were performed for protein sequences from the *E. affinis* complex and the selected species. Single-copy gene sequences were aligned using MAFFT v7.313 with the L-INS-I algorithm[115]. A maximum-likelihood phylogeny was constructed with RAxML v8.0.19[116]. Statistical support for tree topology was obtained through 100 bootstrap replicates. Divergence times were estimated using MCMCTree from PAML v4.9[117], with calibration of the MRCA age for copepod (183–365 Mya) from the TIMETREE v5 database[51]. CAFÉ5[118] was used to analyze patterns of gene family expansions and contractions within the phylogeny. Expanded and unique gene families in the *E. affinis* complex were subjected to GO enrichment analyses using TBtools v1.112[119].

## Ancestral karyotype reconstruction and synteny analyses

Ancestral karyotypes were reconstructed for the common ancestor of *E. carolleeae* and *E. gulfia*, as well as for the entire *E. affinis* species complex. Initially, AGORA v3.1[52] was employed to reconstruct the Continuous Ancestral Regions (CARs) using shared orthologs at each ancestral node, applying the command "agora-generic.py species-tree.nwk orthologyGroups/*orthologyGroups.list genes/*genes.list". Due to the highly fragmented output from AGORA, ANGES v1.01[53] was subsequently used to refine the ancestral karyotype reconstruction, anchoring the orthologs to the CARs determined by AGORA.

Syntenic relationships among the three sibling species of the *E. affinis* complex were analyzed with MCScan in JCVI[120]. This analysis involved identifying co-linear gene blocks within the genome, using the longest coding sequence for each gene. Genome-wide alignment and similarity analysis among the genomes of the three *E. affinis* complex sibling species was conducted using MUMmer v4.0[121]. In addition, key ion transport-related genes in *E. carolleeae* that showed evolutionary shifts in gene expression and/or signatures of selection were manually annotated in the genomes of *E. gulfia* and *E. affinis* proper and mapped onto their syntenic plots.

## Gene repositioning analysis within chromosomes

To determine whether chromosomal fusions resulted in significant repositioning of key ion transport-related genes in the *E. carolleeae* and *E. gulfia* chromosomes, relative to *E. affinis* proper chromosomes, the mean and median positions of the 25 key ion transport-related genes were compared between the different sibling species. These genes had been identified as targets of selection in response to salinity transitions in our previous studies[29,30,34]. Positions of these genes were calculated by measuring their distances from the nearest edges of each chromosome divided by the chromosome length, with values ranging from 0% (edges of chromosome) to 50% (center of chromosome).

Within the *E. carolleeae* genome, statistical analyses were conducted to compare the chromosomal repositioning of 25 key ion transport-related genes under selection against three gene sets: (1) 36 gene paralogs lacking selection signatures but belonging to the same gene families as the 25 key ion transport-related genes under selection, (2) 253 genes annotated with putative ion transporting functions[46], excluding the 25 key ion transport-related genes under selection, and (3) subsets of 25 genes randomly selected from the 253 genes with putative ion transporting functions.

For the first two gene sets, normality of data distributions was evaluated using the Shapiro-Wilk tests before applying Welch's *t*-tests to compare mean positions between the sibling species. For median positions, Mann–Whitney *U* tests were employed to assess differences between the sibling species. A permutation testing approach was employed to generate the randomly selected sets of 25 genes to compare against the 25 selected genes. This approach involved

repeatedly executing both Welch's *t* test and Mann–Whitney *U* test $10^6$ times with different random samplings of 25 genes for each replicate run. The randomization sampling processes in our study were all conducted using the 'sample' function in R, which employs the Mersenne Twister algorithm to generate random numbers. This approach ensures an impartial and unbiased sampling process. All statistical tests were conducted using the software package R.

## Linking selection signatures in wild populations with chromosomal positioning

To explore the association between chromosomal fusions, and the formation of novel gene linkages, with signatures of selection in *E. affinis* species complex populations, Illumina-seq population genomic data collected previously from wild populations of *E. carolleeae* and *E. gulfia*[30] were reexamined. These data had sequence coverage ranging from 16× to 30×, with a mean coverage of 25× (Supplementary Data 23). These samples included four wild populations from *E. carolleeae* and five from *E. gulfia*, representing three independent invasions of freshwater environments (0–0.9 PSU) from native saline habitats (4–40 PSU) within the past ~80 years. From each population, 100 adult individuals were sampled for pooled sequencing, maintaining an approximate 1:1 sex ratio. DNA libraries, sequenced to yield an average of 179 million 100-bp read pairs per population, were subjected to quality control using Fastp. This step involved filtering out low-quality reads and adapters. Clean reads were then mapped onto the reference genomes of *E. carolleeae* and *E. gulfia* using BWA-MEM v0.7.12-r1039[122]. Duplicate reads were removed using Picard v2.21.6. The resulting genome mappings were sorted, converted to BAM format, and further transformed into Pileup format with SAMtools v1.21.1[123], discarding low-quality alignments and bases (Q < 20). SNPs were called using VarScan v2.4.6[124], and raw SNPs were filtered using BCFtools v1.21[125]. The resulting VCF files were processed using the R package poolfstat v1.1.1[126], retaining only high-quality biallelic SNPs, with MAF > 0.05, at least four reads for a base call, and a minimum of 20 and a maximum of 200 total read counts for all populations. In total, 6,492,879 SNPs were identified for *E. carolleeae*, and 6,357,602 SNPs were identified for *E. gulfia* population comparisons.

To identify genomic signatures of selection linked to salinity adaptation in *E. carolleeae* and *E. gulfia* populations, a genome-wide analysis was performed using population fixation statistics ($F_{ST}$) and nucleotide diversity ($\theta_{\pi}$)[127], as implemented in the software grenedalf v0.6.3[128]. These analyses compared ancestral saline populations to their freshwater invading counterparts. $F_{ST}$ and $\log_2(\theta_{\pi\text{-invasive}}/\theta_{\pi\text{-native}})$ across the genome were calculated using sliding windows of 10 kb with a step size of 5 kb. Windows ranking in the top 5% for $F_{ST}$ and lowest 5% for $\log_2(\theta_{\pi\text{-invasive}}/\theta_{\pi\text{-native}})$ were identified as having signatures of selection between the saline and freshwater populations (referred to as "top 5% $F_{ST}$ and $\theta_{\pi}$ outliers"). To enhance the robustness of our selection signature detection, a more stringent threshold was also applied, identifying the top 1% of windows for high $F_{ST}$ and low $\theta_{\pi\text{-invasive}}/\theta_{\pi\text{-native}}$ values (referred to as "top 1% $F_{ST}$ and $\theta_{\pi}$ outliers").

In addition, our analysis included BayPass v2.41[56], a powerful tool for detecting loci significantly associated with environmental variables, such as salinity. This analysis involved estimating the posterior distributions of model parameters through multiple Markov Chain Monte Carlo (MCMC) procedures. These procedures included 15 pilot runs of 500 iterations, a burn-in phase of 2500 iterations, and 1000 MCMC samples with a thinning interval of 25, repeated over three independent runs. This approach allowed us to robustly estimate the variance-covariance matrix for SNP frequencies and assess the correlation between SNP alleles and salinity levels at the time of sample collection (Supplementary Data 23). SNP frequencies were standardized to a mean of 0 and variance of 1[56]. Bayes Factors (BFs) were calculated to quantify the strength of the association between each SNP and salinity, using the δ parameter from BayPass's auxiliary model with a default prior

distribution. The genomic windows that encompassed loci with a BF greater than 10, and that also ranked within the top 5% for both $F_{ST}$ and $\theta_\pi$ outliers, were identified as strong candidates that harbored selection signatures linked to freshwater invasions (referred to as "BayPass + 5% $F_{ST}$ and $\theta_\pi$ outliers"). SnpEff v4.3t[129] was used to annotate these genomic SNPs and selected genomic windows, based on the reference genome annotations of *E. carolleeae* and *E. gulfia*.

Importantly, statistical analyses were performed to determine whether the signatures of selection associated with salinity change were significantly enriched at the chromosomal fusion sites in the *E. carolleeae* and *E. gulfia* genomes. To approximate the background distribution of selection signatures for windows across each chromosome, $10^6$ 1 Mb (and 2 Mb) intervals were randomly sampled across each chromosome. Within these intervals, the number of 10 kb windows with signatures of selection were counted. The signatures of selection in our population genomic analyses were defined based on the number of 10 kb windows within 1 or 2 Mb intervals identified as top 5% $F_{ST}$ and $\theta_\pi$ outliers, top 1% $F_{ST}$ and $\theta_\pi$ outliers, and BayPass + 5% $F_{ST}$ and $\theta_\pi$ outliers (previous paragraphs).

Our approach for evaluating the significance of selection signatures at specific fusion sites involved comparing the number of 10 kb windows with signatures of selection in the 1 Mb (and 2 Mb) intervals encompassing each fusion site against the background genomic distribution of the corresponding chromosome. A fusion site was considered to have a significantly higher count of selection signatures than the background distribution of the corresponding chromosome if it fell into the upper 5% quantile of the background distribution. Kolmogorov-Smirnov (KS) and Cramér-von Mises (CM) goodness-of-fit tests were further employed to compare the selection signatures at all fusion sites combined on each chromosome against their respective background distribution of the corresponding chromosome. This test was performed to assess whether the fusion sites on each chromosome as a whole differed significantly from their background chromosome-wide distributions. Finally, KS and CM goodness-of-fit tests were conducted to compare all fusion sites combined for all chromosomes against the genome-wide average background distributions of all chromosomes, weighted by their respective lengths.

To assess whether chromosomal fusion sites exhibited reduced recombination and elevated linkage disequilibrium (LD), whole genome sequencing was performed for an additional 14 individuals of *E. carolleeae*. These copepods were collected in 2022 from Baie de L'Isle Verte, St. Lawrence Estuary, QC, Canada and frozen at −80 °C. DNA extracted from each individual copepod was amplified to satisfy sequencing requirements using the REPLI-g Single Cell Kit (No. 150343, Qiagen, Hilden, Germany). These amplified DNA samples were then sequenced on the NovaSeq 6000 platform, achieving coverage ranging from 30× to 46×, with a mean coverage of 35×. Data processing procedures were consistent with those described above. SNPs were called using GATK v4.6.1.0[130], and all GVCFs were merged for joint genotyping to generate SNPs. PopLDdecay[131] was utilized to calculate LD ($r^2$) and to generate patterns of LD decay across both fusion sites and background genomic regions. Statistical analyses were then performed to determine whether the chromosomal fusion sites exhibited elevated LD relative to the genomic background. To directly compare LD across equivalent genomic distances, a pairwise comparison of mean $r^2$ values was performed between 1 Mb (and 2 Mb) genomic regions encompassing fusion sites versus the genome-wide background at shared distance bins (20 bp–300 kb) using Paired *t*-tests in R.

To explore the demographic history of the native saline populations from *E. carolleeae* and *E. gulfia*, SMC + + v1.15.2[132] was employed for estimating changes in effective population size ($N_e$) over time. For this analysis, the mutation rate from *Drosophila melanogaster* of $2.8 \times 10^{-9}$ mutations per base pair per generation was used[133], coupled with an assumed generation time of three weeks for the *E. affinis* complex, as reported in previous studies[36,37].

## Data availability

The raw sequencing data and genome assemblies generated in this study have been deposited in the NCBI databases under BioProject PRJNA1075304, with raw reads available in the Sequence Read Archive (SRA) and genome assemblies available under accession numbers JBQWOI000000000 (*E. gulfia*, https://www.ncbi.nlm.nih.gov/nuccore/JBQWOI000000000) and JBQWOJ000000000 (*E. affinis* proper, https://www.ncbi.nlm.nih.gov/nuccore/JBQWOJ000000000). Genome assemblies and gene annotations are also available at figshare (https://doi.org/10.6084/m9.figshare.29104271). Previously published population genomic sequencing data from BioProject PRJNA610547 and transcriptome data from BioProject PRJNA278152 and BioProject PRJNA275666 were reanalyzed in this study (Supplementary Data 1). Source data are provided with this paper.

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

## Acknowledgements

This project was funded by National Science Foundation grants IOS-2412790, OCE-1658517, and DEB-2055356, and French National Research Agency ANR-19-MPGA-0004 (Macron's "Make Our Planet Great Again" award) to C.E.L. Nicholas Mathers and Alexander Taylor contributed to supervising the maintenance of copepod cultures. Monica Poelchau provided valuable suggestions on data management. Sarah Swanson provided training and access to the imaging facilities at the Newcomb Imaging Center at the Department of Botany, University of Wisconsin–Madison for imaging of chromosomes.

## Author contributions

C.E.L. and Z.D. conceptualized the study. Z.D. and J.W. conducted the genomic and statistical analyses. Z.D. and Y.J.Z. performed the laboratory experiments. A.J., T.O., A.P., C.M., G.W.G., and C.E.L. provided the necessary resources. Z.D. drafted the initial manuscript, and Z.D., J.W., and C.E.L. revised it. C.E.L. supervised the research project and secured the funding. All authors have read and approved the final manuscript.

## Competing interests

The authors declare no competing interests.
