## [Transparent Peer review file · Nature Communications]

Genome architecture evolution in an invasive copepod species complex

Corresponding Author: Dr Zhenyong Du

Version 0:

Reviewer comments:

Reviewer #1

(Remarks to the Author)
Significance

The following comments fall under both Significance and Analytical Approach sections. The authors have applied every approach imaginable to the problem of how adaptation to saline environments has allowed a species complex of copepod to invade saline habitats, with impacts on genome architecture. The authors have left 'no stone unturned' and their approach is akin to throwing wet strands of spaghetti against the wall to 'see what sticks.' This is not a weakness, because Guess What? Each approach contributed in one way or another to building a well supported argument that chromosomal evolution (writ broadly – at both gross and detailed levels) plays a major role in adaptation, and even as a mechanism of speciation.

Studies at the whole chromosome level have often been absent, or at best, undervalued, in modern evolutionary studies. Six or seven decades ago studies of chromosomal evolution were more in vogue (particularly focusing on inversions), followed by decades when studies of chromosomal structure fell out of favor as many believed they would shed little light on mechanisms of evolution. This study goes a long way toward reversing this thinking in the context of epigenetics. The proposed scenarios of chromosomal fission and fusion, in light of the breadth and depth of the published and new data presented here, paints an encouraging path to understanding evolution at several levels. The data are even relevant to mechanisms of speciation. In so many parts of the manuscript, concepts are tested and supported.

This amalgamation of diverse approaches and kinds of data at different levels of biological organization has resulted in a powerful story of how three clades of an estuarine copepod have invaded freshwaters to different degrees.

Data and methodology

Much of the value of the data lies in the fact that inbred lines formed the material for many of the studies, enabling the investigators to more easily 'make sense' of the data. This is uncommon in studies of copepods, as well as in many other 'model' organisms that are not 'lab rats.' Decades of work in the Lee lab on genetic basis of invasive ability in *Eurytemora* populations have been nicely integrated with new data.

Additionally, attention to transposable elements (TEs), was given to the Europe and Gulf clades. This also is also rare in studies of crustaceans. Integrating chromosomal and molecular evolution is novel, and so is incorporating TEs into the thinking about evolution. The discipline of molecular evolution of the populations and communities of TEs is in its infancy, and the authors probably analyzed their TE data to the full (but necessarily limited) extent possible. It is probably premature to conclude what any differences between the two clades might mean with respect to TEs, but the presentation of their survey is important to document. The presentation of data on TEs suggests paths to pursue in the future, and so the story of population level (speciation) in these *Eurytemora* clades is not over.

331 – 337: Despite the voluminous data set, the authors resist the temptation to provide interpretations that the data cannot support with confidence. For example, alternative models of ion uptake from freshwater are presented.

In some areas of the manuscript, reference is made to 'random' procedures. The methods section did not explain how randomness was achieved. For example, in the Randomization section of the study design, it seems more likely that samples used for sequencing were haphazardly collected from these beakers of inbred lines. Also, Table of Contents, line

22: how were 1 Mb intervals selected? In line 433 a reference is made to a randomly chosen subset of 25 genes from 253 gene. How was the random selection performed?

Figure S4, 68: I have some reservations about using the truly credible age of fossil *Daphnia* to calibrate the divergence times of any copepod species. Most *Daphnia* reproduce both clonally and sexually, with clonal reproduction likely predominating in any one year. Would rates of evolution differ between that reproductive strategy and obligate sexual reproduction used by copepods? Also, define the horizontal blue lines (a kind of confidence interval?). Elsewhere in the Discussion (lines 683-684) the authors refer to the uncertainty in these estimates of divergence time, which is appropriate.

Analytical approach

The *Eurytemora* species complex is a marvelous model for making headway into studying evolution at a variety of levels, because clades vary in their capacity to invade habitats new to them and their biological and genetic properties are so amenable to investigation. This study has worldwide impact, partly because of the geographically separated populations studied may allow some generalizable inferences, but also because investigators worldwide can follow up on these studies. The approaches yield a nice blend of descriptive data (e.g. DNA sequences, chromosomal counts) with application of models to estimate effective population size and timing of bottlenecks (Figure S11).

While the chromosomal figures in Figure S2 are not 'pretty', they suffice for making the arguments. "Pretty" chromosomal figures in copepods are rare and very difficult to obtain, for some reason. So, those unfamiliar with the chromosome biology of copepods should not judge these images harshly.

Suggested improvements: Clarity and context

The manuscript is well written. The organization is generally good, but is complicated a bit by the fact that some data of a particular kind are lacking for one of the 3 populations (= species). This lack of data for one population or another (e.g. no transcriptome data from *E. carolee* population) is not a fatal flaw; the number of approaches applied to all three populations is extraordinary and in instances where one approach might not have been applied to one of the clades, strong inferences can still be made. I found myself making a table of which kind of data was available for which population. I mention this as it might help the reader follow the flow of arguments in the published paper and how the different approaches build upon other approaches.

Throughout the manuscript I was asking myself "what about crossbreeding studies? Haven't they been done?" Crossbreeding studies are not even mentioned in the study design section. Then these studies were reported at the very end of the manuscript in the context of a manuscript in preparation. Perhaps these studies were viewed as less convincing evidence, as no backcrosses were performed and every possible combination of crosses was not performed? I can understand the reluctance to report unpublished data and considering the voluminous data set presented in the current version, a separate publication on crossbreeding is merited. I think a table of the list of approaches by population would alleviate this problem and encourage the reader to integrate the crossbreeding studies into their evaluation of the other kinds of data in a timeline that approximates presentation of other kinds of data. It is tricky to handle this, but some of the results were provided in the Discussion. A table will help organize the kinds of data in a way that will make it easier for the reader to understand how each piece fits into the big story. So even the crossbreeding studies (Weinberg and Lee, in prep) might be referenced here, without the supporting data.

Figures S5 & S6: Why are top 1% and 5% signatures of selection presented using histograms for the Atlantic clade, but a very different kind of graph in Figure S67 for the Gulf clade?. Is this related to a difference in resolution of the data? I do see the histogram form for the Gulf clade used in Figure S9 and S10

Minor comments:

I have comments about wording: the word 'novel' is sometimes appropriately used, but other times used in places where "new" would be more appropriate. For example,

67: By today's standards, integrating chromosomal evolution into studies of adaptation at the molecular level is novel.

But 137: transition into novel salinities is not novel. Rather, certain salinities are 'new' or 'foreign' to *Eurytemora*.

Caption of Table S10 should define the percentage values in the position column.

609: A brief reference is made to acclimatory shifts across salinities (Popp et al in review). Can the authors expand on this?

656: I think 'experiment' rather than 'experimental' is intended.

663-664 Casual language such as 'coming together' has a vague context.

By the way, I appreciated having the figures embedded in the manuscript for review purposes.

References

The literature is extremely well covered. In addition to supporting the authors' arguments this manuscript could serve as a review of relevant literature to the topic of adaptation to new environments in the context of chromosomal evolution at the detailed (e.g. HiC sequencing of DNA) and broader (chromosome counts) levels.

Reviewer #2

(Remarks to the Author)

Noteworthy results - This manuscript presents a fascinating study of adaptation via chromosomal fusions, using repeated bouts of salinity adaptation in a copepod species complex. Much of the "supergene" literature has been focused on inversions, but this study provides an interesting case of co-localization of ion transporter genes within two centromeres in the highly invasive clade in this complex. It's a comprehensive study that includes whole genome assemblies, phylogenetics, identification of ancestral groups, gene ontology, gene expansion, and population genomics. I commend the authors on bringing all this data together and I hope they find my comments constructive.

Significance - The work is in many ways a synthesis of the senior authors research in this study system, which has already shown the co-localization. What makes this paper a significant advancement is the addition of whole genome sequencing for other species in the complex. The work is of significance in the fields of evolutionary biology, genome biology, and physiology.

Interpretation and Presentation of evidence in support of conclusions:

Overall, the analyses are thorough and support the conclusions regarding chromosomal fusion. Except for Figure 7 and 8 (see below), the figures are excellent, easy to interpret, and provide evidence in support of the conclusions. Some of the discussion of the physiology of ion uptake (e.g. Figure 5) could be made more accessible to a wider audience - see specific comments below. There are some parts of the discussion with regard to the functional advantages of co-localization of ion transporter genes that draw largely on unpublished studies and data, which may need major revisions prior to publication (specific comments below).

Given that a main conclusion is that this arrangement of genes was a "supergene" favored due to suppressed recombination within centromeres, I was expecting an analysis that showed elevated linkage disequilibrium between these genes in the Atlantic clade as opposed to the other clades. The current version of the manuscript lacks this analysis, which weakens the main "supergene" conclusion. The authors could use their population genomic data to conduct an LD analysis and show that there is indeed elevated LD, which would provide stronger support for their interpretation. Thus, while the reorganization of genes in the genomes was well supported, the "supergene" conclusion was not as well supported by the analyses.

Methodology and Reproducibility:

With a few exceptions with the data analysis, the methodology meets the expected standards in the field and are described in sufficient detail. A few areas of the methods where minor revisions would improve reproducibility (specific comments below).

Data analysis:

The presentation and analysis of the F_{ST} selection scan could be more rigorous. The transformation of F_{ST} to a z-score is unconventional, and it makes Figure 8a confusing (a z-score should have positive and negative values). Plotting actual F_{ST} would be biological interpretable. In addition, it would be more rigorous to identify F_{ST} outliers above what is expected by neutrality, rather than arbitrarily use the top 5% of F_{ST} values. This control for structure is important in selection scans, because demographic processes can inflate the variance of the F_{ST} distribution. There are several programs designed for this kind of analysis (OutFLANK, pcadapt, Bayesian, BayPass).

Specific comments:

The use of red and green color in Figure 1 is not color-blind friendly. Adding a legend for dots and triangles in Figure 1 would assist interpretation, so readers do not have to search for the descriptive in the middle of the caption.

Figure 5 is not really interpretable to readers who do not have expertise in ion uptake. It would be more relevant to compare intracellular ion control in a salinity vs. freshwater environment. Explaining how differences in the transport of specific ions leads to higher fresh vs. higher salinity tolerance would greatly improve the accessibility of this section to a larger audience. Showing the relative abundance of different ions in salinity vs. freshwater, versus within the cell, would also help readers.

Line 419 is the first introduction of results from "selection signatures", but this is a broad term and it would be helpful for readers here to describe the specific signatures.

Line 444: typo in "and all paralogs"  "all ARE paralogs" (?)

Figure 7

7a - including a visual legend for the centromeres would be helpful, so readers don't need to scour the text. I may have missed it, but it would be helpful to specifically address the lack of a centromere on Chromosome 2.

7b-d: It would be helpful to show the location of the centromeres in the Atlantic clade.

Figure 7d appears to be missing the colored ribbons? Please add for continuity.

With this figure, it's confusing that multiple colors of ion transporting genes are shown in 7a, but only a subset of those are shown in b-d. It feels a bit like the authors are cherry-picking genes that show localization to the centromeres.

Figure 8

Figure 8 has a lot going on, and it would be helpful for readers to include a visual legend for the centromeres, fusion points, terminology (e.g. what is "C1S1" etc), and green points. I could not see the dark blue dashed line in the figure. There are no "vertical dashed lines", do you mean vertical grey bars? It would be helpful to also see where the fusion sites are in Figure 7, which would provide readers with some reference points between the two figures. Plotting F-st in Figure 8a-d would be more interpretable than Z-FST.

Discussion

The discussion is almost entirely focused on the study species, which greatly limits its appeal to a broader audience. Addition some literature about supergene evolution and salinity adaptation in other species would greatly improve its appeal.

Compared to the rest of the paper, the discussion is not as well organized and has some repetitive text. Some revisions and reorganization here would greatly improve the manuscript. It would also be helpful at the beginning of the discussion to remind the readers about the ecology and history of the three clades, rather than lower down. I thought the discussion would flow better if it was organized in the opposite manner that describes the temporal evolution of the system: starting with "divergence and chromosomal evolution", then "central repositioning" of ion transporter genes, then "impacts of the new genomic architecture" on selection response. The section on "enrichment of selection signatures" was largely repetitive of the results, and could be reduced and integrated into the other sections.

Sections of the "central repositioning" are based on unpublished research - the authors should check with the journal guidelines and make sure that In Review and In Prep citations can be included (typically they cannot). Regardless, this section could be improved by bringing in some literature on coordinated regulation of gene expression and GWAS in other species.

While much of the message is focused on what is happening at the centromeres in the Atlantic clade, the Gulf clade also adapts to salinity and does not have such fusions. Thus, chromosomal fusions are not a prerequisite for salinity adaptation and this point feels lost in the discussion.

Line 673- "beneficial reversal of dominance" comes out of nowhere - either this needs some more context or could be removed

Methods

How were the RNA libraries prepared? It read as if the analysis was based on total RNA as opposed to mRNA.

Lines 868-869: "annotated using a comprehensive approach..." was vague. Please add detail or a reference.

"Population genomic analysis" section was missing the type of sequencing (presumably Illumina) and the average coverage per pool.

Overall impression

I enjoyed reviewing this paper and commend the authors on a nearly thorough job. With an LD analysis, a more rigorous and interpretable FST outlier analysis, and some revisions, it will be a strong contribution to the literature.

Reviewer #3

(Remarks to the Author)
see attached comments

Reviewer #4

(Remarks to the Author)

Version 1:

Reviewer comments:

Reviewer #1

(Remarks to the Author)

The author has addressed all of my suggestions and concerns in this revision.

Reviewer # 1

Reviewer #2

(Remarks to the Author)

The authors have done a very thorough and thoughtful job in addressing the extensive comments by the reviewers. The figures, results, and discussion are more streamlined and accessible. I recommend the manuscript for publication.

Reviewer #3

(Remarks to the Author)

In the present manuscript, Du et al have substantially revised their analysis of copepod genome evolution, providing useful clarification and specificity to the conclusions, as well as conducting numerous new analyses. Overall, we feel that the manuscript is greatly improved and, in our opinion, that this work merits publication in Nature Communications, pending some remaining modest revisions. In general, the work is excellent, and we commend the authors on their achievement.

In our previous review letter, we raised a few minor concerns stemming from semantic issues and from a handful of missing analyses that could greatly strengthen the core observations. The authors addressed these comments. Resolution of the two semantic misunderstandings by describing deeply-diverged lineages as 'species' rather than 'populations' now makes the findings easier to understand, and use of a stricter definition of supergenes and limiting discussion of them is also helpful. Our minor comments about analyses that could strengthen the work also were addressed; the addition of new selection analyses and, especially, tests for suppressed recombination, have strengthened the evidence for the key findings.

We also raised a major concern about the earlier draft: certain passages in the text led us to believe the authors were using contemporary signatures of selection as a test of whether ancient chromosomal fusions were positively selected once they arose. We appreciate that the authors now intend to present these findings in two separate analyses. First, they present analyses of ancient evolution that strengthen and extend this group's prior claim that the chromosomal fusions were non-random and therefore presumably adaptive – possibly due to a particular bias for clustering functionally-similar ion channel genes in low-recombination regions near centromeres. The key new finding of the current paper comes from analyses of contemporary evolution showing that these centromeric ion channel clusters do indeed display low recombination and are hotspots of recent selection that may facilitate the transition between salt and fresh water habitats. A critical goal of the text, then, should be to make it clear that no causal link between a) and b) can be drawn with the present data: the selective forces that caused certain types of chromosomal fusions to be preferentially retained must be clearly flagged as unknown, and the focus should instead be on the adaptive value that the presence of these fusions provided to contemporary populations. We appreciate the possibility mentioned in the response letter: The "long evolutionary history of salinity fluctuations in the genus *Eurytemora*" might be a common cause for ancient as well as contemporary selection. This is a plausible hypothesis that merits description in the Discussion, but we ask that it be kept out of the key results and conclusions (unless direct tests of this idea can be devised).

Only relatively minor issues remain to resolve this ambiguity. Therefore, we list several opportunities to improve the manuscript further, and we organize these into three categories:

1) In certain parts of the Abstract, Introduction, Results, and Discussion, ancient and contemporary evolution are discussed back-and-forth in a way that can make them seem conflated. We suggest that the authors attempt to reorganize all relevant sections by discussing ancient evolution first (i.e., evidence that fusions were adaptive for an unknown reason), then discuss contemporary evolution second (i.e., regardless of why they exist, the presence of these fusions has facilitated contemporary evolution). The discussion can then discuss the possible causal link between these evolution in these two epochs, but this should be mentioned as a hypothesis rather than a key conclusion of the work.

2) We are skeptical of the specific findings about the timeline of chromosomal fusions. We understand that such ancient events are inherently difficult to reconstruct, so we don't feel that it is necessary to unambiguously resolve this in the paper. If the authors would like to provide additional analyses that do resolve it, that would be great, but we are not requesting this; it is perfectly acceptable to publish the findings as is, as long as the authors soften some of their conclusions and add clarifying text to emphasize that multiple scenarios of chromosomal fusion remain possible.

3) Ambiguity in the language of some passages hinders the reading and comprehension. For instance, the passage mentioned above, the "long evolutionary history of salinity fluctuations in the genus *Eurytemora*" should be amended to the "long evolutionary history of salinity fluctuations experienced by copepods [or individuals] in the genus *Eurytemora*" because the fluctuations clearly do not occur in the genus but in the environment that the members of a given taxon inhabit. Similar or other problems arise on lines 70-71 (distribution...are), line 107 (a species complex cannot be a grazer, only its constituent individual members), line 124 (I doubt that entire populations were transported), line 135 (what sibling species? The term generally only makes sense with respect to another species), and elsewhere.

Minor concern 1

In the Abstract, the sentence at Lines 53-55 pertains to ancient evolution, then 55-57 pertains to contemporary evolution, then 57-59 returns to ancient evolution. Would it be possible to put the two ancient evolution sentences together, then the

contemporary evolution sentence to follow?

Lines 83-84 are great.

Lines 102-104: We feel that this expresses the key novelty of the paper. Past work and work in preparation by the authors has focused on the non-random nature of these fusions. It is helpful that, to a modest degree, the authors replicated and extended those findings here. However, the primary focus should be as discussed in this paper: asking whether those ancient adaptive fusions are relevant to contemporary selection.

Lines 146-151: This section is extremely important because it outlines the motivation for this study, but it does not do a good job of clarifying that there are actually two hypotheses: 1) whether ancient fusions were adaptive, and 2) whether contemporary populations are experiencing selection at these fusion sites. We hope the authors can rephrase this to make it more clear.

Lines 155-159: Similar issues as before. First, (1) and (3) are ancient but (2) is contemporary; can this be re-ordered? Second, can (1) and (3) be broken into a separate sentence from (2) to emphasize the distinct hypotheses for these distinct epochs?

Lines 415-418: This sentence conflates the distinct timescales so thoroughly that it is likely to mislead the average reader. We feel that the discussion of contemporary evolution is unwarranted in a section titled "Chromosomal fusions reposition ion transport-related genes within genomes;" we request that this section be used solely to discuss ancient events so that the contemporary evolution can be solely and separately discussed in the following section.

Lines 437-439: Same issue; the inclusion of this sentence amidst a discussion of events that happened much earlier in geological time is confusing. We suggest that this citation of previous work be moved to the introductory text describing your analyses of contemporary evolution.

Line 450: Can a new title be given for this and the following paragraphs? The previous title no longer applies; we aren't talking about repositioning of genes in this part, but instead about the selection those repositioned genes are experiencing.

Lines 473-475: This summary sentence is out of place; it belongs at line 449, by which point the non-random nature of the repositioning was already established. A summary sentence for this section should instead summarize the new results (that those repositioned genes are under contemporary selection), then set up the subsequent attempt to understand why they are under selection.

Lines 607-619: This section does an excellent job at clarifying the key ideas.

Minor concern 2

We request some clarification on the parallel chromosome reduction hypothesis positing that the ancestor of *gulfia*+*carolleae*, the ancestor of the *E. affinis* clade, and extant *E. affinis* all had 15 chromosomes. We agree with the argument (Lines 397-399) that it is not parsimonious to propose 7 chromosomes in the *gulfia*+*carolleae* ancestor, and understand that the statistical package used inferred a 15-chromosome ancestor. However, this claim seems to suggest an implausible degree of parallel evolution.

Based on the plots in Figure 6b,c, and allowing for lineage-specific inversions, both *gulfia* and *carolleae* seem to share six fusions relative to *E. affinis*: 1+2, 3+4, 5+6, 8+9, 12+13, and 14+15. Manually checking each of these in Figure 6d seems consistent with a homologous origin: if we allow for ~2-3 inversions, all of these fusions seem to represent near-perfect alignment between the two species. In contrast, if the fusions happened at different breakpoints, wouldn't we expect Figure 6d to show small slivers of sequence at each fusion that align elsewhere (or not at all)? Based on this reasoning, our quick guess was that the ancestor of *gulfia*+*carolleae* had 9 chromosomes. *E. carolleae* would then have 5 lineage-specific fusions (1+2 to 3+4 | 7 to 5+6 | 10 to 8+9 | 11 to 8+9+10 | 12+13 to 14+15). *E. affinis* would have two lineage-specific fusions (10 to 5+6 | 7 to 11).

We are not sure whether the above hypothesis is consistent with your data or if your evidence for a 15-chromosome ancestor is definitive already (e.g., perhaps you have observed that breakpoints always differ in the "shared" fusions we proposed above?). If it is definitive, this needs to be made clearer in the Results, and the extraordinarily nature of the parallel fusions should be highlighted more directly in the Discussion.

If it is not definitive, (i.e., a scenario like the one we outlined above is also possible) then we request that the section be altered to emphasize the fact that multiple ancestral arrangements are consistent with the data. This would not undermine the major findings of the paper because the key conclusion in Lines 397-399 is certainly true even if the 15-chromosome ancestor is not supported. Note that if this path is taken, some of the interpretation elsewhere in the paper will also need to be softened (i.e., Line 52 of the Abstract) to make it clear that the precise number of independent versus shared fusions is not known. Alternatively, it would of course be great if the authors find a way to unambiguously determine the ancestral chromosome number (i.e., by using a breakpoint analysis), but the authors can best judge whether new analyses of that kind would represent feasible and worthwhile additions at this late stage.

Summary

The authors have done an admirable job of substantially strengthening their work. Our minor comments from the prior draft were addressed and our major comment was largely addressed, with only a few minor concerns remaining. If these issues can be resolved, we think the paper will become a valuable contribution to the field and merits publication in Nature Communications.

Response to the Reviewers' Comments:

Reviewers' comments are in black, whereas authors' responses are in dark blue.

Reviewer #1 (Remarks to the Author):

Significance

The following comments fall under both Significance and Analytical Approach sections. The authors have applied every approach imaginable to the problem of how adaptation to saline environments has allowed a species complex of copepod to invade saline habitats, with impacts on genome architecture. The authors have left 'no stone unturned' and their approach is akin to throwing wet strands of spaghetti against the wall to 'see what sticks.' This is not a weakness, because Guess What? Each approach contributed in one way or another to building a well supported argument that chromosomal evolution (writ broadly – at both gross and detailed levels) plays a major role in adaptation, and even as a mechanism of speciation.

Studies at the whole chromosome level have often been absent, or at best, undervalued, in modern evolutionary studies. Six or seven decades ago studies of chromosomal evolution were more in vogue (particularly focusing on inversions), followed by decades when studies of chromosomal structure fell out of favor as many believed they would shed little light on mechanisms of evolution. This study goes a long way toward reversing this thinking in the context of epigenetics. The proposed scenarios of chromosomal fission and fusion, in light of the breadth and depth of the published and new data presented here, paints an encouraging path to understanding evolution at several levels. The data are even relevant to mechanisms of speciation. In so many parts of the manuscript, concepts are tested and supported.

This amalgamation of diverse approaches and kinds of data at different levels of biological organization has resulted in a powerful story of how three clades of an estuarine copepod have invaded freshwaters to different degrees.

We sincerely appreciate the reviewer's thoughtful and supportive comments on the comprehensiveness of our study. We are grateful for the recognition of our efforts.

Data and methodology

Much of the value of the data lies in the fact that inbred lines formed the material for many of the studies, enabling the investigators to more easily 'make sense' of the data. This is uncommon in studies of copepods, as well as in many other 'model' organisms that are not 'lab rats.' Decades of work in the Lee lab on genetic basis of invasive ability in *Eurytemora* populations have been nicely integrated with new data.

Additionally, attention to transposable elements (TEs), was given to the Europe and Gulf clades. This also is also rare in studies of crustaceans. Integrating chromosomal and molecular evolution is novel, and so is incorporating TEs into the thinking about evolution. The discipline of molecular evolution of the populations and communities of TEs is in its infancy, and the authors probably analyzed their TE data to the full (but necessarily limited) extent possible. It is probably premature to conclude what any differences between the two clades might mean with respect to TEs, but the presentation of their survey is important to document. The presentation of data on TEs suggests paths to pursue in the future, and so the story of population level (speciation) in these *Eurytemora* clades is not over.

331 – 337: Despite the voluminous data set, the authors resist the temptation to provide interpretations that the data cannot support with confidence. For example, alternative models of ion uptake from freshwater are presented.

We thank the reviewer for these thoughtful and detailed assessments of the data and methodology in our manuscript.

We appreciate the acknowledgment of our work on transposable elements (TEs) in the Europe and Gulf clades, an area seldom explored in crustacean studies.

We also appreciate the acknowledgment of our cautious approach toward data interpretation, especially on speculative topics, such as ion uptake models in fresh water.

1. In some areas of the manuscript, reference is made to ‘random’ procedures. The methods section did not explain how randomness was achieved. For example, in the Randomization section of the study design, it seems more likely that samples used for sequencing were haphazardly collected from these beakers of inbred lines. Also, Table of Contents, line 22: how were 1 Mb intervals selected? In line 433 a reference is made to a randomly chosen subset of 25 genes from 253 gene. How was the random selection performed?

We thank the reviewer for identifying the need for more detailed descriptions of the randomized procedures employed in our study.

***In response to this comment, we have updated the Methods section to more precisely detail how randomness was achieved throughout our experimental design.

Specifically, regarding copepod sample selection for sequencing, we added a new sentence on Page 28, Lines 825–826 as follows:

“The copepods were randomly selected with an approximate sex ratio of 1:1 from our beakers of inbred lines to ensure a representative mix.”

Regarding the random selection processes of 1 Mb intervals and subset of 25 chosen genes, we added a new sentence on Page 32, Lines 977–980 as follows:

“The randomization sampling processes in our study were all conducted using the ‘sample’ function in R, which employs the Mersenne Twister algorithm to generate random numbers. This approach ensures an impartial and unbiased sampling process.”

2. Figure S4, 68: I have some reservations about using the truly credible age of fossil *Daphnia* to calibrate the divergence times of any copepod species. Most *Daphnia* reproduce both clonally and sexually, with clonal reproduction likely predominating in any one year. Would rates of evolution differ between that reproductive strategy and obligate sexual reproduction used by copepods? Also, define the horizontal blue lines (a kind of confidence interval?). Elsewhere in the Discussion (lines 683-684) the authors refer to the uncertainty in these estimates of divergence time, which is appropriate.

We appreciate the reviewer’s feedback regarding the use of *Daphnia* fossil ages for calibrating divergence times in copepod species. Given the reproductive strategies of *Daphnia*, which may affect evolutionary rates, we agree that relying on their fossil record for calibration could introduce biases.

***In response to this comment, we have excluded Supplementary Figure S4 and revised our results and methods to solely utilize divergence time estimates based on the calibration from the TIMETREE database (shown in Figure 3).

We also updated the figure legend of Figure 3 on Page 9, Lines 251–253 as follows:

“Mean estimated divergence times are shown at each node, with the horizontal blue lines and numbers in parentheses indicating 95% highest posterior density intervals.”

Analytical approach

The *Eurytemora* species complex is a marvelous model for making headway into studying evolution at a variety of levels, because clades vary in their capacity to invade habitats new to them and their biological and genetic properties are so amenable to investigation. This study has worldwide impact, partly because of the geographically separated populations studied may allow some generalizable inferences, but also because investigators worldwide can follow up on these studies.

The approaches yield a nice blend of descriptive data (e.g. DNA sequences, chromosomal counts) with application of models to estimate effective population size and timing of bottlenecks (Figure S11).

While the chromosomal figures in Figure S2 are not ‘pretty’, they suffice for making the arguments. “Pretty” chromosomal figures in copepods are rare and very difficult to obtain, for some reason. So, those unfamiliar with the chromosome biology of copepods should not judge these images harshly.

We are grateful for the acknowledgment of our analytical approaches. Regarding the chromosomal figures in Supplementary Figure 2, we appreciate the recognition of the challenges of producing high-quality karyotype images for copepods.

Suggested improvements: Clarity and context

3. The manuscript is well written. The organization is generally good, but is complicated a bit by the fact that some data of a particular kind are lacking for one of the 3 populations (= species). This lack of data for one population or another (e.g. no transcriptome data from *E. carolee* population) is not a fatal flaw; the number of approaches applied to all three populations is extraordinary and in instances where one approach might not have been applied to one of the clades, strong inferences can still be made. I found myself making a table of which kind of data was available for which population. I mention this as it might help the reader follow the flow of arguments in the published paper and how the different approaches build upon other approaches.

We appreciate the positive feedback on the manuscript’s structure and the suggestion to clarify data availability across the populations and sibling species studied.

To clarify, we have extensive transcriptome data for *Eurytemora carolleae* from previous studies, with 49 datasets collected (Eyun et al. 2017; Posavi et al. 2020). All these data were used in the structural annotation of our two new genomes.

***In response to this comment, we have added a new table, Supplementary Table 1. This table details the types of genomic resources available for each of the three sibling species studied. In this table, we clearly list the transcriptomes, genomes (and sequencing platforms), and accession numbers for each sibling species.

4. Throughout the manuscript I was asking myself “what about crossbreeding studies? Haven’t they been done?” Crossbreeding studies are not even mentioned in the study design section. Then these studies were reported at the very end of the manuscript in the context of a manuscript in preparation. Perhaps these studies were viewed as less convincing evidence, as no backcrosses were performed and every possible combination of crosses was not performed? I can understand the reluctance to report unpublished data and considering the voluminous data set presented in the current version, a separate publication on crossbreeding is merited. I think a table of the list of approaches by population would alleviate this problem and encourage the reader to integrate the crossbreeding studies into their evaluation of the other kinds of data in a timeline that approximates presentation of other kinds of data. It is tricky to handle this, but the some of the results were provided in the Discussion. A table will help organize the kinds of data in a way that will make it easier for the reader to understand how each piece fits into the big story. So even

the crossbreeding studies (Weinberg and Lee, in prep) might be referenced here, without the supporting data.

We do cite prior crossbreeding studies between clades (Lee 2000). This previous study includes the results of crossbreeding between three different clades (Atlantic, North Pacific, and North Atlantic clades).

We have also performed crossbreeding studies specifically between Atlantic × Gulf clades (Weinberg and Lee, In Prep.) and Atlantic × Europe clades (Lee et al., In Prep.). For the Atlantic × Gulf clade crosses, we find that F1 offspring are often produced, but that F2 hybrids are produced in only one reciprocal cross. For the Atlantic × Europe clade crosses, we obtain F1 cross in only one reciprocal cross and F2 egg clutches are produced but fail to hatch. These studies are being prepared for publication.

Given that these latter studies (Weinberg and Lee, In Prep; Lee et al., In Prep) are not yet published, we are citing the Lee (2000) study in this manuscript on Pages 4, 6, and 26.

***In response to this comment, we clarified on Page 26, Lines 761–763 that crossbreeding studies have been performed between genetically distinct clades (sibling species) of the *E. affinis* complex, and we cited Lee (2000).

5. Figures S5 & S6: Why are top 1% and 5% signatures of selection presented using histograms for the Atlantic clade, but a very different kind of graph in Figure S67 for the Gulf clade?). Is this related to a difference in resolution of the data? I do see the histogram form for the Gulf clade used in Figure S9 and S10.

We appreciate the reviewer’s comment regarding these Supplementary Figures.

To clarify, Supplementary Figure S7 (now Supplementary Figure 11) shows the distribution of selection signatures across the seven chromosomes of the Gulf clade, corresponding directly to Figures 7a–d for the Atlantic clade.

Meanwhile, Supplementary Figures 13–18 for the Gulf clade present histograms showing the genome-wide distribution of the top 5% and 1% F_{ST} and θ_{π} outliers, and BayPass + 5% F_{ST} and θ_{π} outliers within 1 Mb and 2 Mb intervals. These figures directly correspond to Supplementary Figures 5–10 for the Atlantic clade, which use the same graphical format and analytical approaches.

***In response to this comment, we have added clarifying sentences to the legends of the relevant supplementary figures to explicitly indicate the counterpart graphs between *E. carolleae* and *E. gulfia*. For example, the legend of Supplementary Figure 11 now includes:

“This figure corresponds to Figs. 7a–d for the chromosome plots of *E. carolleae*.”

Similarly, the legend of Supplementary Figure 5 now states:

“This figure corresponds to Supplementary Fig. 12 for the same analysis performed in *E. gulfia*.”

Minor comments:

6. I have comments about wording: the word ‘novel’ is sometimes appropriately used, but other times used in places where “new” would be more appropriate. For example, 67: By today’s standards, integrating chromosomal evolution into studies of adaptation at the molecular level is novel.

But 137: transition into novel salinities is not novel. Rather, certain salinities are ‘new’ or ‘foreign’ to Eurytemora.

We appreciate the reviewer's comment on the usage of the terms "novel" and "new."

***In response to this comment, we have made the revisions as suggested, and changed "novel" to "new" on Page 4, Lines 130–133.

7. Caption of Table S10 should define the percentage values in the position column.

In this table (now Supplementary Table 11), we now include an explicit definition of the "Position" column at the bottom of the table as follows:

"The positions of these genes are indicated by their distances from the nearest edges of each chromosome divided by the chromosome length, with values ranging from 0% (edges of chromosome) to 50% (center of chromosome)."

8. 609: A brief reference is made to acclimatory shifts across salinities (Popp et al in review). Can the authors expand on this?

Now on Page 23, Lines 663–665, we cite the full reference of Popp et al., as this study is now published.

***In response to this comment, we have modified the sentence to be more specific, as follows:

"Our previous results had found the coordinated regulation of gene expression of these ion transporters and carbonic anhydrase, and their colocalization within ion transporting cells (Popp et al. 2024)."

We refer to this previous study to indicate that the set of ion transporter genes are coordinately regulated and that they are coexpressed in ion transporting cells. As the other reviewers have requested that we shorten the Discussion section, we provide more detail in the sentence above, but do not elaborate further.

9. 656: I think 'experiment' rather than 'experimental' is intended.

This has been corrected as suggested on Page 24, Line 707.

10. 663-664 Casual language such as 'coming together' has a vague context.

We have revised this sentence on Page 24, Lines 711–715 as follows:

"The extent of parallelism among the selection lines increased with more generations of selection, favoring the same freshwater-adapted alleles. This result indicated that recombination events, which would be more frequent with greater numbers of chromosomes, **brought together freshwater-adapted alleles within genomes**, likely facilitated by positive synergistic epistasis among the beneficial alleles (Stern et al. 2022)."

By the way, I appreciated having the figures embedded in the manuscript for review purposes.

We thank the reviewer for recognizing our efforts to make this manuscript easier to read for the reader.

References

The literature is extremely well covered. In addition to supporting the authors' arguments this manuscript could serve as a review of relevant literature to the topic of adaptation to new environments in the context of chromosomal evolution at the detailed (e.g. HiC sequencing of DNA) and broader (chromosome counts) levels.

We thank the reviewer for this supportive feedback regarding the comprehensive coverage of the background literature in our manuscript.

Reviewer #2 (Remarks to the Author):

Noteworthy results - This manuscript presents a fascinating study of adaptation via chromosomal fusions, using repeated bouts of salinity adaptation in a copepod species complex. Much of the “supergene” literature has been focused on inversions, but this study provides an interesting case of co-localization of ion transporter genes within two centomeres in the highly invasive clade in this complex. It’s a comprehensive study that includes whole genome assemblies, phylogenetics, identification of ancestral groups, gene ontology, gene expansion, and population genomics. I commend the authors on bringing all this data together and I hope they find my comments constructive.

Significance - The work is in many ways a synthesis of the senior authors research in this study system, which has already shown the co-localization. What makes this paper a significant advancement is the addition of whole genome sequencing for other species in the complex. The work is of significance in the fields of evolutionary biology, genome biology, and physiology.

We truly appreciate the reviewer’s recognition of our efforts to synthesize vast amounts of data for this study.

Interpretation and Presentation of evidence in support of conclusions:

11. Overall, the analyses are thorough and support the conclusions regarding chromosomal fusion. Except for Figure 7 and 8 (see below), the figures are excellent, easy to interpret, and provide evidence in support of the conclusions. Some of the discussion of the physiology of ion uptake (e.g. Figure 5) could be made more accessible to a wider audience - see specific comments below. There are some parts of the discussion with regard to the functional advantages of co-localization of ion transporter genes that draw largely on unpublished studies and data, which may need major revisions prior to publication (specific comments below).

We appreciate this reviewer’s valuable suggestions for improving the figures and discussion.

***In response to this comment, we have extensively revised Figures 5, 7 (now Figure 6), and Figure 8 (now Figure 7) to enhance clarity and interpretability (see specific responses below).

We have also extensively rewritten the Discussion section. We removed references to unpublished data and instead focused our discussion on findings supported by published sources and data included in our study.

12. Given that a main conclusion is that this arrangement of genes was a “supergene” favored due to suppressed recombination within centromeres, I was expecting an analysis that showed elevated linkage disequilibrium between these genes in the Atlantic clade as opposed to the other clades. The current version of the manuscript lacks this analysis, which weakens the main “supergene” conclusion. The authors could use their population genomic data to conduct an LD analysis and show that there is indeed elevated LD, which would provide stronger support for their interpretation. Thus, while the reorganization of genes in the genomes was well supported, the “supergene” conclusion was not as well supported by the analyses.

We appreciate the reviewer’s useful suggestion to substantiate our “supergene” conclusion with linkage disequilibrium (LD) analysis. The population genomic results shown in the initial

manuscript, based on Pool-seq data from wild populations, did not allow for accurate estimation of LD patterns.

***In response to this comment, we have now included new analyses based on data from whole-genome sequencing of 14 additional individuals of *E. carolleae* collected from Baie de L'Isle Verte, St. Lawrence Estuary, QC. We performed individual sequencing with coverage ranging from 30× to 46×, with a mean coverage of 35×. Using PopLDdecay, we calculated LD (r^2) across genomic regions, with a focus on comparing fusion sites versus genome-wide background regions.

Our analysis revealed that all fusion sites exhibited significantly higher LD levels relative to the genome-wide background (Supplementary Figure 18). In addition, the fusion sites that contained significant enrichment of selection signatures showed even higher LD relative to all fusion sites. These results provide robust support for elevated LD at the fusion sites, especially the fusion sites with signatures of selection, and provide further support for our “supergene” hypothesis.

We added new results on LD analysis in the Results section on Page 21, Lines 586–599 as follows:

“Regarding patterns of recombination across the genome, the *E. carolleae* genome showed evidence of reduced recombination at the fusion sites compared to the genome-wide background (Supplementary Fig. 18). We investigated patterns of linkage disequilibrium (LD) across the genome, using whole-genome sequences of 14 individual copepods, to evaluate whether fusion sites exhibit signatures of reduced recombination relative to genome-wide levels. This analysis was motivated by the hypothesis that chromosomal fusions are clustering co-adapted loci and alleles into regions of suppressed recombination. This clustering of co-adapted alleles would then facilitate their joint inheritance, with the potential to promote adaptive evolution.

Notably, the fusion sites of the *E. carolleae* genome did indeed exhibit significantly elevated LD in both 1 Mb (Paired t -test; $t = 264.4$, $df = 299,980$, $P < 0.0001$) and 2 Mb genomic intervals (Paired t -test; $t = 275.1$, $df = 299,980$, $P < 0.0001$). Fusion sites with significant enrichment of selection signatures demonstrated even higher LD relative to all fusion sites (Supplementary Fig. 18). Recombination patterns were not explicitly analyzed for *E. gulfia* in this study due to the lack of genome sequences for individual copepods.”

We also added new methods on LD analysis in the Methods section on Page 34, Lines 1047–1059 as follows:

“To assess whether chromosomal fusion sites exhibited reduced recombination and elevated linkage disequilibrium (LD), whole genome sequencing was performed for an additional 14 individuals of *E. carolleae*. These copepods were collected in 2022 from Baie de L'Isle Verte, St. Lawrence Estuary, QC, Canada and frozen at -80°C . DNA extracted from each individual copepod was amplified to satisfy sequencing requirements using the REPLI-g Single Cell Kit (Qiagen, Hilden, Germany). These amplified DNA samples were then sequenced on the NovaSeq 6000 platform, achieving coverage ranging from 30× to 46×, with a mean coverage of 35×. Data processing and SNP calling procedures were consistent with those described above. PopLDdecay¹³⁰ was utilized to calculate LD (r^2) and to generate patterns of LD decay across both fusion sites and background genomic regions. Statistical analyses were then performed to determine whether the chromosomal fusion sites exhibited elevated LD relative to the genomic background. To directly compare LD across equivalent genomic distances, a pairwise comparison of mean r^2 values was performed between 1 Mb (and 2 Mb) genomic regions encompassing fusion sites versus the genome-wide background at shared distance bins (20 bp–300 kb) using Paired t -tests in R.”

Methodology and Reproducibility:

With a few exceptions with the data analysis, the methodology meets the expected standards in the field and are described in sufficient detail. A few areas of the methods where minor revisions would improve reproducibility (specific comments below).

Data analysis:

13. The presentation and analysis of the F_{ST} selection scan could be more rigorous. The transformation of F_{ST} to a z-score is unconventional, and it makes Figure 8a confusing (a z-score should have positive and negative values). Plotting actual F_{ST} would be biologically interpretable. In addition, it would be more rigorous to identify F_{ST} outliers above what is expected by neutrality, rather than arbitrarily use the top 5% of F_{ST} values. This control for structure is important in selection scans, because demographic processes can inflate the variance of the F_{ST} distribution. There are several programs designed for this kind of analysis (OutFLANK, pcadapt, Bayesian, BayPass).

We appreciate this reviewer's astute recommendations regarding the presentation and analysis of the F_{ST} selection scan results.

***In response to this comment, we have made the following key changes to our approach:

We shifted from using $Z-F_{ST}$ to directly plotting actual F_{ST} values, as suggested, to allow for a more biologically interpretable presentation of our data. We also updated the figure legend on Page 18, Line 490 to clarify that negative F_{ST} values are not displayed in this figure.

As suggested by this reviewer, we added the use of BayPass to account for population structure and demographic history. We also performed Bayescan. As our Bayescan results overlapped with our BayPass results, we reported only our BayPass results. We performed our enrichment analyses using windows identified as 5% outliers in F_{ST} and θ_π and also significantly associated with salinity based on BayPass. This integrated approach validated the significant enrichment of selection signatures around chromosomal fusion sites in the *E. carolleae* genome.

We added detailed descriptions of our BayPass results in the Results section on Pages 19–20, Lines 534–572 as follows:

“As an additional test for signatures of selection, we also employed the BayPass method to associate changes in allele frequency with salinity change⁵⁸, and integrated these results with the significant outliers based on F_{ST} and θ_π . Thus, we identified the top 5% of F_{ST} and θ_π outliers (10 kb windows) that were also significantly associated with salinity based on BayPass, and we considered these as signatures of selection (referred to as “BayPass + 5% F_{ST} and θ_π outliers”). As a result, we found 234 and 439 10 kb windows designated as BayPass + 5% F_{ST} and θ_π outliers in the *E. carolleae* and *E. gulfia* genomes, respectively.

We also added the methods of the BayPass analysis in the Methods section on Page 33, Lines 1012–1024 as follows:

“In addition, our analysis included BayPass v2.1⁵⁸, a powerful tool for detecting loci significantly associated with environmental variables, such as salinity. This analysis involved estimating the posterior distributions of model parameters through multiple Markov Chain Monte Carlo (MCMC) procedures. These procedures included 15 pilot runs of 500 iterations, a burn-in phase of 2,500 iterations, and 1,000 MCMC samples with a thinning interval of 25, repeated over three independent runs. This approach allowed us to robustly estimate the variance-covariance matrix for SNP frequencies and assess the correlation between SNP alleles and salinity levels at the time of sample collection (Supplementary Table 21). SNP frequencies were standardized to a mean of 0 and variance of 1⁵⁸. Bayes Factors (BFs) were calculated to quantify the strength of the association between each SNP and salinity, using the δ parameter from BayPass's auxiliary model with a default prior distribution. The genomic windows that encompassed loci with a BF greater than 10, and that also ranked within the top 5% for both F_{ST} and θ_π outliers, were identified as strong

candidates that harbored selection signatures linked to freshwater invasions (referred to as “BayPass + 5% F_{ST} and θ_{π} outliers”).”

Specific comments:

14. The use of red and green color in Figure 1 is not color-blind friendly. Adding a legend for dots and triangles in Figure 1 would assist interpretation, so readers do not have to search for the descriptive in the middle of the caption.

We appreciate the reviewer for pointing out this problem.

***In response to this comment, we have updated the color scheme of Figure 1 to be more color-blind friendly. We utilized a Color Blindness Simulator to ensure the new palette is discernible for all readers.

We also added a legend in Figure 1, clearly distinguishing dots from triangles.

15. Figure 5 is not really interpretable to readers who do not have expertise in ion uptake. It would be more relevant to compare intracellular ion control in a salinity vs. freshwater environment. Explaining how differences in the transport of specific ions leads to higher fresh vs. higher salinity tolerance would greatly improve the accessibility of this section to a larger audience. Showing the relative abundance of different ions in salinity vs. freshwater, versus within the cell, would also help readers.

We thank the reviewer for the helpful suggestions to improve the accessibility of Figure 5.

***In response to this comment, we have combined the previous Figures 5a and 5b with Figure 7a into a new, integrated Figure 5.

This revised figure enables the reader to better connect the particular ion transporter genes on the genome (Figure 5a) with the ion transporters in the model of ion uptake (Figures 5b and 5c). We believe that this new format makes the biological significance of these genes more accessible to a broader audience while maintaining focus on their relevance to genome architecture evolution.

We do not include a diagram of ion uptake under saline (brackish) *versus* freshwater environments, as at least in *E. affinis* complex, the diagrams would likely be the same, but with varying degrees of ion uptake performed by these same cells. Instead, we rewrote the legend of Figures 5b and 5c to make more explicit the process of ion uptake by the ion transporters. We hope that this description is much clearer:

“(b, c) Hypothetical models of ion uptake in freshwater habitats performed by ion transporters inside epithelial ion transporting cells (ionocytes, beige). The ion transporter colors are the same as those mapped onto the chromosomes in (a). In freshwater habitats (blue), the concentrations of ions are extremely low, such that the ions must be transported into the ionocyte (beige) against a very steep concentration gradient. (b) Model 1: *VHA* (yellow) pumps out protons (H^+) into the freshwater environment to generate a proton gradient. Then, using this proton gradient, Na^+ is transported into the cell through a Na^+ transporter (likely *NHA*, pink). *CA* (fuchsia) produces protons for *VHA*. (c) Model 2: An ammonia transporter *Rh* protein (dark green) exports NH_3 out of the cell and then this NH_3 reacts with H^+ to form NH_4^+ . This consumption of extracellular H^+ causes *NHE* (light green) to export H^+ out in exchange for the import of Na^+ into the cell. *CA* produces protons for *NHE*. In both models, *NKA* (red) transports Na^+ to the hemolymph (pink).”

16. Line 419 is the first introduction of results from “selection signatures”, but this is a broad term and it would be helpful for readers here to describe the specific signatures.

We thank the reviewer for pointing out the need to clarify this term. We do mention on Page 16, Lines 437–439 that these signatures of selection associated with salinity change were identified in both wild populations and laboratory selection lines in our prior studies (Stern and Lee 2020; Stern et al. 2022). But we now recognize that we need to make the language more explicit.

***In response to this comment, we have revised the sentence on Page 16, Lines 450–453 as follows:

“We found that the 25 ion transport-related genes with selection signatures, associated with salinity change in both wild and laboratory populations^{30,31}, were more centrally located within chromosomes of the *E. carolleae* genome than other sets of ion transport-related genes lacking such selection signatures.”

In addition, we add a more specific description of “signatures of selection” in the Introduction, on Page 5, Lines 155–159:

“Thus, our specific goals were to: (1) explore patterns of chromosomal evolution across three genomes of the *E. affinis* species complex, (2) analyze whether population genomic signatures of natural selection (**e.g., allele frequency shifts, association with salinity, LD**) in response to environmental change are associated with chromosomal fusion events, and (3) determine patterns of repositioning of loci under selection following the fusion events.”

Also, on Page 17, Lines 477–481, we more explicitly define what we mean by selection signatures (or signatures of selection):

“**Signatures of selection enriched at the chromosomal fusion sites**
Our analysis revealed that population genomic signatures of selection associated with salinity change (**SNP frequency shifts [F_{ST} and θ_{π}], association with salinity [BayPass], and LD**) were enriched (or present) at several chromosomal fusion sites, especially at the centromeres, in genomes of the *E. affinis* complex sibling species with fused chromosomes.”

17. Line 444: typo in “and all paralogs”  “all ARE paralogs” (?)

We changed this sentence on Page 12, Lines 312–314 to:

“The vertical lines and circles in various colors indicate 80 key genes that have shown evolutionary shifts in gene expression and/or signatures of selection in our previous studies, as well as all **known** paralogs and subunits from the same gene families.”

Figure 7

18. 7a - including a visual legend for the centromeres would be helpful, so readers don't need to scour the text. I may have missed it, but it would be helpful to specifically address the lack of a centromere on Chromosome 2.

We appreciate the reviewer's valuable suggestion.

***In response to this comment, we have added a visual legend for the centromeres directly on the revised Figure 5a (formerly Figure 7a) for clearer interpretation.

We have also updated the legend for Figure 5 on Page 12, Lines 315–317 to clarify the method of determination of centromere positions as follows:

“The positions of these centromeres on Chr 1, 3, and 4 were determined through the Hi-C analysis of the *E. carolleae* genome (Supplementary Fig. 4). The centromere position for Chr 2 was not identified with our methods and might be absent.”

19. 7b-d: It would be helpful to show the location of the centromeres in the Atlantic clade.

Figure 7d appears to be missing the colored ribbons? Please add for continuity.

We thank the reviewer for this suggestion.

***In response to this comment, we have added the locations of the centromeres on the *E. carolleae* chromosomes to the revised Figures 6b and 6d (formerly Figures 7b and 7d) to improve clarity. We also updated the figure legend to include the following clarification: “The circles within the *E. carolleae* chromosomes indicate the positions of the centromeres.”

We have chosen not to colorize any ribbons in Figure 6d, as doing so would not accurately reflect our assessment of the actual evolutionary history of chromosomal fusion events. Instead, this figure serves to support our hypothesis that it was highly unlikely that the *E. gulfia* chromosomes (7 chromosomes) fused to achieve the genome architecture of the Atlantic clade genome (4 chromosomes), given that many inversions and translocations are required to align these genomes. We regard the current visual representation as instructive in supporting that these fusions were unlikely events.

20. With this figure, it’s confusing that multiple colors of ion transporting genes are shown in 7a, but only a subset of those are shown in b-d. It feels a bit like the authors are cherry-picking genes that show localization to the centromeres.

We thank the reviewer for this comment. To clarify, our intention in Figure 7a (now Figure 5a) was to provide a comprehensive overview of all identified ion transport-related gene families across the genome, using distinct colors to identify key gene families.

Meanwhile, Figures 7b–d (now Figures 6b–d) were designed to illustrate a few key fusion events as representative examples. That is, this figure shows how ion transport-related genes come together following chromosomal fusion events.

***In response to this comment, we have added a new sentence in the legend of Figure 6, on Page 14, Lines 374–375 as follows:

“Also shown are a few examples of ion transport-related genes aggregating following chromosomal fusion events.”

Figure 8

21. Figure 8 has a lot going on, and it would be helpful for readers to include a visual legend for the centromeres, fusion points, terminology (e.g. what is “C1S1” etc), and green points. I could not see the dark blue dashed line in the figure. There are no “vertical dashed lines”, do you mean vertical grey bars? It would be helpful to also see where the fusion sites are in Figure 7, which would provide readers with some reference points between the two figures.

Plotting F-st in Figure 8a-d would be more interpretable than Z-FST.

We thank the reviewer for these useful suggestions to improve the clarity and interpretability of Figure 7 (formerly Figure 8).

***In response to this comment, we have made several enhancements in the revised Figure 7 to increase the clarity of this figure:

We added a detailed visual legend, which now defines the color dots, centromeres, and fusion sites. We have also adjusted the color of the dashed lines in Figures 7a–d to increase their visibility.

We now state explicitly in the legend for Figures 7e and 7f (histograms) that the vertical dashed lines indicate for each chromosomal fusion site the number of windows with signatures of selection (on the x-axis), as follows:

“Vertical dotted lines indicate the number of 10 kb windows under selection (based on F_{ST} and θ_{π}) within 1 Mb of the chromosomal fusion sites.”

We now mark the fusion sites in the revised Figure 5a (formerly Figure 7a) as suggested.

We shifted from using $Z-F_{ST}$ to directly plotting actual F_{ST} values in Figures 7a–d.

Discussion

22. The discussion is almost entirely focused on the study species, which greatly limits its appeal to a broader audience. Addition some literature about supergene evolution and salinity adaptation in other species would greatly improve its appeal.

Compared to the rest of the paper, the discussion is not as well organized and has some repetitive text. Some revisions and reorganization here would greatly improve the manuscript. It would also be helpful at the beginning of the discussion to remind the readers about the ecology and history of the three clades, rather than lower down. I thought the discussion would flow better if it was organized in the opposite manner that describes the temporal evolution of the system: starting with “divergence and chromosomal evolution”, then “central repositioning” of ion transporter genes, then “impacts of the new genomic architecture” on selection response. The section on “enrichment of selection signatures” was largely repetitive of the results, and could be reduced and integrated into the other sections.

We thank the reviewer for the constructive suggestions regarding the structure and broader framing of the Discussion.

***In response to this comment, we broadened the appeal of the Discussion by emphasizing the broader context of this study. For instance, on Pages 21–22, Lines 607–613, we state:

“Chromosomal fusions have long been hypothesized to facilitate adaptation, but empirical support had been scarce and indirect^{5,6,13,17}. Theoretical studies had hypothesized that genomic rearrangements such as chromosomal fusions or inversions could enhance adaptive potential by altering the recombination landscape and repositioning functionally interacting genes into closer proximity^{6,10}. While several empirical studies have found that chromosomal inversions could reposition adaptive loci and reduce recombination among beneficial alleles, thereby altering how populations respond to selection^{12,14,15,59,60}, none have linked chromosomal fusions with population responses to selection.”

We also discuss the broader context on Page 24, Lines 687–693:

“Our results complement previous work indicating that chromosomal inversions serve as a key mechanism for supergene formation^{12,14,59,60}. Previous examples of supergenes, such as those governing mimicry in butterflies or social behavior in ants and birds^{7,8,11,12}, are typically facilitated by inversions that lock together multiple loci that are already on the same chromosome. In contrast, our study reveals that chromosomal fusions can form novel linkages among adaptive genes by joining previously unlinked chromosomes. This mechanism can profoundly alter the recombination landscape of the genome, generating new low-recombination regions that harbor coadapted allelic combinations.”

In addition, we greatly reorganized the Discussion section by consolidating the subsections by topic, rather than by results. Thus, we removed much of the redundancy. For example, we originally had separate subsections within the Discussion on the ‘enrichment of selection signatures at the fusion sites’ and on the ‘repositioning of key ion transporter genes,’ such that we

repeated several discussion points multiple times. Now the Discussion section is far more streamlined.

We now describe how the aggregation of ion transporter genes might be associated with the ecology and invasiveness of different clades (sibling species), with the more invasiveness clades having greater aggregation of ion transporter genes. For example, the Atlantic clade *E. carolleae* (4 chromosomes), which has the greatest number of fusions and ion transporter aggregations, is clearly the most invasive and adaptable of all the clades.

23. Sections of the “central repositioning” are based on unpublished research - the authors should check with the journal guidelines and make sure that In Review and In Prep citations can be included (typically they cannot). Regardless, this section could be improved by bringing in some literature on coordinated regulation of gene expression and GWAS in other species.

We thank the reviewer for this important comment. In response, we have removed all references to unpublished research and extensively reorganized the Discussion section.

24. While much of the message is focused on what is happening at the centromeres in the Atlantic clade, the Gulf clade also adapts to salinity and does not have such fusions. Thus, chromosomal fusions are not a prerequisite for salinity adaptation and this point feels lost in the discussion.

We appreciate the reviewer’s thoughtful point.

In this revised manuscript, we now add results using BayPass, which together with F_{ST} and θ_{π} , did reveal significant enrichment of selection signatures at several fusion sites in the Gulf clade *E. gulfia* genome.

***In response to this comment, we have added these new results in the Results section on Pages 20–21, Lines 573–585 as follows:

“Similarly, chromosomal fusion sites of the *E. gulfia* genome also showed significant enrichment of selection signatures at three out of eight fusion sites (Supplementary Figs. 12–17), though the signals were less pronounced than for the *E. carolleae* genome. Specifically, when we examined all fusion sites relative to the genomic background for *E. gulfia*, we found significant enrichment of selection signatures at fusion sites based on BayPass + 5% F_{ST} and θ_{π} outliers within 1 Mb intervals (KS test, $P = 9.22e-4$). Using 2 Mb intervals, significant enrichment was evident based on multiple measures, including top 5% F_{ST} and θ_{π} outliers (KS test, $P = 0.0401$), top 1% F_{ST} and θ_{π} outliers (KS test, $P = 0.0280$), and BayPass + 5% F_{ST} and θ_{π} outliers (KS test, $P = 0.0226$) (Supplementary Tables 16 and 17). Three out of eight fusion sites, one each on Chromosomes 1, 5, and 7, exhibited significantly higher numbers of genomic windows with signatures of selection, based on BayPass + 5% F_{ST} and θ_{π} outliers, relative to the genomic background (Supplementary Fig. 17). One of these fusion sites, on Chromosome 5, exhibited more robust enrichment of selection signatures, as this fusion site was also enriched for genomic windows that were top 1% F_{ST} and θ_{π} outliers (Supplementary Figs. 14, 16, and 17).”

We now have revised our manuscript to state that we have found genomic signatures of selection at the chromosomal fusion sites in both sibling species (Atlantic and Gulf clades) of the *E. affinis* complex. While *E. carolleae* (Atlantic clade) displayed more prominent and statistically robust enrichment of selection signatures, particularly around clearly defined centromeres, *E. gulfia* (Gulf clade) also showed significant enrichment of selection signatures at several fusion sites.

25. Line 673- “beneficial reversal of dominance” comes out of nowhere - either this needs some more context or could be removed.

We have added more context regarding “beneficial reversal of dominance” on Page 25, Lines 729–733 as follows:

“Under such conditions, dominance would switch between environments, such that freshwater tolerance is dominant under freshwater conditions, whereas saltwater tolerance is dominant under saltwater conditions. This beneficial reversal of dominance potentially arises from the more fit allele in a heterozygote compensating for the lower function of the less fit allele^{66,67}.”

Methods

26. How were the RNA libraries prepared? It read as if the analysis was based on total RNA as opposed to mRNA.

We appreciate the reviewer’s attention to detail regarding the RNA library preparation.

***In response to this comment, we have added a new sentence on Page 28, Lines 850–853 to clarify our RNA libraries preparation process based on mRNA as follows:

“Subsequently, messenger RNA (mRNA) was enriched by oligo(dT) magnetic bead capture, with other RNA types being excluded. This procedure was followed by mRNA fragmentation and cDNA reverse transcription prior to RNA library preparation. The resulting cDNA samples were processed into libraries for sequencing.”

27. Lines 868-869: “annotated using a comprehensive approach...” was vague. Please add detail or a reference.

We have revised this sentence on Page 31, Lines 928–930 as follows:

“These genome assemblies of other copepod and daphnid species were annotated using a comparable approach to our study (see Section on Genome annotation above), which incorporates transcriptome data, *ab initio* predictions, and homology to known proteins.”

28. “Population genomic analysis” section was missing the type of sequencing (presumably Illumina) and the average coverage per pool.

We have added the information regarding sequencing type and average coverage on Page 32, Lines 983–986 as follows:

“To explore the association between chromosomal fusions, and the formation of novel gene linkages, with signatures of selection in *E. affinis* species complex populations, Illumina-seq population genomic data collected previously from wild populations of *E. carolleae* and *E. gulfia*³¹ were reexamined. These data had sequence coverage ranging from 16× to 30×, with a mean coverage of 25× (Supplementary Table 21).”

Overall impression

I enjoyed reviewing this paper and commend the authors on a nearly thorough job. With an LD analysis, a more rigorous and interpretable FST outlier analysis, and some revisions, it will be a strong contribution to the literature.

Again, we thank the reviewer for all their thoughtful and insightful comments, which has greatly improved the quality and rigor of our manuscript.

Reviewer #3 (Remarks to the Author):

In the present manuscript, Du et al generate and analyze two new chromosome-level genome assemblies from a copepod species complex, then analyze these in tandem with a prior assembly to study the role of natural selection on genome architecture in this clade. The authors argue that selection-driven chromosomal fusions can explain their prior observation that certain ion transporter genes, which show

signatures of contemporary selection, are clustered within the genome. Consequently, they propose that fusion-mediated supergene formation provides an explanation for rapid evolution in response to environmental change, and they argue that their results provide conclusive evidence for this mechanism not found in prior studies.

In general, the empirical work appears to be very thorough and well thought out, the manuscript is quite well written, the figures are useful and mostly well designed, and the authors provide a variety of interesting observations. The inference that the three clades under consideration actually represent evolutionarily ancient species is compelling and intriguing. We found the reconstruction of karyotype evolution and the depiction of how chromosomes have fused to be a particular highlight of the study. Finally, the fact that genes near fusion sites appear to be under selection in only one of two sibling species that experienced such fusions is thought provoking.

We greatly appreciate the recognition of the rigor of our empirical work, the clarity of the writing and figures, and the importance of our key findings.

Incidentally, we have included an additional analysis of selection signatures, BayPass, which in conjunction with F_{ST} and θ_π outlier analyses, does reveal signatures of selection enriched at the fusion sites in both the Atlantic clade *E. carolleae* and Gulf clade *E. gulfia* genomes.

Nonetheless, we find ourselves unconvinced by several of the central conclusions of the paper. We are sympathetic to the authors' belief that the weight of the evidence supports a "supergene formation" hypothesis, but we feel that this remains a hypothesis and that the basic claims made by the authors cannot be confirmed based on the current data and analyses. We cannot recommend publication of the manuscript in its present form but suggest that a different but equally consequential framing of the results can be adopted to make the paper more suitable for publication in a journal of the standing of *Nature Communications*. However, all of our recommendations would require a radical alteration to the framing and the interpretation of the data (e.g., focusing on the morphological stasis or providing a more descriptive study of karyotype evolution). In the spirit of reviewing the paper as intended, we do not provide detailed suggestions for how this might be done, but instead comment on whether or not the central conclusions in the current report are supported by the data and, if not, what new analyses might address this.

Our four major comments are as follows:

29. By all accounts the fusion of chromosomes was an ancient event, so what is the justification for using genetic data from contemporary populations (ca. 65-80 years diverged) to infer whether the chromosomal fusions were maintained due to selection?

The hypothesis that ancient genomic rearrangements could impact selection responses of contemporary populations is a longstanding hypothesis generated by many theoretical studies (Yeaman 2013; Guerrero and Kirkpatrick 2014). The most concrete support has emerged from the studies on chromosomal inversions, where the changes in positions of adaptive loci due to the inversions altered the selection responses of contemporary populations (Lowry and Willis 2010; Joron et al. 2011; Schwander et al. 2014; Thompson and Jiggins 2014). A notable example is the inversions that promoted temperature adaptation in populations of *Drosophila* (Hoffmann and Rieseberg 2008). Our study is notable in being the first to examine the impacts of chromosomal fusions on population responses to selection.

Our own interest in comparing the ancient and contemporary patterns of ion transporter evolution arose from results of our other studies. For instance, our molecular evolution studies on the ion transporter gene families *NHA*, *NKA*, and *NKCC*, now being prepared for publication, revealed that the paralogs under selection on longer time scales are also under selection (show allele frequency shifts) during contemporary saline to freshwater invasions in *E. affinis* complex

populations. In many cases, the same amino acid position is under selection across the different timescales. This result led us to surmise that certain ion transporter sites are more “evolvable” and prone to respond to habitat change. Thus, the genomic region with a long history of evolutionary change appears to be the same region that responds to contemporary environmental change.

In addition, these ion transporters need to cooperate to perform the action of ion uptake from the environment (Lee et al. 2022) and are coregulated (Posavi et al. 2020). Thus, being localized together on the genome would impede recombination and facilitate coinheritance of coadapted alleles, as well as facilitate coregulation.

The long evolutionary history of salinity fluctuations in the genus *Eurytemora* might be linked to the incredible ability of some members of this genus to invade across salinities. The genus *Eurytemora* is quite remarkable in occupying the widest range of salinities among all copepods known. This genus has an evolutionary history of salinity fluctuations along the coasts of the subarctic region. For certain members of this genus, the high degree of invasiveness across salinities is comparable with notable saline-to-freshwater invaders, such as zebra mussels, quagga mussels and the fishhook waterflea *Cercopagis*.

More generally, the paper is framed by, and uses methods derived from, a population genomics perspective to make claims about the deep past (at least tens of millions of years) when it is not clear that the basic assumptions of these methods are met. For instance, why would one expect expansions in ion transporter gene copy number to underpin colonization of different salinity environments when the timescales involved are sufficient for natural selection to elevate the frequency of rare regulatory or amino acid mutations that improve the functions of ancestral genes? This is not to say that changes in copy number are irrelevant, just that the assumption that they must be under selection (rather than neutral) does not seem warranted. Therefore, the central idea from the title and abstract – that supergenes can facilitate rapid adaptation to novel environments – is questionable in light of the results presented.

The reviewer raises intriguing questions regarding the premise of this study.

We are not stating that the gene family expansions themselves underpin the recent colonization of different salinity environments, but that the contemporary salinity colonization are associated with allele frequency shifts (natural selection) at these multiple paralogs for many ion transporter gene families. Our measures of F_{ST} and θ_{π} outliers are indeed measuring allele frequency shifts between the contemporary saline and freshwater populations. And these allele frequency shifts are taking place at the expanded ion transporter gene families, especially at the fusion sites.

We determined that a multitude of ion transporter paralogs are under selection (undergo allele frequency shifts) based on many prior studies on this system. We found population genomic signatures of selection (allele frequency shifts) across independent saline to freshwater invasions, with the multiple ion transporter paralogs within gene families showing the strongest signatures of selection (this study and Stern and Lee, 2020). We found the same results in our laboratory evolution experiments, with multiple ion transporter paralogs within gene families showing parallel selection across replicate selection lines (Stern et al. 2022). Thus, sets of ion transporter paralogs are under selection during salinity change.

In addition, our prior results (cited in this study) indicate that different ion transporter paralogs are functionally differentiated and that multiple paralogs within gene families contribute to salinity adaptation. Our physiological studies indicate that the paralogs are expressed in different tissues and show differences in patterns of gene expression across salinities. For instance, *NHA-7* is expressed in the gut, whereas *NHA-5* is expressed in the swimming legs, which operate like gills. Other paralogs are expressed in the maxillary glands, which function like kidneys.

Consequently, multiple ion transporter paralogs appear to be essential for salinity adaptation. Therefore, it does appear that natural selection at the expanded ion transporter gene families might have played an important role in promoting the invasiveness in certain clades of the *E. affinis* complex.

***In response to this comment, we now add in the Discussion section text regarding the potentially essential roles of the multitude of ion transporter paralogs in salinity adaptation on Page 23, Lines 668–676, as follows:

“Notably, the multiple paralogs within ion transport-related gene families appear to collectively contribute to salinity adaptation and undergo natural selection (allele frequency shifts) during saline to freshwater invasions. The different ion transport-related paralogs appear to show functional differentiation, as they are expressed in different osmoregulatory tissues (e.g., maxillary glands, swimming legs, or digestive tract) and show differences in gene expression patterns^{32,61-63}. In addition, the multiple ion transport-related paralogs appear to be nonredundant and essential for salinity adaptation, given that sets of paralogs are repeatedly under selection in wild populations and laboratory selection lines^{30,31}. Therefore, it does appear that the ion transport-related gene family expansions might have played an important role in promoting the invasiveness in certain clades of the *E. affinis* complex.”

30. There are a few issues regarding supergenes. First, on a semantic level, it appears that the authors are discussing fixed differences between species, not intraspecific polymorphisms, so ‘supergene’ probably is not the appropriate term. Regardless, the defining features of supergenes are: a) Mendelian inheritance of haplotypes underlying complex traits, and b) suppression of recombination linking multiple non-neutral alleles that presumably confer the traits. As far as we are aware, these features were not examined in this study (proximity to the centromere is not evidence of suppressed recombination). If the loci under consideration are not supergenes in the sense we describe above (but rather, for instance, are hotspots of adaptive evolution), that would severely undermine the authors’ key conclusions as it would not suggest any specific mechanism allowing these loci in particular to facilitate rapid evolution.

We thank the reviewer for this thoughtful and important comment on the usage of the term “supergene.” We respond to three of the reviewer’s comments below.

First of all, the clusters of beneficial alleles at the fusion sites (ion transporter and other selected alleles) do not violate the first defining feature of supergenes. The ion transporter and other selected alleles at the fusion sites do not show fixed differences between the sibling species or between populations within species. There are high levels of allelic variation at these loci in the populations from each of the clades. In fact, as we have shown in our previous studies, many of the alleles (SNPs) favored by selection during saline to freshwater invasions across distinct clades also show signatures of balancing selection in the native range populations (Stern and Lee 2020). That is, many of the same SNPs are under balancing selection in saline populations across different clades (Stern and Lee 2020).

In response to the reviewer’s comment, we make more explicit throughout the paper that the signatures of selection between the saline and freshwater populations represent allele frequency shifts between the populations, indicating that the alleles are not fixed within or between populations. We refer to the allele frequency shifts between saline and freshwater populations at selected loci on Page 15, Line 416; Page 16, Lines 418–422, Lines 437–439; Page 17, Line 479; Page 19, Lines 511, 519, and 521.

Second, we really appreciate the reviewer’s extremely wise advice to explore signatures of LD at the chromosomal fusion sites.

***In response to this comment, we have now included new analyses to determine LD patterns at fusion sites based on data from whole-genome sequencing of 14 individual copepods. We used

genome sequences of individual *E. carolleae* collected in 2022 from Baie de L'Isle Verte, St. Lawrence Estuary, QC. We performed individual sequencing with coverage ranging from 30× to 46×, with a mean coverage of 35×. Using PopLDdecay, we calculated LD (r^2) across genomic regions, with a focus on comparing LD at the fusion sites versus genome-wide background regions.

Our analysis revealed that all fusion sites exhibited significantly higher LD levels relative to the genome-wide background (Supplementary Figure 18). In addition, the fusion sites that contained significant enrichment of contemporary selection signatures showed even higher LD relative to all fusion sites. These results provide robust support for elevated LD at the fusion sites, especially the fusion sites with signatures of selection, and provide further support for our “supergene” hypothesis.

We added these new results on the LD analysis in the Results section on Page 21, Lines 586–599 as follows:

“Regarding patterns of recombination across the genome, the *E. carolleae* genome showed evidence of reduced recombination at the fusion sites compared to the genome-wide background (Supplementary Fig. 18). We investigated patterns of linkage disequilibrium (LD) across the genome, using whole-genome sequences of 14 individual copepods, to evaluate whether fusion sites exhibit signatures of reduced recombination relative to genome-wide levels. This analysis was motivated by the hypothesis that chromosomal fusions are clustering co-adapted loci and alleles into regions of suppressed recombination. This clustering of co-adapted alleles would then facilitate their joint inheritance, with the potential to promote adaptive evolution.

Notably, the fusion sites of the *E. carolleae* genome did indeed exhibit significantly elevated LD in both 1 Mb (Paired t -test; $t = 264.4$, $df = 299,980$, $P < 0.0001$) and 2 Mb genomic intervals (Paired t -test; $t = 275.1$, $df = 299,980$, $P < 0.0001$). Fusion sites with significant enrichment of selection signatures demonstrated even higher LD relative to all fusion sites (Supplementary Fig. 18). Recombination patterns were not explicitly analyzed for *E. gulfia* in this study due to the lack of genome sequences for individual copepods.”

We also added new methods on LD analysis in the Methods section on Page 34, Lines 1047–1059 as follows:

“To assess whether chromosomal fusion sites exhibited reduced recombination and elevated linkage disequilibrium (LD), whole genome sequencing was performed for an additional 14 individuals of *E. carolleae*. These copepods were collected in 2022 from Baie de L'Isle Verte, St. Lawrence Estuary, QC, Canada and frozen at -80°C . DNA extracted from each individual copepod was amplified to satisfy sequencing requirements using the REPLI-g Single Cell Kit (Qiagen, Hilden, Germany). These amplified DNA samples were then sequenced on the NovaSeq 6000 platform, achieving coverage ranging from 30× to 46×, with a mean coverage of 35×. Data processing and SNP calling procedures were consistent with those described above. PopLDdecay¹³⁰ was utilized to calculate LD (r^2) and to generate patterns of LD decay across both fusion sites and background genomic regions. Statistical analyses were then performed to determine whether the chromosomal fusion sites exhibited elevated LD relative to the genomic background. To directly compare LD across equivalent genomic distances, a pairwise comparison of mean r^2 values was performed between 1 Mb (and 2 Mb) genomic regions encompassing fusion sites versus the genome-wide background at shared distance bins (20 bp–300 kb) using Paired t -tests in R.”

Third, in response to the reviewer's comment, we now temper our discussion on supergenes, and use this term much more sparingly in this paper. For instance, we avoid referring to ion transporter gene clusters as definitive supergenes. Instead, we refer to these clusters more generally as “linked adaptive loci” or “coadapted gene complexes” when appropriate. While we refrain from concluding that these represent true supergenes, we now clarify on Pages 23–24, Lines 677–686 that the observed chromosomal fusions bring together ion transporter loci into

low-recombination regions, consistent with theoretical pathways to supergene formation, as follows:

“The large clusters of ion transport-related genes brought together in the *E. carolleeae* genome might constitute “supergenes” underlying salinity tolerance. Supergenes are distinct from “genomic islands of divergence,” in forming tightly linked sets of co-functional loci maintained by suppressed recombination. The clusters of ion transport-related genes found in this study constitute not only islands of high differentiation, but also potential coadapted gene complexes that could be inherited and evolve together as a unit. Furthermore, our evidence of high LD and functional relatedness among the fused loci in *E. carolleeae* are consistent with these fusions forming supergenes. The findings of this study suggest that chromosomal fusions might serve as an important mechanism for generating supergenes. Such a process of supergene formation through chromosomal fusions has been hypothesized in theory^{5,10} but rarely documented empirically in the past.”

31. Beyond the concerns addressed above, it is difficult to understand why one particular mechanism (selection) is privileged to explain the observation that chromosomal fusions have been maintained for tens of millions of years and ion channel genes have become clustered. Are hypothetical signatures of selection in contemporary populations (e.g., elevated F_{ST} and depressed nucleotide diversity) really expected in ancient macromutations involving fundamental changes in chromosomal architecture? The issue of recombination seems especially relevant, given that fusion points now lie in proximity to centromeres, and the fact that the correspondence between chromosome fusion and “contemporary selection” was observed in only one species seems particularly problematic. Is it possible that this species is just an outlier in its recombination mechanisms or rates, for instance, leading to patterns of divergence and diversity that spuriously resemble the outcome of selection?

The reviewer’s question on whether the Atlantic clade (*E. carolleeae*) is an outlier motivated us to perform additional analysis. We appreciate that the reviewer’s comments compelled us to conduct more thorough analyses.

The lack of observed selection signatures at the fusion sites for the Gulf clade *E. gulfia* was likely due to a power issue, so we performed additional analyses. When we performed BayPass analyses, together with our F_{ST} and θ_{π} analyses, we did find the enrichment at the fusion sites of SNPs with selection signatures and association with salinity for both *E. carolleeae* and *E. gulfia*.

We have added these new results in the Results section on Pages 20–21, Lines 573–585 as follows:

“Similarly, chromosomal fusion sites of the *E. gulfia* genome also showed significant enrichment of selection signatures at three out of eight fusion sites (Supplementary Figs. 12–17), though the signals were less pronounced than for the *E. carolleeae* genome. Specifically, when we examined all fusion sites relative to the genomic background for *E. gulfia*, we found significant enrichment of selection signatures at fusion sites based on BayPass + 5% F_{ST} and θ_{π} outliers within 1 Mb intervals (KS test, $P = 9.22e-4$). Using 2 Mb intervals, significant enrichment was evident based on multiple measures, including top 5% F_{ST} and θ_{π} outliers (KS test, $P = 0.0401$), top 1% F_{ST} and θ_{π} outliers (KS test, $P = 0.0280$), and BayPass + 5% F_{ST} and θ_{π} outliers (KS test, $P = 0.0226$) (Supplementary Tables 16 and 17). Three out of eight fusion sites, one each on Chromosomes 1, 5, and 7, exhibited significantly higher numbers of genomic windows with signatures of selection, based on BayPass + 5% F_{ST} and θ_{π} outliers, relative to the genomic background (Supplementary Fig. 17). One of these fusion sites, on Chromosome 5, exhibited more robust enrichment of selection signatures, as this fusion site was also enriched for genomic windows that were top 1% F_{ST} and θ_{π} outliers (Supplementary Figs. 14, 16, and 17).”

We have now revised our manuscript to state that we have found genomic signatures of selection at the chromosomal fusion sites in both sibling species (Atlantic and Gulf clades) of the *E. affinis* complex. While *E. carolleeae* (Atlantic clade) displayed more prominent and statistically robust

enrichment of selection signatures, particularly around clearly defined centromeres, *E. gulfia* (Gulf clade) also showed significant enrichment of selection signatures at several fusion sites.

We appreciate the question regarding the relationship between ancient chromosomal fusions and contemporary signatures of selection.

Based on our prior and current results, we do find a meaningful connection between the ancient fusions and contemporary signatures of selection (e.g., elevated F_{ST} and reduced nucleotide diversity, and association with salinity using BayPass). In our previous studies, the ion transporters that are clustered at the chromosomal fusion sites are repeatedly under selection during independent saline to freshwater populations in wild populations from both the Atlantic and Gulf clades (Stern and Lee 2020). These same ion transporters show coordinated evolutionary shifts in gene expression (Posavi et al. 2020). In addition, these ion transporters show signatures of positive epistasis during laboratory selection experiments, suggesting that they are under selection as a group (Du et al. In Prep; Stern et al. 2022). Moreover, these ion transporters are significantly associated with salinity survival, with freshwater alleles associated with freshwater survival (Fraitout et al. In Prep.).

There is increasing evidence that genomic architecture can have profound impacts on population responses to natural selection. Changes in the genomic positions of beneficial alleles at different loci can have profound impacts on how natural selection acts on these sites. For instance, numerous studies on chromosomal inversions have found that these ancient genomic rearrangements have impacts on contemporary evolution (Hoffmann and Rieseberg 2008; Lowry and Willis 2010; Joron et al. 2011; Schwander et al. 2014; Thompson and Jiggins 2014). Our study finds this type of pattern on a larger scale, at chromosomal fusion sites across multiple chromosomes. While the chromosomal fusions are ancient, the resulting genome architecture can have profound impacts on how populations respond to selection.

Furthermore, our LD analyses support that recombination is suppressed around these chromosomal fusion sites, creating genomic features where linked selection could produce the observed selection signatures.

Specific comments:

32. Lines 49-51: With such deep time involved in the chromosomal fusions, can we really say they facilitate adaptation? The same question pertains to the subsequent statement (rapid colonization of novel habitats).

Genomic features generated millions of years ago could indeed have profound impacts on contemporary events, as genome architecture (in terms of the arrangement of genes in the genome) could have large impacts on how populations respond to natural selection. In fact, in recent years, several authors have hypothesized that genomic rearrangements (such as chromosomal fusions or inversions) could alter how populations respond to selection and impact rates of adaptation (Hoffmann and Rieseberg 2008; Lowry and Willis 2010; Joron et al. 2011; Yeaman 2013; Schwander et al. 2014; Thompson and Jiggins 2014). These authors have pointed out the importance of repositioning interacting loci as an important factor in promoting adaptation. This study provides important results toward addressing these hypotheses.

While the chromosomal fusions are ancient, selection can now act on combinations of alleles at the loci that are now close together in some clades. In fact, in some of the saline populations, balancing selection appears to maintain genomes that contain either clusters of saltwater or freshwater adapted alleles.

In the case of this study, genome architecture evolution has brought together functionally interacting loci. As clarified in our response to Comment 31, our intention was not to imply that the fusions immediately facilitated adaptation at the time they arose. Rather, these ancient chromosomal fusions reorganized genome architecture by establishing novel linkage relationships and altering recombination landscapes, which may have been subsequently co-opted during more recent adaptive responses, such as during the rapid colonization of freshwater environments by populations of *E. carolleae* (Atlantic clade) and *E. gulfia* (Gulf clade).

***In response to this comment, we have added the statement in the Discussion section on Pages 21–22, Lines 607–619 as follows:

“Chromosomal fusions have long been hypothesized to facilitate adaptation, but empirical support had been scarce and indirect^{5,6,13,17}. Theoretical studies had hypothesized that genomic rearrangements such as chromosomal fusions or inversions could enhance adaptive potential by altering the recombination landscape and repositioning functionally interacting genes into closer proximity^{6,10}. While several empirical studies have found that chromosomal inversions could reposition adaptive loci and reduce recombination among beneficial alleles, thereby altering how populations respond to selection^{12,14,15,59,60}, none have linked chromosomal fusions with population responses to selection.

In our study, ancient chromosomal fusions repositioned key ion transporter genes into central regions of reduced recombination in certain clades (sibling species) of the *E. affinis* complex. While these chromosomal fusions were ancient, these fusion sites exhibited population genomic signatures of selection (see next paragraph), suggesting that selection in contemporary populations could act upon the resulting pre-existing combinations of adaptive alleles. As such, genome architecture, particularly chromosomal fusions, could have profound impacts on population responses to natural selection.”

33. Lines 73-74: Are “genomic islands” being equated with “supergenes” here? In our view, the concepts differ fundamentally; a supergene exists as an intraspecific polymorphism featuring little or no recombination that enables Mendelian inheritance of a polygenic phenotype. Genomic islands, on the other hand, typically are conceived of as nearly-fixed differences between partially reproductively isolated populations or species that are associated with minimal introgression at a block of the genome (probably a better framework for the topic of this paper). Supergenes generally are sparsely distributed throughout the genome (commonly even singular entities), while genomic islands are expected to occur widely throughout diverging genomes. Supergenes may give rise to genomic islands, but other mechanisms can also do so, and the evolutionary interpretation of the observation will differ based on which mechanism is implicated.

We thank the reviewer for this thoughtful and important clarification. We fully agree that genomic islands of divergence and supergenes represent distinct biological phenomena with different evolutionary contexts. In our manuscript, our intent was not to conflate the two, but rather to note that both may arise following certain chromosomal structural changes (e.g., inversions and chromosomal fusions), albeit with different functional outcomes and population genetic signatures.

***In response to this comment, we have extensively revised the relevant sentences to improve clarity and avoid conceptual ambiguity. These changes can be found on Page 3, Lines 70–77 as follows:

“An increasing body of research indicates that the distribution of adaptive loci within genomes are non-random, often clustering in specific genomic regions. Theoretical simulations support this hypothesis, suggesting that the clustering of adaptive loci is more likely explained by genomic rearrangements than by the chance establishment of new mutations near existing adaptive loci⁶. Some of these clusters might form “genomic islands” of differentiation, whereas others might constitute “supergenes,” which are tightly linked loci that function together to control complex traits⁵⁻¹². Despite this emerging recognition, the evolutionary origins and mechanisms leading to

the formation of such genomic features remain largely enigmatic, especially at a genome-wide scale.”

34. Minor comment: Lines 97-102 are a bit superfluous and might be better in the Discussion. If we understood correctly, the authors are suggesting that the results of this paper are more conclusive than prior findings (e.g., Reference 5), but we are not convinced that this is the case.

We have removed these sentences to avoid any potential misunderstanding or overstatement.

35. Line 111: Members of the *E. affinis* are complex grazers of what?

We have clarified on Page 4, Line 107 that members of the *E. affinis* complex are “grazers of algae.”

36. Line 114: Are the “large genetic divergences” among regional populations referred to unique to the mtDNA or does this apply to the nuclear genome as well? It is clear from what comes later that there is indeed very deep nuclear DNA divergence as well. It is difficult to reconcile such divergence with the absence of complete reproductive isolation between some of the presumed cryptic species in this complex (discussed further in lines 707-715). In any case, the authors should provide a scale bar for the amount of sequence divergence in Fig. 1b.

We thank the reviewer for this thoughtful comment. The divergence referenced on Line 114 primarily pertains to mitochondrial gene sequences reported in previous studies. In this study we report on Page 15, Lines 403–409 the nuclear genome-wide divergences, which are also substantial, likely reflecting long-term allopatric isolation among the three cryptic species.

Regarding the expectation of complete reproductive isolation between genetically divergent sibling species, incomplete reproductive isolation among cryptic copepod species, including those differing in chromosome number, is not uncommon. For example, in the copepod *Acanthocyclops vernalis* species complex, Grishanin et al. (2006) reported that crosses between cryptic species with distinct karyotypes ($2n = 8$ and $2n = 10$) produced viable and fertile hybrid offspring that persisted for at least 60 generations. The copepod *Tigriopus californicus* has extremely large mtDNA divergences among populations along the California coast, some larger than the divergences found in this study, yet the populations can intermate and produce viable offspring.

***In response to this comment, we now cite this example of *Acanthocyclops vernalis* in the Discussion section.

In addition, we have revised this sentence on Page 4, Lines 110–112 as follows:

“large genetic divergences in mitochondrial gene sequences separate at least six geographically distinct sibling species (clades) within this species complex (Fig. 1b), with idiosyncratic patterns of reproductive isolation among the clades²²⁻²⁴.”

We also added a scale bar to Figure 1b to clearly indicate the sequence divergence.

37. Lines 171-178, 246-274: The introduction of microevolution here for the first time is confusing given the ~65my timescale relevant to the chromosomal fusion events. Characterization of the *E. affinis* complex as comprising distinct “sibling species” helps mitigate some of this confusion and so should be made clearer earlier in the paper and emphasized throughout. We recommend that these groups consistently be referred to as species, not as clades, and that the authors amend the Abstract and Introduction to immediately make the deep evolutionary timescale clear to the reader.

We appreciate the reviewer’s suggestion to clarify the terminology and evolutionary context of the *E. affinis* complex early in the manuscript.

***In response to this comment, we now consistently refer to the three clades in *E. affinis* complex as “sibling species” to reflect their distinct sibling species status throughout the manuscript. We retained the term “clade” in specific sections where it is necessary to discuss other groups within the *E. affinis* complex that have not yet been formally described as sibling species or thoroughly studied.

38. Lines 281-283: The scarcity of genomic data from taxa or populations of varying evolutionary distance from the focal taxa is an important limitation for any claims that genomic elements have been lost or gained in the ancestor of the *E. affinis* complex and represents, of course, a common problem in reconstructing evolutionary transformation series.

We acknowledge and appreciate the reviewer’s point regarding the limitations imposed by the scarcity of genomic data from closely or distantly related taxa. This indeed is a common challenge in evolutionary genomics, particularly when attempting to determine whether genomic elements have been uniquely lost or gained within a specific lineage, such as the *E. affinis* complex.

***In response to this comment, we have added a new sentence to the manuscript on Page 11, Lines 281–284 as follows:

“Notably, the identification of unique and lost gene families in the *E. affinis* complex genomes might be influenced by limited sampling of other calanoid copepod genomes. As such, assessment of gene gains and losses are tentative and dependent on the availability and analysis of genomic data from additional taxa.”

39. Line 347: This might be the place to mention that the nuclear genome-based phylogeny agrees well with the mtDNA network rooted with the outgroups and displayed as a tree.

We appreciate the suggestion to emphasize the concordance between our nuclear genome-based phylogeny and the mtDNA-based analyses.

***In response to this comment, we have added a new sentence on Page 13, Lines 355–356 as follows:

“This phylogeny, based on 1153 of single-copy ortholog genes, was highly concordant with the mitochondrial phylogeny of *E. affinis* complex (Fig. 1b).”

40. Lines 348-349; 354: Is 15 or 21 chromosomes reconstructed as the ancestral karyotype among the *E.affinis* complex species?

The 15-chromosome genome observed in *E. affinis* (Figure 2f) likely represents the ancestral karyotype for *E. carolleae* and *E. gulfia*. This inference is supported by reconstructions using the ancestral chromosome reconstruction tools Agora and ANGES. For the entire *E. affinis* complex, a 21-chromosome karyotype is reconstructed as the ancestral state.

***In response to this comment, we have clearly revised the sentences on Page 13, Lines 358–363 as follows:

“We gained further support for this inference using the ancestral chromosome reconstruction tools Agora⁵⁵ and ANGES⁵⁶. These computational tools utilize current genome assemblies to model the most likely chromosomal configurations of ancestral states. The analyses revealed that the ancestral genome of *E. carolleae* and *E. gulfia* likely possessed 15 chromosomes, while the ancestral genome of the entire *E. affinis* complex likely contained 21 chromosomes (Fig. 6a).”

41. Lines 365-371: This is great stuff.

We appreciate this positive feedback.

42. Lines 391-392: Were the selection signatures found in *E. carolleae* and/or other species?

We have clarified on Page 15, Line 417 that the population genetic signatures of selection (e.g., allele frequency shifts) were found in *E. affinis* complex populations (including *E. carolleae*, *E. gulfia*, and *E. affinis* proper) in our previous studies.

43. Line 419: The chromosomal fusions occurred long before the selection implicated in recent saline to freshwater invasions in the wild and in laboratory studies, so the claim that such selection favored retention of these fusions is a stretch. Even if adaptive, why should the selection that hypothetically was involved in initial spread of the novel karyotypes be expected to continue to this day?

We thank the reviewer for this valuable comment. We think that this question has been addressed in our responses to Comments 31 and 32.

As clarified in these responses, our interpretation does not propose that contemporary selection is responsible for maintaining the chromosomal fusions themselves. Instead, we argue that the ancient fusion events reorganized the genome in ways that subsequently affected the selection responses of contemporary populations. Such a finding has been made empirically for chromosomal inversions in previous studies (Hoffmann and Rieseberg 2008; Lowry and Willis 2010; Joron et al. 2011; Schwander et al. 2014; Thompson and Jiggins 2014).

Specifically, the fusions established new linkage groups of functionally interacting ion transport-related genes within low-recombination genomic regions. Although these fusions may have been selectively neutral or only weakly advantageous at the time they arose, the genomic architecture that resulted could later become beneficial under changing environmental conditions, such as during historical and contemporary salinity transitions.

The recent selection signatures we detect between contemporary populations, through elevated F_{ST} , reduced nucleotide diversity, strong association with salinity, and elevated LD near fusion sites, reflect selection acting on adaptive alleles now embedded within this ancient chromosomal framework, rather than ongoing selection to maintain the fusions themselves.

44. Lines 434-438: The basis for this analysis is not exactly clear; what is the null expectation with the permutation tests?

We appreciate the reviewer's request for clarification regarding the basis of our permutation tests.

***In response to this comment, we have added a new sentence on Page 17, Lines 468–471 as follows:

“The null hypothesis for this comparison was that the 25 ion transport-related genes with signatures of selection would show no significant difference in mean or median positioning on the chromosomes compared to 25 genes randomly selected from 253 genes with putative ion transporting functions.”

The permutation tests involved randomly selecting gene sets 10^6 times and comparing their positioning using Welch's *t*-test and Mann–Whitney *U* test. The purpose was to determine whether the positioning of the selected 25 ion transporter genes was statistically different from that of other gene sets with no signatures of selection, thereby suggesting a non-random distribution influenced by selective pressures.

45. Lines 609-611: Has LD between the ion transporter genes in these supergenes been measured?

We appreciate the reviewer's question regarding the LD pattern.

We think that this question has been extensively addressed in our response to Comment 30.

Our LD analysis clearly revealed that all fusion sites exhibited significantly higher LD levels relative to the genome-wide background (Supplementary Figure 18). In addition, the fusion sites that contained significant enrichment of selection signatures showed even higher LD relative to all fusion sites. These results provide robust support for elevated LD at the fusion sites, especially the fusion sites with signatures of selection, and provide further support for our "supergene" hypothesis.

46. Lines 663-664: The language is vague here; what precisely is meant by "beneficial alleles were coming together"?

We have revised this sentence on Page 24, Lines 711–715 as follows:

“The extent of parallelism among the selection lines increased with more generations of selection, favoring the same freshwater-adapted alleles. This result indicated that recombination events, which would be more frequent with greater numbers of chromosomes, **brought together freshwater-adapted alleles within genomes**, likely facilitated by positive synergistic epistasis among the beneficial alleles (Stern et al. 2022).”

47. Figure 3: The proposed divergence times for the *E. affinis* complex species seem very high relative to their average nucleotide identities reported in lines 372-377, even if one accepts the lower 95% confidence bounds as the divergence times.

We appreciate the reviewer's astute comment regarding the reported divergence times and nucleotide identities. To clarify, the initially mentioned percentages represent the similarity of orthologous sequence pairs, not the entire genome.

***In response to this comment, we revised our analysis using MUMmer to accurately report the percentage of aligned sequences along with their average nucleotide similarity.

The updated results can be found on Page 15, Lines 403–409:

“Specifically, 357.2 Mb of the *E. carolleae* genome (67.5%) aligned with 356.2 Mb of the *E. gulfia* genome (68.4%), showing an average nucleotide similarity of 90.1%. In contrast, only 137.9 Mb of the *E. carolleae* genome (26.1%) aligned with 138.3 Mb of the *E. affinis* proper genome (20.6%), with an average nucleotide similarity of 87.4%. Likewise, only 128.1 Mb of the *E. gulfia* genome (24.6%) aligned with 128.5 Mb of the *E. affinis* proper genome (19.2%), with an average nucleotide similarity of 87.5%. These results further support that *E. carolleae* and *E. gulfia* have diverged more recently from a closely related common ancestor (closer than *E. affinis* proper), likely with 15 chromosomes (Fig. 6a).”

48. Figure 8: The authors seem to be analyzing the hypothesis that the supergene karyotypes that formed tens of millions of year ago persisted originally and remain today under strong directional selection for single species-specific karyotypes. If this is the intended argument, this hypothesis must be stated and supported more clearly, because this is a secondary hypothesis (i.e., this expectation doesn't necessarily follow from the adaptive fusion model). Furthermore, are the metrics chosen to detect selection the most appropriate? Why were *F_{st}* outliers and nucleotide diversity used rather than statistics explicitly developed to detect selection such as *dN/dS* or the M-K test? Is it possible that divergence and diversity are both affected by recombination rate more than by selection? And even if selection is the predominant evolutionary force, might it not be due to background selection (i.e., acting on a single gene in a low recombination window) rather than haplotype-wide selection as suggested by the idea of consolidated ion-channel gene supergenes?

Our hypothesis is **NOT** that the supergene karyotypes that formed tens of millions of years ago remain today under strong directional selection for single species-specific karyotypes. The dN/dS or MK tests would not be appropriate for our analyses, as we are examining genome-wide allele frequency shifts between contemporary populations in response to salinity change and mapping the SNP frequency shifts onto the chromosomes.

***To make our arguments clearer, we have revised our statements to more explicitly state our hypothesis and inference, which are the following: (1) Chromosomal fusions occurred in the deep evolutionary past, likely tens of millions of years ago; these fusions were possibly favored by selection in the past. (2) These ancient fusions reorganized the genome, clustering ion transporter genes into low-recombination regions. (3) These fusion-linked regions, enriched with ion transporter genes, facilitate coordinated allele inheritance and expression. (4) These aggregations of ion transporter genes are targets of natural selection during contemporary salinity transitions, in terms of allele frequency shifts between ancestral saline and freshwater invading populations.

This argumentation above is similar to that made for chromosomal inversions, where the ancient inversions have repositioned adaptive loci, and this repositioning and reduced recombination impacts contemporary population responses to selection. Such examples include studies on temperature adaptation in *Drosophila* and studies on butterflies and other taxa (Hoffmann and Rieseberg 2008; Lowry and Willis 2010; Joron et al. 2011; Yeaman 2013; Schwander et al. 2014; Thompson and Jiggins 2014).

Regarding the reviewer's feedback on our choice of metrics, we utilized F_{ST} and θ_{π} outliers as these are well-established indicators for detecting directional selection in terms of SNP (allele) frequency shifts between populations adapted to different environments. Recognizing the limitations of these metrics, particularly their sensitivity to recombination rates, we have also incorporated BayPass and Bayescan approaches into our analyses. Our Bayescan results overlapped with our BayPass results, such that we report only our BayPass results. These statistical approaches allow us to more accurately associate allele frequency changes with salinity differences and take into account population structure. As such, these approaches provide a robust framework for identifying genuine selection signatures amidst potential demographic noise. This reanalysis ensures that our findings more reliably reflect the adaptive dynamics between freshwater and saline environments.

Minor comments:

49. Figure 8: Panels e and f are somewhat confusing. Is there some simpler, more intuitive way to present these data?

We appreciate the reviewer's suggestion to enhance the clarity of Figures 8e and 8f.

***In response to this comment, we have revised these panels (now Figures 7e and 7f) to focus exclusively on the fusion sites showing significant enrichment of selection signatures, thereby streamlining the presentation and making the data more intuitive.

We retained the histogram format, as we consider it the most effective way to clearly illustrate the background genome-wide distribution of selection signatures *versus* the significant enrichment of selection signatures at specific fusion sites.

50. Figure 5: Consider moving Figure 5 to supplement (or Figure 7?) and combining key synteny results of Figure 6 and Figure 7.

We thank the reviewer for this suggestion to reorganize the figures for better clarity and presentation.

***In response to this comment, we have combined and reorganized Figure 5 with Figure 7a as a revised Figure 5. We also integrated the key synteny results of Figures 7b–d and Figure 6 as a revised Figure 6.

51. Discussion: This section seems overly long.

We appreciate this reviewer's comment regarding the length of the Discussion section.

***In response to this comment, we have extensively rewritten and shortened the Discussion section. Specifically, we greatly reorganized the Discussion section by consolidating the subsections by topic, rather than by results. Thus, we removed a lot of redundancy. For example, we originally had separate subsections within the Discussion on the 'enrichment of selection signatures at the fusion sites' and on the 'repositioning of key ion transporter genes,' such that we repeated several discussion points multiple times. Now the Discussion section is far more streamlined.

52. Overall: We recommend that the authors be sure to highlight which species is or are being discussed whenever relevant. Also, it should be made clear which analyses pertain to contemporary versus ancient evolutionary events, and care should be taken to revise the paper so that the reader does not mistakenly believe that the chromosomal fusions under consideration happened recently.

We thank the reviewer again for these helpful recommendations.

***In response to this comment, we have revised the manuscript to explicitly indicate which sibling species is being discussed at all relevant points. Additionally, we have carefully revised the text to clearly distinguish between contemporary and ancient evolutionary events.

Reviewer #4 (Remarks to the Author):

We sincerely appreciate the thoughtful comments in this co-review.

Response to the Reviewers' Comments:

Reviewers' comments are in black, whereas authors' responses are in dark blue.

Reviewer #1 (Remarks to the Author):

The author has addressed all of my suggestions and concerns in this revision.

Reviewer # 1

We sincerely appreciate the reviewer's kind assessment and thoughtful feedback throughout the review process.

Reviewer #2 (Remarks to the Author):

The authors have done a very thorough and thoughtful job in addressing the extensive comments by the reviewers. The figures, results, and discussion are more streamlined and accessible. I recommend the manuscript for publication.

We really appreciate the reviewer's thorough evaluation and supportive recommendation of our manuscript.

Reviewer #3 (and #4) (Remarks to the Author):

In the present manuscript, Du et al have substantially revised their analysis of copepod genome evolution, providing useful clarification and specificity to the conclusions, as well as conducting numerous new analyses. Overall, we feel that the manuscript is greatly improved and, in our opinion, that this work merits publication in Nature Communications, pending some remaining modest revisions. In general, the work is excellent, and we commend the authors on their achievement.

In our previous review letter, we raised a few minor concerns stemming from semantic issues and from a handful of missing analyses that could greatly strengthen the core observations. The authors addressed these comments. Resolution of the two semantic misunderstandings by describing deeply-diverged lineages as 'species' rather than 'populations' now makes the findings easier to understand, and use of a stricter definition of supergenes and limiting discussion of them is also helpful. Our minor comments about analyses that could strengthen the work also were addressed; the addition of new selection analyses and, especially, tests for suppressed recombination, have strengthened the evidence for the key findings.

We also raised a major concern about the earlier draft: certain passages in the text led us to believe the authors were using contemporary signatures of selection as a test of whether ancient chromosomal fusions were positively selected once they arose. We appreciate that the authors now intend to present these findings in two separate analyses. First, they present analyses of ancient evolution that strengthen and extend this group's prior claim that the chromosomal fusions were non-random and therefore presumably adaptive – possibly due to a particular bias for clustering functionally-similar ion channel genes in low-recombination regions near centromeres. The key new finding of the current paper comes from analyses of contemporary evolution showing that these centromeric ion channel clusters do indeed display low recombination and are hotspots of recent selection that may facilitate the transition between salt and fresh water habitats. A critical goal of the text, then, should be to make it clear that no causal link between a) and b) can be drawn with the present data: the selective forces that caused certain types of chromosomal fusions to be preferentially retained must be clearly flagged as unknown, and the focus should instead be

on the adaptive value that the presence of these fusions provided to contemporary populations. We appreciate the possibility mentioned in the response letter: The “long evolutionary history of salinity fluctuations in the genus *Eurytemora*” might be a common cause for ancient as well as contemporary selection. This is a plausible hypothesis that merits description in the Discussion, but we ask that it be kept out of the key results and conclusions (unless direct tests of this idea can be devised).

We sincerely thank the reviewers for their thoughtful, detailed, and constructive evaluation of our revised manuscript. We appreciate the recognition of the improvements made by our substantial revisions, including the additional analyses undertaken to clarify our conclusions and strengthen the core findings.

We agree with the reviewers’ point that our results should clearly distinguish between the ancient origin of chromosomal fusions and the contemporary selection acting on the genomic regions influenced by these structural changes.

In response to the remaining suggestion, we have carefully revised the relevant sections to ensure that the distinction between ancient and contemporary evolutionary processes is explicit and that our conclusions do not overstate any direct causal link between them.

Only relatively minor issues remain to resolve this ambiguity. Therefore, we list several opportunities to improve the manuscript further, and we organize these into three categories:

1) In certain parts of the Abstract, Introduction, Results, and Discussion, ancient and contemporary evolution are discussed back-and-forth in a way that can make them seem conflated. We suggest that the authors attempt to reorganize all relevant sections by discussing ancient evolution first (i.e., evidence that fusions were adaptive for an unknown reason), then discuss contemporary evolution second (i.e., regardless of why they exist, the presence of these fusions has facilitated contemporary evolution). The discussion can then discuss the possible causal link between these evolution in these two epochs, but this should be mentioned as a hypothesis rather than a key conclusion of the work.

We thank the reviewers for this thoughtful suggestion to improve the logical order and clarity of the text.

***In response to this comment, we have carefully reorganized relevant sections of the Abstract, Introduction, Results, and Discussion to clearly separate the presentation of ancient versus contemporary evolutionary processes. Specifically, we clearly present the evidence and interpretation for ancient chromosomal fusions first, followed by the analyses and results demonstrating how these pre-existing structural features may have facilitated contemporary selection responses (see specific responses below).

In the Discussion section, we explicitly frame the possible causal link between ancient genome architecture and contemporary adaptation as a plausible hypothesis rather than as a definitive conclusion, in line with the reviewers’ suggestion.

2) We are skeptical of the specific findings about the timeline of chromosomal fusions. We understand that such ancient events are inherently difficult to reconstruct, so we don’t feel that it is necessary to unambiguously resolve this in the paper. If the authors would like to provide additional analyses that do resolve it, that would be great, but we are not requesting this; it is perfectly acceptable to publish the findings as is, as long as the authors soften some of their conclusions and add clarifying text to emphasize that multiple scenarios of chromosomal fusion remain possible.

We thank the reviewers for this thoughtful comment and for recognizing the inherent challenges of reconstructing the precise timeline of ancient chromosomal fusions. We agree that multiple plausible scenarios remain possible.

***In response, we have carefully revised the relevant text in the Results section to soften our conclusions about the exact sequence and timing of these fusion events. We have also added statements to explicitly acknowledge the uncertainty surrounding the ancestral karyotype of *E. gulfia* and *E. carolleae*. We now clearly note that alternative scenarios for the fusion history cannot be ruled out given the current data (see specific responses to Comment 12).

3) Ambiguity in the language of some passages hinders the reading and comprehension. For instance, the passage mentioned above, the “long evolutionary history of salinity fluctuations in the genus *Eurytemora*” should be amended to the “long evolutionary history of salinity fluctuations experienced by copepods [or individuals] in the genus *Eurytemora*” because the fluctuations clearly do not occur in the genus but in the environment that the members of a given taxon inhabit. Similar or other problems arise on lines 70-71 (distribution...are), line 107 (a species complex cannot be a grazer, only its constituent individual members), line 124 (I doubt that entire populations were transported), line 135 (what sibling species? The term generally only makes sense with respect to another species), and elsewhere.

We thank the reviewers for pointing out the ambiguity or imprecision of some of our statements.

***In response to this comment, we have carefully revised several of our statements to ensure accuracy and clarity. These changes could be found on Page 3, Lines 70–71; Page 4, Lines 109–111, 126; Page 5, Lines 137, 140; Page 25, Line 730; and Page 26, Line 762.

Minor concern 1

4. In the Abstract, the sentence at Lines 53-55 pertains to ancient evolution, then 55-57 pertains to contemporary evolution, then 57-59 returns to ancient evolution. Would it be possible to put the two ancient evolution sentences together, then the contemporary evolution sentence to follow?

We thank the reviewers for this helpful suggestion.

***In response, we have rearranged the sentences, on Page 2, Lines 53–59, so that the statements describing ancient evolutionary processes are grouped together, followed by the sentence describing contemporary evolution.

Lines 83-84 are great.

We appreciate this positive feedback.

5. Lines 102-104: We feel that this expresses the key novelty of the paper. Past work and work in preparation by the authors has focused on the non-random nature of these fusions. It is helpful that, to a modest degree, the authors replicated and extended those findings here. However, the primary focus should be as discussed in this paper: asking whether those ancient adaptive fusions are relevant to contemporary selection.

We thank the reviewers for this helpful comment.

***In response to this comment, we have added a new sentence to clarify the main focus of our study on Page 4, Lines 105–106 as follows:

“As such, a key question is whether these chromosomal fusion events are associated with adaptive evolution.”

6. Lines 146-151: This section is extremely important because it outlines the motivation for this study, but it does not do a good job of clarifying that there are actually two hypotheses: 1) whether ancient fusions were adaptive, and 2) whether contemporary populations are experiencing selection at these fusion sites. We hope the authors can rephrase this to make it more clear.

We thank the reviewers for this valuable comment.

We agree that this section should clearly distinguish between the two related hypotheses: (1) that ancient chromosomal fusions were adaptive in the past by promoting physical linkage of beneficial loci, and (2) that it appears that these fusion sites continue to experience natural selection in contemporary populations.

In response, we have revised the text on Page 5, Lines 147–154 as follows:

“As such, we hypothesized that ancient chromosomal fusions in the *E. affinis* complex facilitated adaptation in the past by bringing together beneficial loci into close physical linkage. In addition, such linkages, which reduce recombination and help maintain co-adapted gene complexes, might continue to be beneficial and constitute the targets of natural selection during salinity change in contemporary populations^{5,6,10,30,47}.

Thus, to address these hypotheses and explore patterns of genome architecture evolution, we examined the association between ancient chromosomal evolution events, particularly chromosomal fusions, and genome-wide signatures of natural selection in contemporary populations.”

7. Lines 155-159: Similar issues as before. First, (1) and (3) are ancient but (2) is contemporary; can this be re-ordered? Second, can (1) and (3) be broken into a separate sentence from (2) to emphasize the distinct hypotheses for these distinct epochs?

We thank the reviewers for this comment.

***In response, we have reordered and rephrased these sentences to clarify this distinction. The revised text is on Page 5, Lines 154–160 as follows:

“As such, our specific goals were to explore patterns of chromosomal evolution across the three genomes from sibling species of the *E. affinis* species complex and to determine how functionally beneficial loci (under selection in the past) might have become repositioned following ancient fusion events. In addition, we determined whether contemporary populations show genomic signatures of natural selection in response to salinity change (e.g., allele frequency shifts, association with salinity, LD) at the chromosomal fusion sites.”

8. Lines 415-418: This sentence conflates the distinct timescales so thoroughly that it is likely to mislead the average reader. We feel that the discussion of contemporary evolution is unwarranted in a section titled “Chromosomal fusions reposition ion transport-related genes within genomes;” we request that this section be used solely to discuss ancient events so that the contemporary evolution can be solely and separately discussed in the following section.

We thank the reviewers for this helpful suggestion.

***In response, we have removed the sentence at Lines 415–418 to prevent conflating ancient and contemporary timescales within this section. Instead, we now emphasize that many of the repositioned genes belong to expanded and unique gene families in the *E. affinis* complex, with important roles in models of ion uptake from the environment. As recommended, the discussion of contemporary selection is presented exclusively in the following section.

9. Lines 437-439: Same issue; the inclusion of this sentence amidst a discussion of events that happened much earlier in geological time is confusing. We suggest that this citation of previous work be moved to the introductory text describing your analyses of contemporary evolution.

We thank the reviewers for this comment.

*** In response, we have revised the sentence on Page 16, Lines 441–444 as follows:

“Specifically, the chromosomal fusions resulted in significant shifts from the edges of the *E. affinis* proper chromosomes toward the central regions of the *E. carolleae* chromosomes for 25 ion transport-related gene paralogs and subunits, which had been identified as targets of natural selection in our prior studies^{30,31}.”

We have retained the references here because the 25 ion transport-related genes, which were identified as targets of selection in our previous work, directly motivate the statistical test presented in this section.

10. Line 450: Can a new title be given for this and the following paragraphs? The previous title no longer applies; we aren’t talking about repositioning of genes in this part, but instead about the selection those repositioned genes are experiencing.

We thank the reviewers for this suggestion.

We would like to clarify that this section indeed focuses on the repositioning of selected ion transport-related genes as a result of chromosomal fusions. The analyses described here test whether this repositioning is non-random and statistically significant. Therefore, we believe the current section title appropriately reflects the content.

11. Lines 473-475: This summary sentence is out of place; it belongs at line 449, by which point the non-random nature of the repositioning was already established. A summary sentence for this section should instead summarize the new results (that those repositioned genes are under contemporary selection), then set up the subsequent attempt to understand why they are under selection.

We thank the reviewer for raising this point.

***In response, we have moved the summary sentence forward to Line 454, where the non-random repositioning is first established. In addition, we have added a new statement to this section, on Page 17, Lines 480–483, to summarize the new finding and set up the next section:

“Together, these results demonstrate that ion transport-related genes under contemporary selection for salinity adaptation tend to be non-randomly centralized within chromosomes of the *E. carolleae* genome. This positioning, shaped by ancient chromosomal fusion events, likely continues to influence contemporary selection responses.”

Lines 607-619: This section does an excellent job at clarifying the key ideas.

We appreciate this positive comment.

Minor concern 2

12. We request some clarification on the parallel chromosome reduction hypothesis positing that the ancestor of *gulfia*+*carolleeae*, the ancestor of the *E. affinis* clade, and extant *E. affinis* all had 15 chromosomes. We agree with the argument (Lines 397-399) that it is not parsimonious to propose 7 chromosomes in the *gulfia*+*carolleeae* ancestor, and understand that the statistical package used inferred a 15-chromosome ancestor. However, this claim seems to suggest an implausible degree of parallel evolution.

Based on the plots in Figure 6b,c, and allowing for lineage-specific inversions, both *gulfia* and *carolleeae* seem to share six fusions relative to *E. affinis*: 1+2, 3+4, 5+6, 8+9, 12+13, and 14+15. Manually checking each of these in Figure 6d seems consistent with a homologous origin: if we allow for ~2-3 inversions, all of these fusions seem to represent near-perfect alignment between the two species. In contrast, if the fusions happened at different breakpoints, wouldn't we expect Figure 6d to show small slivers of sequence at each fusion that align elsewhere (or not at all)? Based on this reasoning, our quick guess was that the ancestor of *gulfia*+*carolleeae* had 9 chromosomes. *E. carolleeae* would then have 5 lineage-specific fusions (1+2 to 3+4 | 7 to 5+6 | 10 to 8+9 | 11 to 8+9+10 | 12+13 to 14+15). *E. affinis* would have two lineage-specific fusions (10 to 5+6 | 7 to 11).

We are not sure whether the above hypothesis is consistent with your data or if your evidence for a 15-chromosome ancestor is definitive already (e.g., perhaps you have observed that breakpoints always differ in the “shared” fusions we proposed above?). If it is definitive, this needs to be made clearer in the Results, and the extraordinarily nature of the parallel fusions should be highlighted more directly in the Discussion.

If it is not definitive, (i.e., a scenario like the one we outlined above is also possible) then we request that the section be altered to emphasize the fact that multiple ancestral arrangements are consistent with the data. This would not undermine the major findings of the paper because the key conclusion in Lines 397-399 is certainly true even if the 15-chromosome ancestor is not supported. Note that if this path is taken, some of the interpretation elsewhere in the paper will also need to be softened (i.e., Line 52 of the Abstract) to make it clear that the precise number of independent versus shared fusions is not known. Alternatively, it would of course be great if the authors find a way to unambiguously determine the ancestral chromosome number (i.e., by using a breakpoint analysis), but the authors can best judge whether new analyses of that kind would represent feasible and worthwhile additions at this late stage.

We thank the reviewer for this thoughtful and careful analysis of our proposed parallel chromosomal reduction scenario. We appreciate the reviewer's detailed check of the fusion breakpoints and the alternative hypothesis suggesting a 9-chromosome ancestor for *E. gulfia* + *E. carolleeae*.

At present, our inference of a 15-chromosome ancestor for *E. gulfia* + *E. carolleeae* was based on the bioinformatic reconstruction using Agora and ANGES. However, we agree that this alone does not definitively rule out alternative scenarios, especially given the plausible homologous breakpoints highlighted by the reviewer's careful reasoning.

In fact, a karyotype study of an *E. affinis* complex population found in Rhode Island uncovered a chromosome number of ten (Vaas and Pesch, 1984). Given the location of this sample, it is possible that this population is from the North Atlantic clade (teal branch in Figure 1). Thus, it is also plausible that 15 chromosomes fused into 10 chromosomes, which then independently fused

into 7 and 4 chromosomes. At this time, it remains challenging to clearly reconstruct the precise ancestral karyotype for *E. gulfia* + *E. carolleae*.

***In response, we have revised the results and discussion to explicitly acknowledge this uncertainty. Specifically, we now clarify on Page 15, Lines 401–408:

“While some fusions involving *E. carolleae*’s Chromosomes 2, 3 and *E. gulfia*’s Chromosomes 3, 4, 5 might have arisen independently in each sibling species, it remains possible that certain fusions were shared and inherited from a more recent common ancestor. For example, *E. carolleae*’s Chromosomes 1, 4 and *E. gulfia*’s Chromosomes 1, 2, 6, 7 might have shared chromosomal fusions originating from an intermediate ancestor possessing nine or ten chromosomes (Fig. 6a, n=?). Notably, a ten-chromosome karyotype was documented in a population from Rhode Island, USA that may belong to the North Atlantic clade⁵⁷, which is more closely related to *E. carolleae* and *E. gulfia* than to *E. affinis* proper^{22,25} (Fig. 1).”

We have softened the wording by revising all uses of “independent chromosomal fusions” to simply “chromosomal fusions” throughout the text in this manuscript. We also added statements to reflect that multiple ancestral scenarios remain possible given the current resolution. For example, we have added a statement on Page 15, Lines 417–419 as follows:

“These results further support that *E. carolleae* and *E. gulfia* have diverged more recently from a closely related common ancestor (closer than *E. affinis* proper), likely with 15 chromosomes or an unknown intermediate number of chromosomes (Fig. 6a).”

In addition, we have updated Figure 6a to include an additional circle on the relevant node to visually indicate the uncertainty regarding the ancestral chromosome number for *E. carolleae* and *E. gulfia*.

Summary

The authors have done an admirable job of substantially strengthening their work. Our minor comments from the prior draft were addressed and our major comment was largely addressed, with only a few minor concerns remaining. If these issues can be resolved, we think the paper will become a valuable contribution to the field and merits publication in Nature Communications.

We thank the reviewers again for their thoughtful and constructive comments, which have greatly strengthened the quality and rigor of our manuscript.

In the present manuscript, Du et al generate and analyze two new chromosome-level genome assemblies from a copepod species complex, then analyze these in tandem with a prior assembly to study the role of natural selection on genome architecture in this clade. The authors argue that selection-driven chromosomal fusions can explain their prior observation that certain ion transporter genes, which show signatures of contemporary selection, are clustered within the genome. Consequently, they propose that fusion-mediated supergene formation provides an explanation for rapid evolution in response to environmental change, and they argue that their results provide conclusive evidence for this mechanism not found in prior studies.

In general, the empirical work appears to be very thorough and well thought out, the manuscript is quite well written, the figures are useful and mostly well designed, and the authors provide a variety of interesting observations. The inference that the three clades under consideration actually represent evolutionarily ancient species is compelling and intriguing. We found the reconstruction of karyotype evolution and the depiction of how chromosomes have fused to be a particular highlight of the study. Finally, the fact that genes near fusion sites appear to be under selection in only one of two sibling species that experienced such fusions is thought provoking.

Nonetheless, we find ourselves unconvinced by several of the central conclusions of the paper. We are sympathetic to the authors' belief that the weight of the evidence supports a "supergene formation" hypothesis, but we feel that this remains a hypothesis and that the basic claims made by the authors cannot be confirmed based on the current data and analyses. We cannot recommend publication of the manuscript in its present form but suggest that a different but equally consequential framing of the results can be adopted to make the paper more suitable for publication in a journal of the standing of *Nature Communications*. However, all of our recommendations would require a radical alteration to the framing and the interpretation of the data (e.g., focusing on the morphological stasis or providing a more descriptive study of karyotype evolution). In the spirit of reviewing the paper as intended, we do not provide detailed suggestions for how this might be done, but instead comment on whether or not the central conclusions in the current report are supported by the data and, if not, what new analyses might address this.

Our four major comments are as follows:

- By all accounts the fusion of chromosomes was an ancient event, so what is the justification for using genetic data from contemporary populations (ca. 65-80 years diverged) to infer whether the chromosomal fusions were maintained due to selection?
- More generally, the paper is framed by, and uses methods derived from, a population genomics perspective to make claims about the deep past (at least tens of millions of years) when it is not clear that the basic assumptions of these methods are met. For instance, why would one expect expansions in ion transporter gene copy number to underpin colonization of different salinity environments when the timescales involved are sufficient for natural selection to elevate the frequency of rare regulatory or amino acid mutations that improve the functions of ancestral genes? This is not to say that changes in copy number are irrelevant, just that the assumption that they must be under selection (rather than neutral) does not seem warranted. Therefore, the central idea from the title

and abstract – that supergenes can facilitate rapid adaptation to novel environments – is questionable in light of the results presented.

- There are a few issues regarding supergenes. First, on a semantic level, it appears that the authors are discussing fixed differences between species, not intraspecific polymorphisms, so ‘supergene’ probably is not the appropriate term. Regardless, the defining features of supergenes are: a) Mendelian inheritance of haplotypes underlying complex traits, and b) suppression of recombination linking multiple non-neutral alleles that presumably confer the traits. As far as we are aware, these features were not examined in this study (proximity to the centromere is not evidence of suppressed recombination). If the loci under consideration are not supergenes in the sense we describe above (but rather, for instance, are hotspots of adaptive evolution), that would severely undermine the authors’ key conclusions as it would not suggest any specific mechanism allowing these loci in particular to facilitate rapid evolution.
- Beyond the concerns addressed above, it is difficult to understand why one particular mechanism (selection) is privileged to explain the observation that chromosomal fusions have been maintained for tens of millions of years and ion channel genes have become clustered. Are hypothetical signatures of selection in contemporary populations (e.g., elevated F_{st} and depressed nucleotide diversity) really expected in ancient macromutations involving fundamental changes in chromosomal architecture? The issue of recombination seems especially relevant, given that fusion points now lie in proximity to centromeres, and the fact that the correspondence between chromosome fusion and “contemporary selection” was observed in only one species seems particularly problematic. Is it possible that this species is just an outlier in its recombination mechanisms or rates, for instance, leading to patterns of divergence and diversity that spuriously resemble the outcome of selection?

Specific comments:

Lines 49-51: With such deep time involved in the chromosomal fusions, can we really say they facilitate adaptation? The same question pertains to the subsequent statement (rapid colonization of novel habitats).

Lines 73-74: Are “genomic islands” being equated with “supergenes” here? In our view, the concepts differ fundamentally; a supergene exists as an intraspecific polymorphism featuring little or no recombination that enables Mendelian inheritance of a polygenic phenotype. Genomic islands, on the other hand, typically are conceived of as nearly-fixed differences between partially reproductively isolated populations or species that are associated with minimal introgression at a block of the genome (probably a better framework for the topic of this paper). Supergenes generally are sparsely distributed throughout the genome (commonly even singular entities), while genomic islands are expected to occur widely throughout diverging genomes. Supergenes may give rise to genomic islands, but other mechanisms can also do so, and the evolutionary interpretation of the observation will differ based on which mechanism is implicated.

Minor comment: Lines 97-102 are a bit superfluous and might be better in the Discussion. If we understood correctly, the authors are suggesting that the results of this paper are more conclusive than prior findings (e.g., Reference 5), but we are not convinced that this is the case.

Line 111: Members of the *E. affinis* are complex grazers of what?

Line 114: Are the “large genetic divergences” among regional populations referred to unique to the mtDNA or does this apply to the nuclear genome as well? It is clear from what comes later that there is indeed very deep nuclear DNA divergence as well. It is difficult to reconcile such divergence with the absence of complete reproductive isolation between some of the presumed cryptic species in this complex (discussed further in lines 707-715). In any case, the authors should provide a scale bar for the amount of sequence divergence in Fig. 1b.

Lines 171-178, 246-274: The introduction of microevolution here for the first time is confusing given the ~65my timescale relevant to the chromosomal fusion events. Characterization of the *E. affinis* complex as comprising distinct “sibling species” helps mitigate some of this confusion and so should be made clearer earlier in the paper and emphasized throughout. We recommend that these groups consistently be referred to as species, not as clades, and that the authors amend the Abstract and Introduction to immediately make the deep evolutionary timescale clear to the reader.

Lines 281-283: The scarcity of genomic data from taxa or populations of varying evolutionary distance from the focal taxa is an important limitation for any claims that genomic elements have been lost or gained in the ancestor of the *E. affinis* complex and represents, of course, a common problem in reconstructing evolutionary transformation series.

Line 347: This might be the place to mention that the nuclear genome-based phylogeny agrees well with the mtDNA network rooted with the outgroups and displayed as a tree.

Lines 348-349; 354: Is 15 or 21 chromosomes reconstructed as the ancestral karyotype among the *E.affinis* complex species?

Lines 365-371: This is great stuff.

Lines 391-392: Were the selection signatures found in *E. carolleae* and/or other species?

Line 419: The chromosomal fusions occurred long before the selection implicated in recent saline to freshwater invasions in the wild and in laboratory studies, so the claim that such selection favored retention of these fusions is a stretch. Even if adaptive, why should the selection that hypothetically was involved in initial spread of the novel karyotypes be expected to continue to this day?

Lines 434-438: The basis for this analysis is not exactly clear; what is the null expectation with the permutation tests?

Lines 609-611: Has LD between the ion transporter genes in these supergenes been measured?

Lines 663-664: The language is vague here; what precisely is meant by "beneficial alleles were coming together"?

Figure 3: The proposed divergence times for the *E. affinis* complex species seem very high relative to their average nucleotide identities reported in lines 372-377, even if one accepts the lower 95% confidence bounds as the divergence times.

Figure 8: The authors seem to be analyzing the hypothesis that the supergene karyotypes that formed tens of millions of year ago persisted originally and remain today under strong directional selection for single species-specific karyotypes. If this is the intended argument, this hypothesis must be stated and supported more clearly, because this is a secondary hypothesis (i.e., this expectation doesn't necessarily follow from the adaptive fusion model).

Furthermore, are the metrics chosen to detect selection the most appropriate? Why were F_{st} outliers and nucleotide diversity used rather than statistics explicitly developed to detect selection such as dN/dS or the M-K test? Is it possible that divergence and diversity are both affected by recombination rate more than by selection? And even if selection is the predominant evolutionary force, might it not be due to background selection (i.e., acting on a single gene in a low recombination window) rather than haplotype-wide selection as suggested by the idea of consolidated ion-channel gene supergenes?

Minor comments:

Figure 8: Panels e and f are somewhat confusing. Is there some simpler, more intuitive way to present these data?

Figure 5: Consider moving Figure 5 to supplement (or Figure 7?) and combining key synteny results of Figure 6 and Figure 7.

Discussion: This section seems overly long.

Overall: We recommend that the authors be sure to highlight which species is or are being discussed whenever relevant. Also, it should be made clear which analyses pertain to contemporary versus ancient evolutionary events, and care should be taken to revise the paper so that the reader does not mistakenly believe that the chromosomal fusions under consideration happened recently.